# Stochastic Constrained DRO with a Complexity Independent of Sample Size

**Qi Qi**[*]                                                                          *qi-qi@uiowa.edu*
*Department of Computer Science*
*The University of Iowa, Iowa City, IA 52242, USA*
**Jiameng Lyu**[*]                                                          *lvjm21@mails.tsinghua.edu.cn*
*Department of Mathematical Sciences*
*Tsinghua University, Beijing, 100084, China*
**Kung-Sik Chan**                                                          *kung-sik-chan@uiowa.edu*
*Department of Statistics and Actuarial Science*
*The University of Iowa, Iowa City, IA 52242, USA*
**Er-Wei Bai**                                                                *er-wei-bai@uiowa.edu*
*Department of Electrical and Computer Engineering*
*The University of Iowa, Iowa City, IA 52242, USA*
**Tianbao Yang**                                                          *tianbao-yang@tamu.edu*
*Department of Computer Science & Engineering*
*Texas A&M University, College Station, TX 77843, USA*

**Reviewed on OpenReview:** *https://openreview.net/forum?id=VpaXrBFYZ9*

## Abstract

Distributionally Robust Optimization (DRO), as a popular method to train robust models against distribution shift between training and test sets, has received tremendous attention in recent years. In this paper, we propose and analyze stochastic algorithms that apply to both non-convex and convex losses for solving Kullback–Leibler divergence constrained DRO problem. Compared with existing methods solving this problem, our stochastic algorithms not only enjoy competitive if not better complexity independent of sample size but also just require a constant batch size at every iteration, which is more practical for broad applications. We establish a nearly optimal complexity bound for finding an $\epsilon$-stationary solution for non-convex losses and an optimal complexity for finding an $\epsilon$-optimal solution for convex losses. Empirical studies demonstrate the effectiveness of the proposed algorithms for solving non-convex and convex constrained DRO problems.

## 1 Introduction

Large-scale optimization of DRO has recently garnered increasing attention due to its promising performance on handling noisy labels, imbalanced data and adversarial data (Namkoong & Duchi, 2017; Zhu et al., 2019; Qi et al., 2020a; Chen & Paschalidis, 2018). Various primal-dual algorithms can be used for solving various DRO problems (Rafique et al., 2021; Nemirovski et al., 2009). However, primal-dual algorithms inevitably suffer from additional overhead for handling a $n$ dimensionality dual variable, where $n$ is the sample size. This is an undesirable feature for large-scale deep learning, where $n$ could be in the order of millions or even billions. Hence, a recent trend is to design dual-free algorithms for solving various DRO problems (Qi et al., 2021; Jin et al., 2021; Levy et al., 2020).

In this paper, we provide efficient dual-free algorithms solving the following constrained DRO problem, which are still lacking in the literature,

$$\min_{\mathbf{w} \in \mathcal{W}} \max_{\{\mathbf{p} \in \Delta_n : D(\mathbf{p}, \mathbf{1}/n) \leq \rho\}} \sum_{i=1}^{n} p_i \ell_i(\mathbf{w}) - \lambda_0 D(\mathbf{p}, \mathbf{1}/n), \tag{1}$$

---

[*]Equal Contribution.

where $\mathbf{w}$ denotes the model parameter, $\mathcal{W}$ is closed convex set, $\Delta_n = \{\mathbf{p} \in \mathbb{R}^n : \sum_{i=1}^n p_i = 1, p_i \geq 0\}$ denotes a $n$-dimensional simplex, $\ell_i(\mathbf{w})$ denotes a loss function on the $i$-th data, $D(\mathbf{p}, \mathbf{1}/n) = \sum_{i=1}^n p_i \log(p_i n)$ represents the Kullback–Leibler (KL) divergence measure between $\mathbf{p}$ and uniform probabilities $\mathbf{1}/n \in \mathbb{R}^n$, and $\rho$ is the constraint parameter, and $\lambda_0 > 0$ is a small constant. A small KL regularization on $\mathbf{p}$ is added to ensure the objective in terms of $\mathbf{w}$ is smooth for deriving fast convergence.

There are several reasons for considering the above constrained DRO problem. First, existing dual-free algorithms are not satisfactory (Qi et al., 2021; Jin et al., 2021; Levy et al., 2020; Hu et al., 2021). They are either restricted to problems with no additional constraints on the dual variable $\mathbf{p}$ except for the simplex constraint (Qi et al., 2021; Jin et al., 2021), or restricted to convex analysis or have a requirement on the batch size that depends on accuracy level (Levy et al., 2020; Hu et al., 2021). Second, the Kullback–Leibler divergence measure is a more natural metric for measuring the distance between two distributions than other divergence measures, e.g., Euclidean distance. Third, compared with the KL-regularized DRO problem without constraints, the above KL-constrained DRO formulation allows it to automatically decide a proper regularization effect that depends on the optimal solution by tuning the constraint upper bound $\rho$. In other words, solving the constrained DRO with $\rho$ offers the capability of optimizing the temperature parameter $\lambda$ in Eq. (2), which corresponds to the log-sum-exponential form with a temperature parameter $\lambda$ is widely used in many ML/AI methods, e.g., constrastive self-supervised learning (Yuan et al., 2022; Qiu et al., 2023b). Empirical studies have demonstrated that selecting an appropriate value for $\lambda$ is crucial for achieving good performance (Goel et al., 2022; Li et al., 2021a; Radford et al., 2021). Therefore, solving the constrained distributionally robust optimization problem provides the added advantage of identifying an optimal temperature during the training process.

The question to be addressed is the following:

> *Can we develop stochastic algorithms whose oracle complexity is optimal for both convex and non-convex losses, and its per-iteration complexity is independent of sample size n without imposing any requirements on the (large) batch size in the meantime?*

We address the above question by (i) deriving an equivalent primal-only formulation that is of a compositional form; (ii) designing two algorithms for non-convex losses and extending them for convex losses; (iii) establishing an optimal complexity for both convex and non-convex losses. In particular, for a non-convex and smooth loss function $\ell_i(\mathbf{w})$, we achieve an oracle complexity of $\widetilde{\mathcal{O}}(1/\epsilon^3)$[1] for finding an $\epsilon$-stationary solution; and for a convex and smooth loss function, we achieve an oracle complexity of $\mathcal{O}(1/\epsilon^2)$ for finding an $\epsilon$-optimal solution. We would like to emphasize that these results are on par with the best complexities that can be achieved by primal-dual algorithms (Huang et al., 2020; Namkoong & Duchi, 2016). But our algorithms have a per-iteration complexity of $\mathcal{O}(d)$, which is independent of the sample size $n$. The convergence comparison of different methods for solving (1) is shown in Table 1.

To achieve these results, we first convert the problem (1) into an equivalent problem:

$$\min_{\mathbf{w} \in \mathcal{W}} \min_{\lambda \geq \lambda_0} F(\mathbf{w}, \lambda) := \lambda \log \left( \frac{1}{n} \sum_{i=1}^n \exp \left( \frac{\ell_i(\mathbf{w})}{\lambda} \right) \right) + (\lambda - \lambda_0)\rho. \tag{2}$$

By considering $\mathbf{x} = (\mathbf{w}^\top, \lambda)^\top \in \mathbb{R}^{d+1}$ as a single variable to be optimized, the objective function is a compositional function of $\mathbf{x}$ in the form of $f(g(\mathbf{x}))$, where $g(\mathbf{x}) = \left[ \lambda, \frac{1}{n} \sum_{i=1}^n \exp \left( \frac{\ell_i(\mathbf{w})}{\lambda} \right) \right] \in \mathbb{R}^2$ and $f(g) = g_1 \log(g_2) + g_1 \rho$. However, there are several challenges to be addressed for achieving optimal complexities for both convex and non-convex loss functions $\ell_i(\mathbf{w})$. First, the problem $F(\mathbf{x})$ is non-smooth in terms of $\mathbf{x}$ given the domain constraint $\mathbf{w} \in \mathcal{W}$ and $\lambda \geq \lambda_0$. Second, the outer function $f(g)$'s gradient is non-Lipschtiz continuous in terms of the second coordinate $g_2$ if $\lambda$ is unbounded, which is essential for all existing stochastic compositional optimization algorithms. Third, to the best of our knowledge, no optimal complexity in the order of $\mathcal{O}(1/\epsilon^2)$ has been achieved for a convex compositional function except for Zhang & Lan (2021), which assumes $f$ is convex and component-wisely non-decreasing and hence is not applicable to (2).

---

[1] $\widetilde{\mathcal{O}}$ omits a logarithmic dependence over $\epsilon$.

Table 1: Summary of algorithms solving KL-constrained DRO problem. Complexity represents the oracle complexity for achieving $\mathbb{E}[\text{dist}(0, \hat{\partial}\bar{F}(\mathbf{x}))] \leq \epsilon$ or other first-order stationarity convergence for the non-convex setting and $\mathbb{E}[F(\mathbf{x}) - F(\mathbf{x}_*)] \leq \epsilon$ for the convex setting. Per Iter Cost denotes the per-iteration computational complexity. The algorithm styles include primal-dual (PD), primal only (P), and compositional (COM). "-" means not available in the original paper.

| Setting | Algorithms | Reference | Complexity | Batch Size | Per Iter Cost | Style |
|---|---|---|---|---|---|---|
| Non-convex | PG-SMD2[2] | (Rafique et al., 2021) | $\mathcal{O}(1/\epsilon^4)$ | $\mathcal{O}(1)$ | $\mathcal{O}(n+d)$ | PD |
| | AccMDA | (Huang et al., 2020) | $\mathcal{O}(1/\epsilon^3)$ | $\mathcal{O}(1)$ | $\mathcal{O}(n+d)$ | PD |
| | Dual SGM | (Levy et al., 2020) | - | $\mathcal{O}(1)$ | $\mathcal{O}(d)$ | P |
| | SCDRO | This work | $\mathcal{O}(1/\epsilon^4)$ | $\mathcal{O}(1)$ | $\mathcal{O}(d)$ | COM |
| | ASCDRO | | $\widetilde{\mathcal{O}}(1/\epsilon^3)$ | $\mathcal{O}(1)$ | $\mathcal{O}(d)$ | COM |
| Convex | FastDRO[3] | (Levy et al., 2020) | $\mathcal{O}(1/\epsilon^3)$ | $\mathcal{O}(1/\epsilon)$ | $\mathcal{O}(\frac{d}{\epsilon})$ | P |
| | SPD | (Namkoong & Duchi, 2016) | $\mathcal{O}(1/\epsilon^2)$ | $\mathcal{O}(1)$ | $\mathcal{O}(n+d)$ | PD |
| | Dual SGM | (Levy et al., 2020) | $\mathcal{O}(1/\epsilon^2)$ | $\mathcal{O}(1)$ | $\mathcal{O}(d)$ | P |
| | RSCDRO | This work | $\mathcal{O}(1/\epsilon^3)$ | $\mathcal{O}(1)$ | $\mathcal{O}(d)$ | COM |
| | RASCDRO | | $\mathcal{O}(1/\epsilon^2)$ | $\mathcal{O}(1)$ | $\mathcal{O}(d)$ | COM |

To address the first two challenges, we derive an upper bound for the optimal $\lambda$ assuming that $\ell_i(\mathbf{w})$ is bounded for $\mathbf{w} \in \mathcal{W}$, i.e., $\lambda \in [\lambda_0, \tilde{\lambda}]$, which allows us to establish the smoothness condition of $F(\mathbf{x})$ and $f(g)$. Then we consider optimizing $\bar{F}(\mathbf{x}) = F(\mathbf{x}) + \delta_{\mathcal{X}}(\mathbf{x})$, where $\delta_{\mathcal{X}}(\mathbf{x}) = 0$ if $\mathbf{x} \in \mathcal{X} = \{\mathbf{x} = (\mathbf{w}^\top, \lambda)^\top : \mathbf{w} \in \mathcal{W}, \lambda \in [\lambda_0, \tilde{\lambda}]\}$. By leveraging the smoothness conditions of $F$ and $f$, we design stochastic algorithms by utilizing a recursive variance-reduction technique to compute a stochastic estimator of the gradient of $F(\mathbf{x})$, which allows us to achieve a complexity of $\widetilde{\mathcal{O}}(1/\epsilon^3)$ for finding a solution $\bar{\mathbf{x}}$ such that $\mathbb{E}[\text{dist}(0, \hat{\partial}\bar{F}(\bar{\mathbf{x}}))] \leq \epsilon$. To address the third challenge, we consider optimizing $\bar{F}_\mu(\mathbf{x}) = \bar{F}(\mathbf{x}) + \mu\|\mathbf{x}\|^2/2$ for a small $\mu$. We prove that $\bar{F}_\mu(\mathbf{x})$ satisfies a Kurdyka-Łojasiewicz inequality, which allows us to boost the convergence of the aforementioned algorithm to enjoy an optimal complexity of $\mathcal{O}(1/\epsilon^2)$ for finding an $\epsilon$-optimal solution to $\bar{F}(\mathbf{x})$. Besides the optimal algorithms, we also present simpler algorithms with worse complexity, which are more practical for deep learning applications without requiring two backpropagations at two different points per iteration as in the optimal algorithms.

In the existing analysis of compositional optimization algorithms, either (i) the problem is assumed to be unconstrained, e.g., Qi et al. (2020a; 2021), or (ii) the complexity is sub-optimal, e.g., Ghadimi et al. (2020), or (iii) the problem is restricted, e.g., the outer function $f$ is convex and non-decreasing as assumed in (Zhang & Lan, 2021). To the best of our knowledge, this is **the first result** for stochastic compositional optimization with a domain constraint that enjoys the optimal complexities for both convex and non-convex objectives.

## 2 Related Work

DRO springs from the robust optimization literature (Bertsimas et al., 2018; Ben-Tal et al., 2013) and has been extensively studied in machine learning and statistics (Ahmadi-Javid, 2012; Namkoong & Duchi, 2017; Duchi et al., 2016; Staib & Jegelka, 2019; Deng et al., 2020; Qi et al., 2020b; Duchi & Namkoong, 2021), and operations research (Rahimian & Mehrotra, 2019; Delage & Ye, 2010). Depending on how to constrain or regularize the uncertain variables, there are constrained DRO formulations that specify a constraint set for the uncertain variables, and regularized DRO formulations that use a regularization term in the objective for regularizing the uncertain variables (Levy et al., 2020). Duchi et al. (2016) showed that minimizing constrained DRO with $f$-divergence including a $\chi^2$-divergence constraint and a KL-divergence constraint, is equivalent to adding variance regularization for the Empirical Risk Minimization (ERM) objective, which is able to reduce the uncertainty and improve the generalization performance of the model.

**Primal-Dual Algorithms.** Many primal-dual algorithms designed for the min-max problems (Nemirovski et al., 2009; Juditsky et al., 2011; Yan et al., 2019; Namkoong & Duchi, 2016; Yan et al., 2020; Song et al., 2021; Alacaoglu et al., 2022) are applicable to solving (1) when $\ell$ is a convex function. For non-convex loss

---

[2]PG-SMD2 refers to PG-SMD algorithm under Assumption D2 in Rafique et al. (2021).

[3]FastDRO is name of the GitHub repository of Levy et al. (2020), and we use the name "FastDRO" to refer to the algorithm based on mini-batch gradient estimator in Levy et al. (2020).

functions, recently, Rafique et al. (2021) and Yan et al. (2020) proposed non-convex stochastic algorithms for solving non-convex strongly convex min-max problems, which are applicable to solving (1) when $\ell$ is a weakly convex function or smooth. Many primal-dual stochastic algorithms have been proposed for solving non-convex strongly concave problems with a state of the art oracle complexity of $\mathcal{O}(1/\epsilon^3)$ for finding a stationary solution (Huang et al., 2020; Luo et al., 2020; Tran-Dinh et al., 2020). However, the primal-dual algorithms require maintaining and updating an $\mathcal{O}(n)$ dimensional vector for updating the dual variable.

**Constrained DRO.** Wang et al. (2021) studies the Sinkhorn distance constraint DRO, a variant of Wasserstein distance based on entropic regularization. An efficient batch gradient descent with a bisection search algorithm has been proposed to obtain a near-optimal solution with an arbitrarily small sub-optimality gap. However, no non-asymptotic convergence results are established in their paper. Duchi & Namkoong (2021) developed a convex DRO framework with $f$-divergence constraints to improve model robustness. The author developed the finite-sample minimax upper and lower bounds and the non-asymptotic convergence rate of $\mathcal{O}(1/\sqrt{n})$, and provided the empirical studies on real distributional shifts tasks with existing interior point solver (Udell et al., 2014) and gradient descent with backtracking Armijo line-searches (Boyd et al., 2004). However, no stochastic algorithms that directly optimize the considered constrained DRO with non-asymptotic convergence rates are provided in their paper.

Recently, Levy et al. (2020) proposed sample independent algorithms based on gradient estimators for solving a group of DRO problems in the convex setting. To be more specific, they achieved a convergence rate of $\widetilde{\mathcal{O}}(1/\epsilon^2)$ for the $\chi^2$-constrained/regularized and CVaR-constrained convex DRO problems and the batch size of logarithmically dependent on the inverse accuracy level $\mathcal{O}(\log(1/\epsilon))$ with the help of multi-level Monte-Carlo (MLMC) gradient estimator. For the KL-constrained DRO objective and other more general setting, they achieve a convergence rate of $\mathcal{O}(1/\epsilon^3)$ under a Lipschitz continuity assumption on the inverse CDF of the loss function and a mini-batch gradient estimator with a batch size in the order $\mathcal{O}(1/\epsilon)$ (please refer to Table 3 in Levy et al. (2020)). In addition, Levy et al. (2020) also proposed a simple stochastic gradient method for solving the dual expression of the DRO formulation, which is called Dual SGM. In terms of convergence, they only discussed the convergence guarantee for the $\chi^2$-regularized and CVaR penalized convex DRO problems (cf. Claim 3 in their paper). However, there is still gap for proving the convergence rate of Dual SGM for non-convex KL-constrained DRO problems due to similar challenges mentioned in the previous section, in particular establishing the smoothness condition in terms of the primal variable and the Lagrangian multipliers (denoted as $\mathbf{x}, \nu, \eta$ respectively in their paper). This paper makes unique contributions for addressing these challenges by (i) removing $\eta$ in Dual SGM and deriving the box constraint for our Lagrangian multiplier $\lambda$ for proving the smoothness condition; (ii) establishing an optimal complexity in the order of $\mathcal{O}(1/\epsilon^3)$ in the presence of non-smooth box constraints, which, to the best of our knowledge, is the first time for solving a non-convex constrained compositional optimization problem.

Furthermore, it is noteworthy that the KL-constrained DRO formulation (2) offers a distinct advantage over the KL-regularized DRO problem without constraints. Specifically, the proposed algorithms enable automatic determination of an optimal regularization effect for the constrained DRO (2) upon the optimizing of $\lambda$, through the fine-tuning of the constraint upper bound $\rho$. This innovative approach has been empirically demonstrated to yield significant efficacy in the realm of contrastive learning, as substantiated by the findings of Qiu et al Qiu et al. (2023b).

**Regularized DRO.** DRO with KL divergence regularization objective has shown superior performance for addressing data imbalanced problems (Qi et al., 2021; 2020a; Li et al., 2020; 2021b). Jin et al. (2021) proposed a mini-batch normalized gradient descent with momentum that can find a first-order $\epsilon$ stationary point with an oracle complexity of $\mathcal{O}(1/\epsilon^4)$ for KL-regularized DRO and $\chi^2$ regularized DRO with a non-convex loss. They solve the challenge that the loss function could be unbounded. Qi et al. (2021) proposed online stochastic compositional algorithms to solve KL-regularized DRO. They leveraged a recursive variance reduction technique (STORM (Cutkosky & Orabona, 2019)) to compute a gradient estimator for the model parameter $\mathbf{w}$ only. They derived a complexity of $\widetilde{\mathcal{O}}(1/\epsilon^3)$ for a general non-convex problem and improved it to $\mathcal{O}(1/(\mu\epsilon))$ for a problem that satisfies an $\mu$-PL condition. Qi et al. (2020a) reports a worse complexity for a simpler algorithm for solving KL-regularized DRO. Li et al. (2020; 2021b) studied the effectiveness of KL regularized objective on different applications, such as enforcing fairness between subgroups, and handling the class imbalance.

**Compositional Functions and DRO.** The connection between compositional functions and DRO formulations have been observed and leveraged in the literature. Dentcheva et al. (2017) studied the statistical estimation of compositional functionals with applications to estimating conditional-value-at-risk measures, which is closely related to the CVaR constrained DRO. However, they do not consider stochastic optimization algorithms. To the best of our knowledge, Qi et al. (2021) was the first to use stochastic compositional optimization algorithms to solve KL-regularized DRO problems. Our work is different in that we solve KL-constrained DRO problems, which is more challenging than KL-regularized DRO problems. The benefits of using compositional optimization for solving DRO include (i) we do not need to maintain and update a high dimensional dual variable as in the primal-dual methods (Rafique et al., 2021); (i) we do not need to worry about the batch size as in MLMC-based stochastic methods (Levy et al., 2020; Hu et al., 2021).

**Optimizing the Temperature Parameter.** Our formulation and algorithm can be applied to optimizing the temperature parameter in the temperature-scaled cross-entropy loss, which has wide applications in machine learning and artificial intelligence, e.g., knowledge distillation Hinton et al. (2015) and self-supervised learning (Chen et al., 2020). Recently, Qiu et al. (2023a) leveraged the optimization technique proposed in this paper for optimizing the individualized temperature parameter in the global contrastive loss of self-superivsed learning.

## 3 Preliminaries

**Notations:** Let $\|\cdot\|$ denotes the Euclidean norm of a vector or the spectral norm of a matrix. And $\mathbf{x} = (\mathbf{w}^\top, \lambda)^\top \in \mathbb{R}^{d+1}$, $g_i(\mathbf{x}) = \exp(\frac{\ell_i(\mathbf{w})}{\lambda})$ and $g(\mathbf{x}) = \mathbb{E}_{i \sim \mathcal{D}}[\exp(\frac{\ell_i(\mathbf{w})}{\lambda})]$ where $\mathcal{D}$ denotes the training set and $i$ denotes the index of the sample randomly generated from $\mathcal{D}$. Let $f_\lambda(\cdot) = \lambda \log(\cdot) + \lambda\rho$, and $\nabla f_\lambda(g) = \frac{\lambda}{g}$ denotes the gradient of $f$ in terms of $g$. Let $\Pi_{\mathcal{X}}(\cdot)$ denote an Euclidean projection onto the domain $\mathcal{X}$. Let $[T] = \{1, \ldots, T\}$ and $\tau \sim [T]$ denotes a random selected index. We make the following standard assumptions regarding to the problem (2).

**Assumption 1.** *There exists $R$, $G$, $C$, and $L$ such that*

- *(a) The domain of model parameter $\mathcal{W}$ is bounded such that there exists $R > 0$ it holds $\|\mathbf{w}\| \leq R$ for any $\mathbf{w} \in \mathcal{W}$*

- *(b) $\ell_i(\mathbf{w})$ is $L$-smooth, i.e., $\|\nabla \ell_i(\mathbf{w}_1) - \nabla \ell_i(\mathbf{w}_2)\| \leq L\|\mathbf{w}_1 - \mathbf{w}_2\|$, $\forall \mathbf{w}_1, \mathbf{w}_2 \in \mathcal{W}, i \in \mathcal{D}$.*

- *(c) $\ell_i(\mathbf{w})$ is $G$-Lipschitz continuous function and bounded by $C$, i.e., $\|\nabla \ell_i(\mathbf{w})\| \leq G$ and $|\ell_i(\mathbf{w})| \leq C$ for all $\mathbf{w} \in \mathcal{W}$ and $i \in \mathcal{D}$.*

- *(d) There exists a positive constant $\Delta < \infty$ and an initial solution $(\mathbf{w}_1, \lambda_1)$ such that $F(\mathbf{w}_1, \lambda_1) - \min_{\mathbf{w} \in \mathcal{W}} \min_{\lambda \geq \lambda_0} F(\mathbf{w}, \lambda) \leq \Delta$.*

**Assumption 2.** *Let $\sigma_g$, $\sigma_{\nabla g}$ be positive constants and $\sigma^2 = \max\{\sigma_g, \sigma_{\nabla g}\}$. For $i \in \mathcal{D}$, assume that $\mathbb{E}[\|g_i(\mathbf{x}) - g(\mathbf{x})\|^2] \leq \sigma_g^2$, $\mathbb{E}[\|\nabla g_i(\mathbf{x}) - \nabla g(\mathbf{x})\|^2] \leq \sigma_{\nabla g}^2$.*

**Remark:** Assumption 1 (a), i.e., the boundness condition of $\mathcal{W}$ is also assumed in Levy et al. (2020), which is mainly used for convex analysis. Assumption 1(b), (c), i.e., the Lipstchiz continuity and smoothness of loss function, and the variance bounds for $g_i$ and its gradient in Assumption 2 can be derived from Assumption 1 (b), such that $\mathbb{E}[\|g_i(\mathbf{x}) - g(\mathbf{x})\|^2] \leq \mathbb{E}[\|g_i(\mathbf{x})\|^2] \leq \exp(\frac{2C}{\lambda_0})$, and $\mathbb{E}[\|\nabla g_i(\mathbf{x}) - \nabla g(\mathbf{x})\|^2] \leq \mathbb{E}[\|\nabla g_i(\mathbf{x})\|^2] \leq \exp(\frac{2C}{\lambda_0})(G^2 + \frac{C^2}{\lambda_0})^4$

However, $F(\mathbf{w}, \lambda)$ is not necessarily smooth in terms of $\mathbf{x} = (\mathbf{w}^\top, \lambda)^\top$ if $\lambda$ is unbounded. To address this concern, we prove that optimal $\lambda$ is indeed bounded.

**Lemma 1.** *The optimal solution of the dual variable $\lambda^*$ to the problem (2) is upper bounded by $\tilde{\lambda} = \lambda_0 + C/\rho$, where $C$ is the upper bound of the loss function and $\rho$ is the constraint parameter.*

---

[4] We would like to point out that the variance bound and the smoothness constant $L_F$ are exponentially dependent on the problem parameters, so are these constants in some other stochastic methods solving constrained DRO, like Dual SGM in Levy et al. (2020).

Thus, we could constrain the domain of $\lambda$ in the DRO formulation (2) with the upper bound $\tilde{\lambda}$, and obtain the following equivalent formulation:

$$\min_{\mathbf{w}\in\mathcal{W}} \min_{\lambda_0 \leq \lambda \leq \tilde{\lambda}} \lambda \log\left(\frac{1}{n}\sum_{i=1}^{n}\exp\left(\frac{\ell_i(\mathbf{w})}{\lambda}\right)\right) + \lambda\rho. \tag{3}$$

The upper bound $\tilde{\lambda}$ guarantees the smoothness of $F(\mathbf{w}, \lambda)$ and the smoothness of $f_\lambda(\cdot)$, which are critical for the proposed algorithms to enjoy fast convergence rates.

**Lemma 2.** *$F(\mathbf{w}, \lambda)$ is $L_F$-smooth for any $\mathbf{w} \in \mathcal{W}$ and $\lambda \in [\lambda_0, \tilde{\lambda}]$, where $L_F = \tilde{\lambda}L_g^2 + 2L_g + \tilde{\lambda}L_{\nabla_g} + 1 + \tilde{\lambda}$. $L_g$ and $L_{\nabla_g}$ are constants independent of sample size $n$ and explicitly derived in Lemma 7 .*

Below, we let $\mathcal{X} = \{\mathbf{x}|\mathbf{w} \in \mathcal{W}, \lambda_0 \leq \lambda \leq \tilde{\lambda}\}$, $\delta_\mathcal{X}(\mathbf{x}) = 0$ if $\mathbf{x} \in \mathcal{X}$, and $\delta_\mathcal{X}(\mathbf{x}) = \infty$ if $\mathbf{x} \notin \mathcal{X}$. The problem (3) is equivalent to :

$$\min_{\mathbf{x}\in\mathbb{R}^{d+1}} \bar{F}(\mathbf{x}) := F(\mathbf{x}) + \delta_\mathcal{X}(\mathbf{x}), \tag{4}$$

Since $\bar{F}$ is non-smooth, we define the regular subgradient as follows.

**Definition 1** (Regular Subgradient). *Consider a function $\Phi : \mathbb{R}^n \to \overline{\mathbb{R}}$ and $\Phi(\bar{\mathbf{x}})$ is finite at a point $\bar{\mathbf{x}}$. For a vector $\mathbf{v} \in \mathbb{R}^n$, $\mathbf{v}$ is a regular subgradient of $\Phi$ at $\bar{\mathbf{x}}$, written $\mathbf{v} \in \hat{\partial}\Phi(\bar{\mathbf{x}})$, if*

$$\liminf_{\mathbf{x}\to\bar{\mathbf{x}}} \frac{\Phi(\mathbf{x}) - \Phi(\bar{\mathbf{x}}) - \mathbf{v}^\top(\mathbf{x}-\bar{\mathbf{x}})}{\|\mathbf{x}-\bar{\mathbf{x}}\|} \geq 0.$$

Since $F(\mathbf{x})$ is differentiable, we use $\hat{\partial}\bar{F}(\mathbf{x}) = \nabla F(\mathbf{x}) + \hat{\partial}\delta_\mathcal{X}(\mathbf{x})$ (see Exercise 8.8 in Rockafellar & Wets (1998)) in the analysis. Recall the definition of subgradient of a convex function $\bar{F}$ which is denoted by $\partial\bar{F}$. When $\bar{F}(\mathbf{x})$ is convex, we have $\hat{\partial}\bar{F}(\mathbf{x}) = \partial\bar{F}(\mathbf{x})$ (see Proposition 8.2 in Rockafellar & Wets (1998)). The $\text{dist}(0, \hat{\partial}\bar{F}(\mathbf{x}))$ measures the distance between the origin and the regular subgradient set of $\bar{F}$ at $\mathbf{x}$. The oracle complexity is defined below:

**Definition 2** (Oracle Complexity). *Let $\epsilon > 0$ be a small constant, the oracle complexity is defined as the number of processing samples $\mathbf{z}$ in order to achieve $\mathbb{E}[\text{dist}(0, \hat{\partial}\bar{F}(\mathbf{x}))] \leq \epsilon$ for a non-convex loss function or $\mathbb{E}[F(\mathbf{x}) - F(\mathbf{x}_*)] \leq \epsilon$ for a convex loss function.*

### 3.1 Equivalence Derivation

Before we move to the proposed algorithms in the next section, we derive the equivalence between equation between equations (1), (2), and (3). Recall the original KL-constrained DRO problem:

$$\min_{\mathbf{w}\in\mathcal{W}} \max_{\{\mathbf{p}\in\Delta_n : D(\mathbf{p},\mathbf{1}/n)\leq\rho\}} \sum_{i=1}^{n} p_i\ell_i(\mathbf{w}) - \lambda_0 D(\mathbf{p}, 1/n),$$

where $\Delta_n = \{\mathbf{p} \in \mathbb{R}^n : \sum_{i=1}^n p_i = 1, 0 \leq p_i \leq 1\}$, $D(\mathbf{p}, 1/n)$ is the KL divergence and $\lambda_0$ is a small positive constant.

In order to tackle this problem, let us first consider the robust loss

$$\max_{\{\mathbf{p}\in\Delta_n : D(\mathbf{p},\mathbf{1}/n)\leq\rho\}} \sum_{i=1}^{n} p_i\ell_i(\mathbf{w}) - \lambda_0 D(\mathbf{p}, 1/n).$$

And then we invoke the dual variable $\lambda$ to transform this primal problem to the following form

$$\max_{\mathbf{p}\in\Delta_n} \min_{\bar{\lambda}\geq 0} \sum_{i=1}^{n} p_i\ell_i(\mathbf{w}) - \bar{\lambda}(D(\mathbf{p}, 1/n) - \rho) - \lambda_0 D(\mathbf{p}, 1/n).$$

Since this problem is concave in term of $\mathbf{p}$ given $\mathbf{w}$, by strong duality theorem, we have

$$\max_{\mathbf{p}\in\Delta_n} \min_{\bar{\lambda}\geq 0} \sum_{i=1}^{n} p_i\ell_i(\mathbf{w}) - \bar{\lambda}(D(\mathbf{p},1/n)-\rho) - \lambda_0 D(\mathbf{p},1/n)$$

$$= \min_{\bar{\lambda}\geq 0} \max_{\mathbf{p}\in\Delta_n} \sum_{i=1}^{n} p_i\ell_i(\mathbf{w}) - \bar{\lambda}(D(\mathbf{p},1/n)-\rho) - \lambda_0 D(\mathbf{p},1/n).$$

Let $\lambda = \bar{\lambda} + \lambda_0$, we have

$$\min_{\bar{\lambda}\geq 0} \max_{\mathbf{p}\in\Delta_n} \sum_{i=1}^{n} p_i\ell_i(\mathbf{w}) - \bar{\lambda}(D(\mathbf{p},1/n)-\rho) - \lambda_0 D(\mathbf{p},1/n)$$

$$= \min_{\lambda\geq\lambda_0} \max_{\mathbf{p}\in\Delta_n} \sum_{i=1}^{n} p_i\ell_i(\mathbf{w}) - \lambda(D(\mathbf{p},1/n)-\rho) - \lambda_0\rho.$$

Then the original problem is equivalent to the following problem

$$\min_{\mathbf{w}\in\mathcal{W}} \min_{\lambda\geq\lambda_0} \max_{\mathbf{p}\in\Delta_n} \sum_{i=1}^{n} p_i\ell_i(\mathbf{w}) - \lambda(D(\mathbf{p},1/n)-\rho) - \lambda_0\rho,$$

Next we fix $\mathbf{x} = (\mathbf{w}^\top, \lambda)^\top$ and derive an optimal solution $\mathbf{p}^*(\mathbf{x})$ which depends on $\mathbf{x}$ and solves the inner maximization problem. We consider the following problem

$$\min_{\mathbf{p}\in\Delta_n} -\sum_{i=1}^{n} p_i\ell_i(\mathbf{w}) + \lambda D(\mathbf{p},1/n).$$

which has the same optimal solution $\mathbf{p}^*(\mathbf{x})$ with our problem.

There are three constraints to handle, i.e., $p_i \geq 0, \forall i$ and $p_i \leq 1, \forall i$ and $\sum_{i=1}^{n} p_i = 1$. Note that the constraint $p_i \geq 0$ is enforced by the term $p_i\log(p_i)$, otherwise the above objective will become infinity. As a result, the constraint $p_i < 1$ is automatically satisfied due to $\sum_{i=1}^{n} p_i = 1$ and $p_i \geq 0$. Hence, we only need to explicitly tackle the constraint $\sum_{i=1}^{n} p_i = 1$. To this end, we define the following Lagrangian function

$$L_{\mathbf{x}}(\mathbf{p},\mu) = -\sum_{i=1}^{n} p_i\ell_i(\mathbf{w}) + \lambda\left(\log n + \sum_{i=1}^{n} p_i\log(p_i)\right) + \mu(\sum_{i=1}^{n} p_i - 1),$$

where $\mu$ is the Lagrangian multiplier for the constraint $\sum_{i=1}^{n} p_i = 1$. The optimal solutions satisfy the KKT conditions:

$$-\ell_i(\mathbf{w}) + \lambda\left(\log(p_i^*(\mathbf{x})) + 1\right) + \mu = 0 \text{ and } \sum_{i=1}^{n} p_i^*(\mathbf{x}) = 1.$$

From the first equation, we can derive $p_i^*(\mathbf{x}) \propto \exp(\ell_i(\mathbf{w})/\lambda)$. Due to the second equation, we can conclude that $p_i^*(\mathbf{x}) = \frac{\exp(\ell_i(\mathbf{w})/\lambda)}{\sum_{i=1}^{n}\exp(\ell_i(\mathbf{w})/\lambda)}$. Plugging this optimal $\mathbf{p}^*(\mathbf{w})$ into the inner maximization problem, we have

$$\sum_{i=1}^{n} p_i^*(\mathbf{x})\ell_i(\mathbf{w}) - \lambda\left(\log n + \sum_{i=1}^{n} p_i^*(\mathbf{w})\log(p_i^*(\mathbf{w}))\right) = \lambda\log\left(\frac{1}{n}\sum_{i=1}^{n}\exp\left(\frac{\ell_i(\mathbf{w})}{\lambda}\right)\right),$$

Therefore, we get the following equivalent problem to the original problem

$$\min_{\mathbf{w}\in\mathcal{W}} \min_{\lambda\geq\lambda_0} \lambda\log\left(\frac{1}{n}\sum_{i=1}^{n}\exp\left(\frac{\ell_i(\mathbf{w})}{\lambda}\right)\right) + \lambda\rho.$$

which is Eq. (2) in the paper.

## 4 Stochastic Constrained DRO with Non-convex Losses

In this section, we present two stochastic algorithms for solving (4). The first algorithm is simpler yet practical for deep learning applications. The second algorithm is an accelerated one with a better complexity, which is more complex than the first algorithm.

### 4.1 Basic Algorithm: SCDRO

A major concern of the algorithm design is to compute a stochastic gradient estimator of the gradient of $F(\mathbf{x})$. At iteration $t$, the gradient of $F(\mathbf{x}_t)$ is given by

$$\nabla_{\mathbf{w}} F(\mathbf{x}_t) = \nabla f_{\lambda_t}(g(\mathbf{x}_t)) \nabla_{\mathbf{w}} g(\mathbf{x}_t)$$
$$\nabla_\lambda F(\mathbf{x}_t) = \nabla f_{\lambda_t}(g(\mathbf{x}_t)) \nabla_\lambda g(\mathbf{x}_t) + \log(g(\mathbf{x}_t)) + \rho. \tag{5}$$

Both $\nabla_\lambda g(\mathbf{x}_t)$ and $\nabla_{\mathbf{w}} g(\mathbf{x}_t)$ can be estimated by unbiased estimator denoted by $\nabla g_i(\mathbf{x}_t)$. The concern lies at how to estimate $g(\mathbf{x}_t)$ inside $\nabla f_{\lambda_t}(\cdot)$. The first algorithm SCDRO is applying existing techniques for two-level compositional function. In particular, we estimate $g(\mathbf{x}_t)$ by a sequence of $s_t$, which is updated by moving average $s_t = (1-\beta)s_{t-1} + \beta g_i(\mathbf{x}_t)$. Then we substitute $g(\mathbf{x}_t)$ in $\nabla_{\mathbf{w}} F(\mathbf{x}_t)$ and $\nabla_\lambda F(\mathbf{x}_t)$ with $s_t$, and invoke the following moving average to obtain the gradient estimators in terms of $\mathbf{w}_t$ and $\lambda_t$, respectively,

$$\mathbf{v}_t = (1-\beta)\mathbf{v}_{t-1} + \beta \nabla f_{\lambda_t}(s_t) \nabla_{\mathbf{w}} g_i(\mathbf{x}_t) \tag{6}$$
$$u_t = (1-\beta)u_{t-1} + \beta(\nabla f_{\lambda_t}(s_t)\nabla_\lambda g_i(\mathbf{x}_t) + \log(s_t) + \rho).$$

Finally we complete the update step of $\mathbf{x}_t$ by $\mathbf{x}_{t+1} = \Pi_{\mathcal{X}}(\mathbf{x}_t - \eta \mathbf{z}_t)$, where $\mathbf{z}_t = (\mathbf{v}_t^\top, u_t)^\top$.

---

**Algorithm 1** SCDRO$(\mathbf{x}_1, \mathbf{v}_1, u_1, s_1, \eta_1, T_1)$

1: **Input:** $\mathbf{w}_1 \in \mathcal{W}, \lambda_1 \geq \lambda_0, \mathbf{x}_1 = (\mathbf{w}_1^\top, \lambda_1)^\top$
2: **Initialization:** Draw a sample $\xi_1 \sim \mathcal{D}$, and calculate $s_1 = \exp(\ell_i(\mathbf{w}_1)/\lambda_1)$,

$$\mathbf{v}_1 = \nabla f_{\lambda_1}(s_1)\nabla_{\mathbf{w}} g_i(\mathbf{x}_1)) \in \mathbb{R}^d$$

$$u_1 = \nabla f_{\lambda_1}(s_1)\nabla_\lambda g_i(\mathbf{x}_1) + \log(s_1) + \rho \in \mathbb{R}$$

3: **for** $t = 1, \cdots, T$ **do**
4:    Update $\mathbf{x}_{t+1} = \Pi_{\mathcal{X}}(\mathbf{x}_t - \eta\mathbf{z}_t)$
5:    Draw a sample $\xi_i \sim \mathcal{D}$
6:    Let $s_{t+1} = (1-\beta)s_t + \beta g_i(\mathbf{x}_{t+1})$
7:    Update $\mathbf{v}_{t+1}, u_{t+1}$ according to (1):
    $\mathbf{v}_t = (1-\beta)\mathbf{v}_{t-1} + \beta\nabla f_{\lambda_t}(s_t)\nabla_{\mathbf{w}} g_i(\mathbf{x}_t)$
    $u_t = (1-\beta)u_{t-1}$
    $\quad + \beta(\nabla f_{\lambda_t}(s_t)\nabla_\lambda g_i(\mathbf{x}_t) + \log(s_t) + \rho).$
8: **end for**
9: **return:** $(\mathbf{x}_\tau, \mathbf{v}_\tau, u_\tau, s_\tau)$, where $\tau \sim [T]$

**Algorithm 2** ASCDRO$(\mathbf{x}_1, \mathbf{v}_1, u_1, s_1, \eta_1, T_1)$

1: **Input:** $\mathbf{w}_1 \in \mathcal{W}, \lambda_1 \geq \lambda_0, \mathbf{x}_1 = (\mathbf{w}_1^\top, \lambda_1)^\top$
2: **Initialization:** Draw a sample $\xi_1 \sim \mathcal{D}$, and calculate $s_1 = \exp(\ell_i(\mathbf{w}_1)/\lambda_1)$,

$$\mathbf{v}_1 = \nabla_{\mathbf{w}} g_i(\mathbf{x}_1) \in \mathbb{R}^d, \ u_1 = \nabla_\lambda g_i(\mathbf{x}_1) \in \mathbb{R}$$

3: **for** $t = 1, \cdots, T$ **do**
4:    Update $\mathbf{x}_{t+1} = \Pi_{\mathcal{X}}(\mathbf{x}_t - \eta\mathbf{z}_t)$, where $\mathbf{z}_t$ is given in (8):
    $\mathbf{z}_t = (\nabla f_{\lambda_t}(s_t)\mathbf{v}_t^\top, \nabla f_{\lambda_t}(s_t)u_t + \log(s_t) + \rho)^\top$
5:    Draw a sample $\xi_i \sim \mathcal{D}$
6:    Update $s_{t+1}, \mathbf{v}_{t+1}, u_{t+1}$:
    $\mathbf{v}_t = \nabla_{\mathbf{w}} g_i(\mathbf{x}_t) + (1-\beta)(\mathbf{v}_{t-1} - \nabla_{\mathbf{w}} g_i(\mathbf{x}_{t-1}))$
    $u_t = \nabla_\lambda g_i(\mathbf{x}_t) + (1-\beta)(u_{t-1} - \nabla_\lambda g_i(\mathbf{x}_{t-1}))$
    $s_t = g_i(\mathbf{x}_t) + (1-\beta)(s_{t-1} - g_i(\mathbf{x}_{t-1})).$
7: **end for**
8: **return:** $(\mathbf{x}_\tau, \mathbf{v}_\tau, u_\tau, s_\tau)$, where $\tau \sim [T]$

---

We would like to point out the moving average estimator for tracking the inner function $g(\mathbf{w})$ is widely used for solving compositional optimization problems (Wang et al., 2017; Qi et al., 2021; Zhang & Xiao, 2019; Zhou et al., 2019). Using the moving average for computing a stochastic gradient estimator of a compositional function was first used in the NASA method proposed in Ghadimi et al. (2020). The proposed method SCDRO is presented in Algorithm 1. It is similar to NASA but with a simpler design on the update of $\mathbf{x}_{t+1}$. We directly use projection after an SGD-tyle update. In contrast, NASA uses two steps to update $\mathbf{x}_{t+1}$. As a consequence, NASA has two parameters for updating $\mathbf{x}_{t+1}$ while SCDRO only has one parameter $\eta$ for updating $\mathbf{x}_{t+1}$. It is this simple change that allows us to extend SCDRO for convex problems in the next section. Below, we present the convergence rate of our basic algorithm SCDRO for a non-convex loss function.

**Theorem 1.** *Suppose the Assumption 1 and 2 hold, and set $\beta = \frac{1}{\sqrt{T}}, \eta = \frac{\beta}{20L_F^2}$. Then after running Algorithm 1 T iterations, we have $\mathbb{E}[\text{dist}(0, \hat{\partial}\bar{F}(\mathbf{x}_\tau))^2] \leq (624\sigma^2 + 280\Delta)\frac{L_F^2}{\sqrt{T}} + \frac{20L_F^2\Delta}{T}$.*

**Remark:** Theorem 1 shows that SCDRO achieves a complexity of $\mathcal{O}(1/\epsilon^4)$ for finding an $\epsilon$-stationary point, i.e., $\mathbb{E}[\text{dist}(0, \hat{\partial}\bar{F}(\mathbf{x}_R))] \leq \epsilon$ for a non-convex loss function. Note that NASA (Ghadimi et al., 2020) enjoys the same oracle complexity but for a different convergence measure, i.e., $\mathbb{E}[\|\mathbf{y}(\mathbf{x}, \mathbf{z}) - \mathbf{x}\|^2 + \|\mathbf{z} - \nabla F(\mathbf{x})\|^2] \leq \epsilon$ for a returned primal-dual pair $(\mathbf{x}, \mathbf{z})$, where $\mathbf{y}(\mathbf{x}, \mathbf{z}) = \prod_{\mathcal{X}}[\mathbf{x} - \mathbf{z}]$. We can see that our convergence measure is more intuitive. In addition, we are able to leverage our convergence measure to establish the

convergence for convex functions by using Kurdyka-Łojasiewicz (KL) inequality and the restarting trick as shown in next section. In contrast, such convergence for NASA is missing in their paper. Compared with stochastic primal-dual methods (Rafique et al., 2021; Yan et al., 2020) for the min-max formulation (1), their algorithms are double looped and have the same oracle complexity for a different convergence measure, i.e., $\mathbb{E}[\text{dist}(0, \hat{\partial}\bar{F}(\mathbf{x}_*))^2] \leq \gamma^2 \|\mathbf{x} - \mathbf{x}_*\|^2 \leq \epsilon$ for some returned solution $\mathbf{x}$, where $\mathbf{x}_*$ is a reference point that is not computable. Our convergence measure is stronger as we directly measure $\mathbb{E}[\text{dist}(0, \hat{\partial}\bar{F}(\mathbf{x}_\tau))^2]$ on a returned solution $\mathbf{x}_\tau$. This is due to that we leverage the smoothness of $F(\cdot)$.

### 4.2 Accelerated Algorithm: ASCDRO

Our second algorithm presented in Algorithm 2 is inspired by Qi et al. (2021) for solving the KL-regularized DRO by leveraging a recursive variance reduced technique (i.e., STORM) to estimate $g(\mathbf{w}_t)$ and $\nabla g(\mathbf{w}_t)$ for computing $\nabla_{\mathbf{w}} F(\mathbf{x}_t)$ and $\nabla_\lambda F(\mathbf{x}_t)$ in (5). In particular, we use $\mathbf{v}_t$ for tracking $\nabla_{\mathbf{w}} g(\mathbf{x}_t)$, use $u_t$ for tracking $\nabla_\lambda g(\mathbf{x}_t)$, and use $s_t$ for tracking $g(\mathbf{x}_t)$, which are updated by:

$$
\begin{aligned}
\mathbf{v}_t &= \nabla_{\mathbf{w}} g_i(\mathbf{x}_t) + (1 - \beta)(\mathbf{v}_{t-1} - \nabla_{\mathbf{w}} g_i(\mathbf{x}_{t-1})) \\
u_t &= \nabla_\lambda g_i(\mathbf{x}_t) + (1 - \beta)(u_{t-1} - \nabla_\lambda g_i(\mathbf{x}_{t-1})) \\
s_t &= g_i(\mathbf{x}_t) + (1 - \beta)(s_{t-1} - g_i(\mathbf{x}_{t-1})).
\end{aligned}
\tag{7}
$$

A similar update to $s_t$ has been used in Chen et al. (2021) for tracking the inner function values for two-level compositional optimization. However, they do not use similar updates for tracking the gradients as $\mathbf{v}_t, u_t$. Hence, their algorithm has a worse complexity.

Then we invoke these estimators into $\nabla_{\mathbf{w}} F(\mathbf{x}_t)$ and $\nabla_\lambda F(\mathbf{x}_t)$ to obtain the gradient estimator

$$
\mathbf{z}_t = (\nabla f_{\lambda_t}(s_t)\mathbf{v}_t^\top, \nabla f_{\lambda_t}(s_t)u_t + \log(s_t) + \rho)^\top.
\tag{8}
$$

Below, we show ASCDRO can achieve a better convergence rate in the non-convex loss function.

**Theorem 2.** *Under Assumption 1 and 2, for any $\alpha > 1$, let $k = \frac{\alpha\sigma^{2/3}}{L_F}$, $w = \max(2\sigma^2, (16L_F^2 k)^3)$ and $c = \frac{\sigma^2}{14L_F k^3} + 130L_F^4$. Then after running Algorithm 2 for $T$ iterations with $\eta_t = \frac{k}{(w + t\sigma^2)^{1/3}}$ and $\beta_t = c\eta_t^2$, we have $\mathbb{E}[\text{dist}(0, \hat{\partial}\bar{F}(\mathbf{x}_\tau))^2] \leq \mathcal{O}\left(\frac{\log T}{T^{2/3}}\right)$.*

**Remark:** Theorem 2 implies that with a polynomial decreasing step size, ASCDRO is able to find an $\epsilon$-stationary solution such that $\mathbb{E}[\text{dist}(0, \hat{\partial}\bar{F}(\mathbf{x}_R))] \leq \epsilon$ with a near-optimal complexity $\widetilde{\mathcal{O}}(1/\epsilon^3)$. Note that the complexity $\widetilde{\mathcal{O}}(1/\epsilon^3)$ is optimal up to a logarithmic factor for solving non-convex smooth optimization problems (Arjevani et al., 2019). State-of-the-art primal-dual methods with variance-reduction for min-max problems (Huang et al., 2020) have the same complexity but for a different convergence measure, i.e, $\mathbb{E}[\frac{1}{\gamma}\|\mathbf{x} - \prod_{\mathcal{X}}[\mathbf{x} - \gamma\nabla F(\mathbf{x})]\|] \leq \epsilon$ for a returned solution $\mathbf{x}$.

## 5 Stochastic Algorithms for Convex Problems

In this section, we presented restarted algorithms for solving (3) with a convex loss function $\ell_i(\mathbf{w})$. The key is to restart SCDRO and ASCDRO by using a stagewise step size scheme. We define a new objective $F_\mu(\mathbf{x}) = F(\mathbf{x}) + \mu\|\mathbf{x}\|^2/2$ and correspondingly $\bar{F}_\mu(\mathbf{x}) = F_\mu(\mathbf{x}) + \delta_{\mathcal{X}}(\mathbf{x})$, where $\mu$ is a constant to be determined later. With this new objective, we have the following lemma.

**Lemma 3.** *Suppose that $\ell_i(\mathbf{w})$ is convex for all $i$, then for all $\mathbf{x} \in \mathcal{X}$, $\bar{F}_\mu(\mathbf{x})$ satisfies the following Kurdyka-Łojasiewicz (KL) inequality $\text{dist}(0, \partial\bar{F}_\mu(\mathbf{x}))^2 \geq 2\mu(\bar{F}_\mu(\mathbf{x}) - \inf_{\mathbf{x}\in\mathcal{X}} \bar{F}_\mu(\mathbf{x}))$.*

Lemma 3 allows us to obtain the convergence guarantee for convex losses. The idea of the restarted algorithm is to apply SCDRO and ASCDRO to the new objective $\bar{F}_\mu(\mathbf{x})$ by adding $\mu\mathbf{x}_t$ to $(\nabla f_{\lambda_t}(s_t)\nabla_{\mathbf{w}} g_i(\mathbf{x}_t)^\top, \nabla f_{\lambda_t}(s_t)\nabla_\lambda g_i(\mathbf{x}_t) + \log(s_t) + \rho)^\top$ in Eq. (1) of Algorithm 1 and substituting $\mathbf{z}_t$ in (8) of Algorithm 2 by $\mathbf{z}_t = (\nabla f_{\lambda_t}(s_t)\mathbf{v}_t^\top, \nabla f_{\lambda_t}(s_t)u_t + \log(s_t) + \rho)^\top + \mu\mathbf{x}_t$, and restarting SCDRO or ASCDRO with a stagewise step size to enjoy the benefit of KL inequality of $\bar{F}_\mu(\mathbf{x})$. It is notable that a stagewise step size is widely and commonly used in practice. The multi-stage restarted version of SCDRO and ASCDRO are shown Algorithm 3, to which we refer as restarted-SCDRO (RSCDRO) and restarted-ASCDRO (RASCDRO).

---

**Algorithm 3** RSCDRO or RASCDRO

---

1: **Input:** $\mathbf{w}_1 \in \mathcal{W}, \lambda_1 \in \mathbb{R}^+, \mathbf{x}_1 = (\mathbf{w}_1^\top, \lambda_1)^\top$
2: **Initialization:** The same as in SCDRO or ASCDRO
3: Let $\Lambda_k = (\mathbf{x}_k, \mathbf{v}_k, u_k, s_k)$
4: **for** $k = 1, \cdots, K$ **do**
5:     $\Lambda_{k+1} = \text{SCDRO}(\Lambda_k, \eta_k, T_k)$ or $\Lambda_{k+1} = \text{ASCDRO}(\Lambda_k, \eta_k, T_k)$
6:     Change $\eta_k, T_k$ according to Lemma 4 or Lemma 5
7: **end for**
8: **return:** $\mathbf{x}_K$

---

### 5.1 Restarted SCDRO for Convex Problems

In this subsection, we present the convergence rate of RSCDRO for convex losses. We first present a lemma that states $F_\mu(\mathbf{x}_k)$ is stagewisely decreasing.

**Lemma 4.** *Suppose Assumptions 1 and 2 hold, $\ell_i(\mathbf{w})$ is convex for all $i$, and $F_\mu(\mathbf{x}_1) - \inf_{\mathbf{x} \in \mathcal{X}} F_\mu(\mathbf{x}) \leq \Delta_\mu < \infty$. Let $\epsilon_1 = \Delta_\mu$, $\epsilon_k = \epsilon_{k-1}/2$, $\beta_k = \min\{\frac{\mu \epsilon_k}{c\sigma^2}, \frac{1}{c}\}, \eta_k = \min\{\frac{\mu \epsilon_k}{12 c L_F^2 \sigma^2}, \frac{1}{12 c L_F^2}\}$ and $T_k = \max\{\frac{384 c L_F^2 \sigma^2}{\mu^2 \epsilon_k}, \frac{384 c L_F^2}{\mu}\}$, where $c = 384 L_F^2$. Run RSCDRO, then we have $\mathbb{E}[F_\mu(\mathbf{x}_k) - \inf_{\mathbf{x} \in \mathcal{X}} F_\mu(\mathbf{x})] \leq \epsilon_k$ for each stage $k$.*

The above lemma implies that the objective gap $\mathbb{E}[F_\mu(\mathbf{x}_k) - \inf_{\mathbf{x} \in \mathcal{X}} F_\mu(\mathbf{x})]$ is decreased by a factor of 2 after each stage. Based on the above lemma, RSCDRO has the following convergence rate

**Theorem 3.** *Under the same assumptions and parameter settings as Lemma 4, after $K = \mathcal{O}(\log_2(\epsilon_1/\epsilon))$ stages, the output of RSCDRO satisfies $\mathbb{E}[F_\mu(\mathbf{x}_K) - \inf_{\mathbf{x} \in \mathcal{X}} F_\mu(\mathbf{x})] \leq \epsilon$, and the oracle complexity is $\mathcal{O}(1/\mu^2\epsilon)$.*

As $F_\mu(\mathbf{x}_K) - F_\mu(\mathbf{x}_*) \leq F_\mu(\mathbf{x}_K) - \inf_{\mathbf{x} \in \mathcal{X}} F_\mu(\mathbf{x})$, where $\mathbf{x}_* = \arg\min_{\mathbf{x} \in \mathcal{X}} F(\mathbf{x})$. Therefore, if after $K$ stages it holds that $\mathbb{E}[F_\mu(\mathbf{x}_K) - \inf_{\mathbf{x} \in \mathcal{X}} F_\mu(\mathbf{x})] \leq \epsilon/2$ with an oracle complexity of $\mathcal{O}(1/\mu^2\epsilon)$, we have $\mathbb{E}[F_\mu(\mathbf{x}_K) - F_\mu(\mathbf{x}_*)] \leq \epsilon/2$ , i.e., $\mathbb{E}[F(\mathbf{x}_K) + \mu\|\mathbf{x}_K\|^2/2 - F(\mathbf{x}_*) - \mu\|\mathbf{x}_*\|^2/2] \leq \epsilon/2$. By Assumption 1(a) $\mathcal{W}$ is bounded by $R$, and then by setting $\mu = \epsilon/(2(R^2 + \tilde{\lambda}^2))$, with $\|\mathbf{x}\|^2 \leq (R^2 + \tilde{\lambda}^2)$ we have

$$\mathbb{E}[F(\mathbf{x}_K) - F(\mathbf{x}_*)] \leq \frac{\epsilon}{2} + (2(R^2 + \tilde{\lambda}^2))\frac{\mu}{2} \leq \frac{\epsilon}{2} + \frac{\epsilon}{2} \leq \epsilon$$

with an oracle complexity of $\mathcal{O}(1/\epsilon^3)$, i.e, the following corollary holds.

**Corollary 1.** *Let $\mu = \epsilon/(2(R^2 + \tilde{\lambda}^2))$. Then under the same assumptions and parameter settings as Lemma 4, after $K = \mathcal{O}(\log_2(\epsilon_1/\epsilon))$ stages, the output of RSCDRO satisfies $\mathbb{E}[F(\mathbf{x}_K) - \inf_{\mathbf{x} \in \mathcal{X}} F(\mathbf{x})] \leq \epsilon$ and the oracle complexity is $\mathcal{O}(1/\epsilon^3)$.*

**Remark:** Corollary 1 shows that RSCDRO achieves an oracle complexity of $\mathcal{O}(1/\epsilon^3)$ for finding an $\epsilon$-optimal solution. i.e., $\mathbb{E}[F(\mathbf{x}) - F(\mathbf{x}_*)] \leq \epsilon$ for the convex loss function with a geometrically decreasing step size in a stagewise manner.

### 5.2 Restarted ASCDRO for Convex Problems

In this subsection, we establish a better convergence rate of RASCDRO for convex losses.

**Lemma 5.** *Suppose Assumptions 1 and 2 hold, $\ell_i(\mathbf{w})$ is convex for all $i$, and $F_\mu(\mathbf{x}_1) - \inf_{\mathbf{x} \in \mathcal{X}} F_\mu(\mathbf{x}) \leq \Delta_\mu < \infty$. Let $\epsilon_1 = \Delta_\mu$, $\epsilon_k = \epsilon_{k-1}/2$, $\beta_k = \min\{\frac{\mu \epsilon_k}{c\sigma^2}, \frac{1}{c}\}, \eta_k = \min\{\frac{\sqrt{\mu \epsilon_k}}{24 c L_F \sigma^2}, \frac{1}{24 c L_F^2}\}$ and $T_k = \max\{\frac{192 c L_F \sigma}{\mu^{3/2}\sqrt{\epsilon_k}}, \frac{192 c L_F^2 \sigma^2}{\mu \epsilon_k}, \frac{192 c L_F^2}{\mu}\}$, where $c = 768 L_F^2$. Run RASCDRO, then we have $\mathbb{E}[F_\mu(\mathbf{x}_k) - \inf_{\mathbf{x} \in \mathcal{X}} F_\mu(\mathbf{x})] \leq \epsilon_k$ for each stage $k$.*

The above lemma implies that the objective gap $\mathbb{E}[F_\mu(\mathbf{x}_k) - \inf_{\mathbf{x} \in \mathcal{X}} F_\mu(\mathbf{x})]$ is decreased by a factor of 2 after each stage. Hence we have the following convergence rate for the RASCDRO.

**Theorem 4.** *Under the same assumptions and parameter settings as Lemma 5, after $K = \mathcal{O}(\log_2(\epsilon_1/\epsilon))$ stages, the output of RASCDRO satisfies $\mathbb{E}[F_\mu(\mathbf{x}_K) - \inf_{\mathbf{x} \in \mathcal{X}} F_\mu(\mathbf{x})] \leq \epsilon$, and the oracle complexity is $\mathcal{O}\left(\max\left(1/\mu\epsilon, 1/\mu^{3/2}\sqrt{\epsilon}\right)\right)$.*

By the same method of derivation of Corollary 1, the following corollary of Theorem 4 holds.

**Corollary 2.** *Let $\mu = \epsilon/(2(R^2 + \tilde{\lambda}^2))$. Then under the same assumptions and parameter settings as Lemma 5, after $K = \mathcal{O}(\log_2(\epsilon_1/\epsilon))$ stages, the output of RASCDRO satisfies $\mathbb{E}[F(\mathbf{x}_K) - \inf_{\mathbf{x} \in \mathcal{X}} F(\mathbf{x})] \leq \epsilon$ and the oracle complexity is $\mathcal{O}(1/\epsilon^2)$.*

**Remark:** Corollary 2 shows that RASCDRO achieves the claimed oracle complexity $\mathcal{O}(1/\epsilon^2)$ for finding an $\epsilon$-optimal solution, which is optimal for solving convex smooth optimization problems (Nemirovsky & Yudin, 1983). Finally, we note that a similar complexity was established in (Zhang & Lan, 2021) for constrained convex compositional optimization problems. However, their analysis requires each level function to be convex, which does not apply to our case as the outer function $f_\lambda(\cdot)$ is non-convex.

# 6  Experiments

In this section, we verify the effectiveness of the proposed algorithms in solving imbalanced classification problems. We show that the proposed methods outperform baselines under both the convex and non-convex settings in terms of convergence speed, and generalization performance. In addition, we study the influence of $\rho$ to the robustness of different optimization methods in the supplement. All our results are conducted on Tesla V100.

**Baselines.** For the comparison of convergence speed, we compare with different algorithms for optimizing the same objective (1), including, stochastic primal-dual algorithms, namely PG-SMD2 (Rafique et al., 2021) for a non-convex loss, and SPD (Namkoong & Duchi, 2016) for a convex loss, Dual SGM (Levy et al., 2020) and mini-batch based SGD named FastDRO (Levy et al., 2020) for both convex and non-convex losses. For the comparison of generalization performance, we compare with different methods for optimizing different objectives, including the traditional ERM with CE loss by SGD with momentum (SGDM), KL-regularized DRO solved by RECOVER (Qi et al., 2021), ABSGD (Qi et al., 2020a; Li et al., 2021b) and CVaR-constrained, $\chi^2$-regularized/-constrained DRO optimized by FastDRO.

**Datasets.** We conduct experiments on four imbalanced datasets, namely CIFAR10-ST, CIFAR100-ST (Qi et al., 2020b), ImageNet-LT (Liu et al., 2019), and iNaturalist2018 (iNaturalist 2018 competition dataset). The original CIFAR10, CIFAR100 are balanced data, where CIFAR10 (resp. CIFAR100) has 10 (resp. 100) classes and each class has 5K (resp. 500) training images. For constructing CIFAR10-ST and CIFAR100-ST, we artificially construct imbalanced training data, where we only keep the last 100 images of each class for the first half classes, and keep other classes and the test data unchanged. ImageNet-LT is a long-tailed subset of the original ImageNet-2012 by sampling a subset following the Pareto distribution with the power value 6. It has 115.8K images from 1000 categories, which include 4980 for head class and 5 images for tail class. iNaturalist 2018 is a real-world dataset whose class-frequency follows a heavy-tail distribution. It contains 437K images from 8142 classes.

**Models.** For a non-convex setting (deep model), we learn ResNet20 for CIFAR10-ST, CIFAR100-ST, and ResNet50 for ImageNet-LT and iNaturalist2018, respectively. On CIFAR10-ST, CIFAR100-ST, we optimize the network from scratch by different algorithms. For the large-scale ImageNet-LT and iNaturalist2018 datasets, we optimize the last block of the feature layers and the classifier weight with other layers frozen of a pretrained ResNet50 model. This is a common training strategy in the literature (Kang et al., 2019; Qi et al., 2020a). For a convex setting (linear model), we freeze the feature layers of the pretrained models, and only fine-tune the last classifier weight. The pretrained models for ImageNet-LT, CIFAR10-ST, CIFAR100-ST are trained from scratch by optimizing the standard cross-entropy (CE) loss using SGD with momentum 0.9 for 90 epochs. The pretrained ResNet50 model for iNaturalist2018 is from the released model by Kang et al. (2019).

**Parameters and Settings.** For all experiments, the batch size is 128 for CIFAR10-ST and CIFAR100-ST, and 512 for ImageNet-LT and iNaturalist2018. The loss function is the CE loss. The $\lambda_0$ is set to 1$e$-3. The (primal) learning rates for all methods are tuned in $\{0.01, 0.05, 0.1, 0.5, 1\}$. The learning rate for updating the dual variable in PG_SMD2 and SPD is tuned in $\{1e\text{-}5, 5e\text{-}5, 1e\text{-}4, 5e\text{-}4)\}$. The momentum parameter $\beta$ in our proposed algorithms and RECOVER are tuned $\{0.1 : 0.1 : 0.9\}$. For RECOVER, the hyper-parameter $\lambda$ is tuned in $\{1, 50, 100\}$. The constrained parameter $\rho$ is tuned in $\{0.1, 0.5, 1\}$ for the comparison of generalization

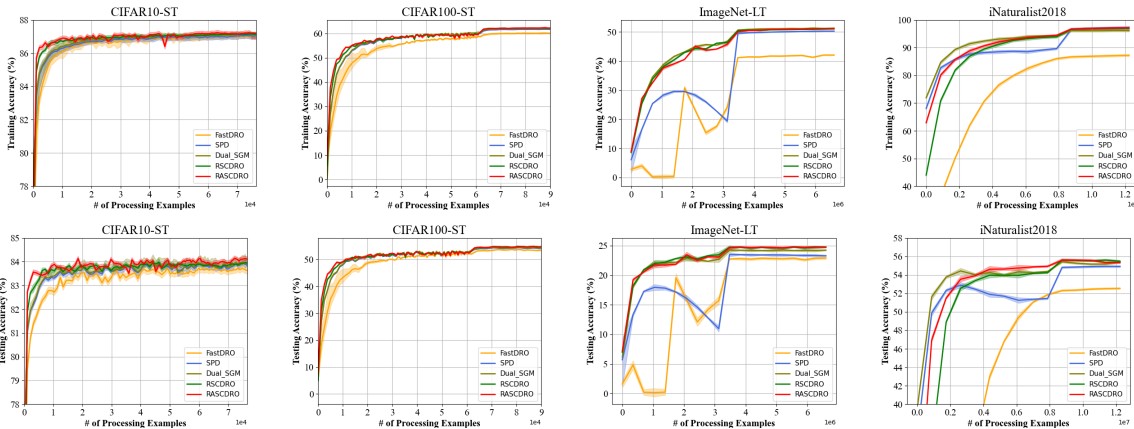

Figure 1: Training accuracy (%) , Testing accuracy (%) vs # of processed training samples for the convex setting. $\rho$ is fixed to 0.5 on CIFAR10-ST and CIFAR100-ST, and 0.1 on ImageNet-LT and iNaturalist2018. The results are averaged over 5 independent runs.

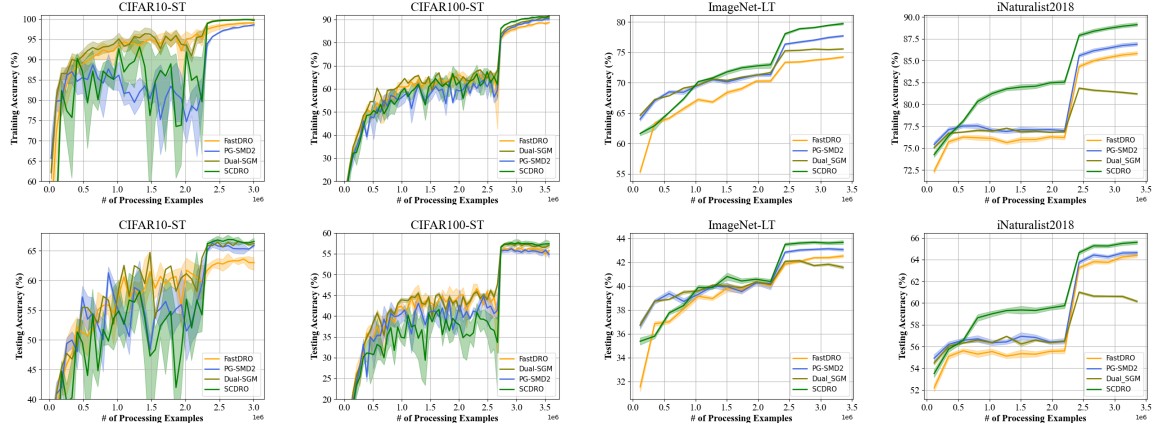

Figure 2: Training accuracy (%), Testing accuracy (%) vs # of processed training samples for the non-convex setting. $\rho$ is fixed to 0.5 on all datasets. The results are averaged over 5 independent runs.

performance unless specified otherwise. The initial $\lambda$ and Larange multiplier in Dual SGM are both tuned in $\{0.1, 1, 10\}$. All our results are conducted on Tesla V100.

**Convergence comparison between different baselines.** In the convex setting, we compare RSCDRO and RASCDRO with SPD, FastDRO and Dual SGM baselines. We report the training accuracy and testing accuracy in terms of the number (#) of processing samples. We denote 1 pass of training data by 1 epoch. We run a total of 3 epochs for CIFAR10-ST and CIFAR100-ST and decay the learning rate by a factor of 10 at the end of 2nd epoch. Similarly, we run 60 epochs and decay the learning rate at the 30th epochs for the ImageNet-LT, and run 30 epochs and decay the learning rate at the 20th epoch for iNaturalist2018. In the nonconvex setting, we compare SCDRO with two baselines, PG-SMD2 and FastDRO. We run 120 epochs for CIFAR10-ST and CIFAR100-ST, and decay the learning rate by a factor of 10 at the 90th epoch. And we run 30 epochs for ImageNet-LT and iNaturalist2018, and decay the learning rate at the 20th epoch.

**Results.** We first report the results for convex setting in Figures 1. It is obvious to see that RSCDRO and RASCDRO are consistently better than baselines on CIFAR10-ST, CIFAR100-ST, and ImageNet-LT. PD-SMD2 and Dual SGM have comparable results with our proposed algorithms on the iNaturalist2018 in terms of training accuracy, but is worse in terms of testing accuracy. FastDRO has the worst performance on all the datasets. RSCDRO and RASCDRO achieve comparable results on all datasets, however, the stochastic estimator in RASCDRO requires two gradient computations per iteration, which incurs more computational cost than RSCDRO. Hence, in the non-convex setting, we focus on SCDRO. Figure 2 reports the results

Table 2: Testing Accuracy in Convex Setting

|  | ImageNet-LT | iNaturalist2018 |
|---|---|---|
| KL-Constraint + SCDRO | **24.08** ($\pm$ 0.01) | **55.63** ($\pm$ 0.03) |
| CVaR-Constraint + FastDRO | 17.23 ($\pm$ 0.03) | 54.52 ($\pm$ 0.11) |
| $\chi^2$-Regularization + FastDRO | 23.98 ($\pm$ 0.01) | 55.03 ($\pm$ 0.03) |
| $\chi^2$-Constraint + FastDRO | 23.61 ($\pm$ 0.01) | 53.71 ($\pm$ 0.05) |

Table 3: Testing Accuracy in Non-Convex Setting

|  | ImageNet-LT | iNaturalist2018 |
|---|---|---|
| KL-Constraint + SCDRO | **43.74** | **65.59** |
| ERM+SGDM | 43.36 | 64.42 |
| KL-Regularization + RECOVER | 42.68 | 64.57 |
| KL-Regularization + ABSGD | 43.44 | 65.01 |

for non-convex setting. We can see that SCDRO achieves the best performance on all the datasets. The margin increases on the large scale ImageNet-LT and iNaturalist2018 datasets. For the three baselines, Dual SGM has better testing performance than FastDRO and PD-SGM2 on CIFAR10-ST and CIFAR100-ST. On the large scale data ImageNet-LT and iNaturalist2018, however, Dual SGM has the worst performance in terms of the testing accuracy. Furthermore, SCDRO is more stable than FastDRO and Dual SGM in different settings as the training of Dual SGM and FastDRO is comparable to SCDRO in convex settings and much worse than SCDRO in non-convex settings.

**Comparison with ERM and KL-regularized DRO.** Next, we compare our method for solving KL-constrained DRO (KL-CDRO) with 1) ERM+SGDM, and KL-regularized DRO (KL-RDRO) optimized by RECOVER, ABSGD in the non-convex setting 2) CVaR-constrained DRO, $\chi^2$-regularized DRO $\chi^2$-constrained DRO optimized by FastDRO in the convex setting. We conduct the experiments on the large-scale ImageNet-LT and iNaturalist2018 datasets. The results shown in Table 2 and 3 vividly demonstrate that our method for constrained DRO outperforms the ERM-based method and other popular $f$-divergence constrained/regularized DRO in different settings.

**Sensitivity to $\rho$.** We study the sensitivity of different methods to $\rho$. The results on CIFAR10-ST and CIFAR100-ST are shown in Table 4 in the supplement, which demonstrates that the testing performance is sensitive to $\rho$. However, our method SCDRO is better than baselines PG-SMD2 and FastDRO for different values of $\rho$.

Table 4: Test accuracy (%) of different methods for different constraint parameter $\rho$ in the non-convex setting. The results are averaged over 5 independent runs.

|  | $\rho$ | 0.01 | 0.05 | 0.1 | 0.5 | 1 |
|---|---|---|---|---|---|---|
| CIFAR10-ST | PG-SMD2 | 67.09 ($\pm$ 0.59) | 66.96 ($\pm$ 0.71) | 67.12 ($\pm$ 0.61) | 67.36 ($\pm$ 0.36) | 67.10 ($\pm$ 0.61) |
|  | FastDRO | 65.41 ($\pm$ 0.33) | 66.15 ($\pm$ 0.09) | 66.24 ($\pm$ 0.63) | 65.98 ($\pm$ 0.45) | 65.68 ($\pm$ 0.52) |
|  | SCDRO | **67.73** ($\pm$ 0.39) | **67.58** ($\pm$ 0.48) | **67.71** ($\pm$ 0.43) | **67.57** ($\pm$ 0.28) | **67.96** ($\pm$ 0.50) |
| CIFAR100-ST | PG-SMD2 | 57.31 ($\pm$ 0.09) | 56.44 ($\pm$ 0.17) | 55.85 ($\pm$ 0.19) | 52.68 ($\pm$ 0.40) | 48.72 ($\pm$ 0.25) |
|  | FastDRO | 57.60 ($\pm$ 0.32) | 57.20 ($\pm$ 0.42) | 56.78 ($\pm$ 0.40) | 55.58 ($\pm$ 0.62) | 52.39 ($\pm$ 0.31) |
|  | SCDRO | **57.84** ($\pm$ 0.15) | **57.60** ($\pm$ 0.15) | **58.32** ($\pm$ 0.43) | **57.90** ($\pm$ 0.26) | **57.71** ($\pm$ 0.24) |

## 7 Conclusions

In this paper, we proposed dual-free stochastic algorithms for solving KL-constrained distributionally robust optimization problems for both convex and non-convex losses. The proposed algorithms have nearly optimal complexity in both settings. Empirical studies vividly demonstrate the effectiveness of the proposed algorithm for solving non-convex and convex constrained DRO problems.

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

## A  Preliminary Lemmas

**Lemma 6.** *For $q \geq 1$, $f_\lambda(q) = \lambda \log(q) + \lambda \rho$ is $L_{f_\lambda}$-Lipschitz continuous and $L_{\nabla f_\lambda}$-smooth, where $L_{\nabla f_\lambda} = L_{f_\lambda} = \lambda$.*

**Remark:**  $g_i(\mathbf{w}, \lambda) = \exp(\frac{\ell_i(\mathbf{w})}{\lambda}) \geq 1$ as $\lambda \geq \lambda_0 \in \mathbb{R}^+$ and $\ell_i(\mathbf{w}) \geq 0$ in problem (3). Thus $g(\mathbf{x}) = \frac{1}{n} \sum_{i=1}^n g_i(\mathbf{w}, \lambda) \geq 1$. Then by this lemma we have $\|\nabla f_\lambda(g(\mathbf{x}))\| \leq \lambda$ and $\|\nabla f_\lambda(g(\mathbf{x}_1)) - \nabla f_\lambda(g(\mathbf{x}_2))\| \leq \lambda \|g(\mathbf{x}_1) - g(\mathbf{x}_2)\|$ for $\mathbf{x}, \mathbf{x}_1, \mathbf{x}_2 \in \mathcal{X}$.

*Proof.* For any $q \geq 1$, we have

$$\nabla f_\lambda(q) = \frac{\lambda}{q} \leq \lambda$$

And for any $q_1, q_2 \geq 1$, we have

$$\|\nabla f_\lambda(q_1) - \nabla f_\lambda(q_2)\| \leq \left\| \frac{\lambda}{q_1} - \frac{\lambda}{q_2} \right\| \leq \left\| \frac{(q_1 - q_2)\lambda}{q_1 q_2} \right\| \leq \lambda \|q_1 - q_2\|$$

This complete the proof.

$\square$

**Lemma 7.** *Let $L_A = \exp(\frac{C}{\lambda_0})(\frac{G^2}{\lambda_0^2} + \frac{L}{\lambda_0})$, $L_B = \exp(\frac{C}{\lambda_0})(\frac{CG}{\lambda_0^3} + \frac{G}{\lambda_0^2})$, $L_C = \exp(\frac{C}{\lambda_0})(\frac{CG + \lambda_0 G}{\lambda_0^3})$ and $L_D = \exp(\frac{C}{\lambda_0})(\frac{C^2 + 2\lambda_0 C}{\lambda_0^4})$. $g_i(\mathbf{w}, \lambda)$ is $L_g$-Lipschitz continuous and $L_{\nabla g}$-smooth in terms of $(\mathbf{w}, \lambda)$, where $L_g = \exp(\frac{C}{\lambda_0})(\frac{G}{\lambda_0} + \frac{C}{\lambda_0^2})$ and $L_{\nabla g} = \sqrt{L_A^2 + L_B^2 + L_C^2 + L_D^2}$,*

*Proof.* The gradient of $g_i(\mathbf{w}, \lambda)$ is given as

$$\nabla_{\mathbf{w},\lambda} g_i(\mathbf{w}, \lambda)^\top = (\nabla_{\mathbf{w}} g_i(\mathbf{w}, \lambda)^\top, \nabla_\lambda g_i(\mathbf{w}, \lambda))$$

$$= \left( \exp\left( \frac{\ell_i(\mathbf{w})}{\lambda} \right) \frac{\nabla_{\mathbf{w}} \ell_i(\mathbf{w})}{\lambda}^\top, -\exp\left( \frac{\ell_i(\mathbf{w})}{\lambda} \right) \frac{\ell_i(\mathbf{w})}{\lambda^2} \right).$$

Then by Assumption 1, we have

$$\|\nabla_{\mathbf{w},\lambda} g_i(\mathbf{w}, \lambda)\| \leq \exp\left( \frac{\ell_i(\mathbf{w})}{\lambda} \right) \left( \left\| \frac{\nabla_{\mathbf{w}} \ell_i(\mathbf{w})}{\lambda} \right\| + \frac{\ell_i(\mathbf{w})}{\lambda^2} \right)$$

$$\overset{\lambda \geq \lambda_0}{\leq} \exp\left( \frac{C}{\lambda_0} \right) \left( \frac{G}{\lambda_0} + \frac{C}{\lambda_0^2} \right).$$

Thus, $L_g = \exp\left( \frac{C}{\lambda_0} \right) \left( \frac{G}{\lambda_0} + \frac{C}{\lambda_0^2} \right)$.

For for all $(\mathbf{w}, \lambda), (\mathbf{w}', \lambda') \in \mathcal{X}$, we have

$$\|\nabla_{\mathbf{w},\lambda} g_i(\mathbf{w}, \lambda) - \nabla_{\mathbf{w},\lambda} g_i(\mathbf{w}', \lambda')\|^2$$

$$\leq \left\| \exp\left( \frac{\ell_i(\mathbf{w})}{\lambda} \right) \frac{\nabla_{\mathbf{w}} \ell_i(\mathbf{w})}{\lambda} + \exp\left( \frac{\ell_i(\mathbf{w}')}{\lambda'} \right) \frac{\nabla_{\mathbf{w}} \ell_i(\mathbf{w}')}{\lambda'} \right\|^2$$

$$+ \left\| \exp\left( \frac{\ell_i(\mathbf{w})}{\lambda} \right) \frac{\ell_i(\mathbf{w})}{\lambda^2} - \exp\left( \frac{\ell_i(\mathbf{w}')}{\lambda'} \right) \frac{\ell_i(\mathbf{w}')}{\lambda'^2} \right\|^2$$

$$\leq \left\| \exp\left( \frac{\ell_i(\mathbf{w})}{\lambda} \right) \frac{\nabla_{\mathbf{w}} \ell_i(\mathbf{w})}{\lambda} - \exp\left( \frac{\ell_i(\mathbf{w}')}{\lambda} \right) \frac{\nabla_{\mathbf{w}} \ell_i(\mathbf{w}')}{\lambda} \right\|^2$$

$$+ \left\| \exp\left( \frac{\ell_i(\mathbf{w}')}{\lambda} \right) \frac{\nabla_{\mathbf{w}} \ell_i(\mathbf{w}')}{\lambda} - \exp\left( \frac{\ell_i(\mathbf{w}')}{\lambda'} \right) \frac{\nabla_{\mathbf{w}} \ell_i(\mathbf{w}')}{\lambda'} \right\|^2$$

$$+ \left\| \exp\left( \frac{\ell_i(\mathbf{w})}{\lambda} \right) \frac{\ell_i(\mathbf{w})}{\lambda^2} - \exp\left( \frac{\ell_i(\mathbf{w}')}{\lambda} \right) \frac{\ell_i(\mathbf{w}')}{\lambda^2} \right\|^2$$

$$+ \left\| \exp\left( \frac{\ell_i(\mathbf{w}')}{\lambda} \right) \frac{\ell_i(\mathbf{w}')}{\lambda^2} - \exp\left( \frac{\ell_i(\mathbf{w}')}{\lambda'} \right) \frac{\ell_i(\mathbf{w}')}{\lambda'^2} \right\|^2.$$

To bound the first term, we first check the Lipschitz continuous of $\exp(\frac{\ell_i(\mathbf{w})}{\lambda})\frac{\nabla_\mathbf{w}\ell_i(\mathbf{w})}{\lambda}$ with respect to $\mathbf{w}$,

$$\left\|\frac{\nabla\left(\exp\left(\frac{\ell_i(\mathbf{w})}{\lambda}\right)\frac{\nabla_\mathbf{w}\ell_i(\mathbf{w})}{\lambda}\right)}{\nabla\mathbf{w}}\right\|$$

$$\leq \left\|\exp\left(\frac{\ell_i(\mathbf{w})}{\lambda}\right)\left(\frac{\nabla_\mathbf{w}\ell_i(\mathbf{w})}{\lambda}\right)\left(\frac{\nabla_\mathbf{w}\ell_i(\mathbf{w})}{\lambda}\right)^\top\right\| + \left\|\exp\left(\frac{\ell_i(\mathbf{w})}{\lambda}\right)\frac{\nabla_\mathbf{w}^2\ell_i(\mathbf{w})}{\lambda}\right\|$$

$$\overset{(a)}{=} \left\|\exp\left(\frac{\ell_i(\mathbf{w})}{\lambda}\right)\left(\frac{\nabla_\mathbf{w}\ell_i(\mathbf{w})}{\lambda}\right)^\top\left(\frac{\nabla_\mathbf{w}\ell_i(\mathbf{w})}{\lambda}\right)\right\| + \left\|\exp\left(\frac{\ell_i(\mathbf{w})}{\lambda}\right)\frac{\nabla_\mathbf{w}^2\ell_i(\mathbf{w})}{\lambda}\right\|$$

$$\overset{(b)}{\leq} \exp\left(\frac{\ell_i(\mathbf{w})}{\lambda}\right)\left\|\left(\frac{\nabla_\mathbf{w}\ell_i(\mathbf{w})}{\lambda}\right)\right\|^2 + \left\|\exp\left(\frac{\ell_i(\mathbf{w})}{\lambda}\right)\frac{\nabla_\mathbf{w}^2\ell_i(\mathbf{w})}{\lambda}\right\|$$

$$\leq \exp\left(\frac{C}{\lambda_0}\right)\left(\frac{G^2}{\lambda_0^2} + \frac{L}{\lambda_0}\right) := L_A.$$

where equality (a) is due to the property of the norm of rank-one symmetric matrix and inequality (b) is due to Cauchy-Schwarz inequality.

Therefore, we have

$$\left\|\exp(\frac{\ell_i(\mathbf{w})}{\lambda})\frac{\nabla_\mathbf{w}\ell_i(\mathbf{w})}{\lambda} - \exp(\frac{\ell_i(\mathbf{w}')}{\lambda})\frac{\nabla_\mathbf{w}\ell_i(\mathbf{w}')}{\lambda}\right\|^2 \leq L_A\|\mathbf{w}-\mathbf{w}'\|^2$$

Furthermore, it holds that

$$\left\|\frac{\nabla\left(\exp\left(\frac{\ell_i(\mathbf{w})}{\lambda}\right)\frac{\nabla_\mathbf{w}\ell_i(\mathbf{w})}{\lambda}\right)}{\nabla\lambda}\right\| = \left\|\exp\left(\frac{\ell_i(\mathbf{w})}{\lambda}\right)\frac{\ell_i(\mathbf{w})\nabla_\mathbf{w}\ell_i(\mathbf{w})}{\lambda^3} + \exp\left(\frac{\ell_i(\mathbf{w})}{\lambda}\right)\left(\frac{\nabla_\mathbf{w}\ell_i(\mathbf{w})}{\lambda^2}\right)\right\|$$

$$\leq \exp\left(\frac{C}{\lambda_0}\right)\left(\frac{CG}{\lambda_0^3} + \frac{G}{\lambda_0^2}\right) := L_B$$

$$\left\|\frac{\nabla\left(\exp\left(\frac{\ell_i(\mathbf{w})}{\lambda}\right)\frac{\ell_i(\mathbf{w})}{\lambda^2}\right)}{\nabla\mathbf{w}}\right\| = \left\|\exp\left(\frac{\ell_i(\mathbf{w})}{\lambda}\right)\frac{\ell_i(\mathbf{w})\nabla_\mathbf{w}\ell_i(\mathbf{w})}{\lambda^3} + \exp\left(\frac{\ell_i(\mathbf{w})}{\lambda}\right)\frac{\nabla_\mathbf{w}\ell_i(\mathbf{w})}{\lambda^2}\right\|$$

$$\leq \exp\left(\frac{C}{\lambda_0}\right)\left(\frac{CG + \lambda_0 G}{\lambda_0^3}\right) := L_C$$

$$\left\|\frac{\nabla\left(\exp\left(\frac{\ell_i(\mathbf{w})}{\lambda}\right)\frac{\ell_i(\mathbf{w})}{\lambda^2}\right)}{\nabla\lambda}\right\| = \left\|\exp\left(\frac{\ell_i(\mathbf{w})}{\lambda}\right)\frac{\ell_i^2(\mathbf{w})}{\lambda^4} + \exp\left(\frac{\ell_i(\mathbf{w})}{\lambda}\right)\frac{2\ell_i(\mathbf{w})}{\lambda^3}\right\|$$

$$\leq \exp\left(\frac{C}{\lambda_0}\right)\left(\frac{C^2 + 2\lambda_0 C}{\lambda_0^4}\right) := L_D.$$

As a result, we obtain

$$\|\nabla_{\mathbf{w},\lambda}g_i(\mathbf{w},\lambda) - \nabla_{\mathbf{w},\lambda}g_i(\mathbf{w}',\lambda')\|^2$$
$$\leq L_A^2\|\mathbf{w}-\mathbf{w}'\|^2 + L_B^2\|\lambda-\lambda'\|^2 + L_C^2\|\mathbf{w}-\mathbf{w}'\|^2 + L_D^2\|\lambda-\lambda'\|^2$$
$$= (L_A^2 + L_C^2)\|\mathbf{w}-\mathbf{w}'\|^2 + (L_B^2 + L_D^2)\|\lambda-\lambda'\|^2$$
$$\leq (L_A^2 + L_B^2 + L_C^2 + L_D^2)\left\|(\mathbf{w}^\top,\lambda)-(\mathbf{w}'^\top,\lambda')\right\|^2.$$

Thus $L_{\nabla_g} = \sqrt{L_A^2 + L_B^2 + L_C^2 + L_D^2}$.

$\square$

**Lemma 8.** $F(\mathbf{w},\lambda)$ is $L_F$-smooth, where $L_F = \tilde{\lambda}L_g^2 + 2L_g + \tilde{\lambda}L_{\nabla_g} + 1 + \tilde{\lambda}$.

**Remark:** Lemma 6, 7 and Lemma 8 imply that $L_{\nabla f_\lambda} = L_{f_\lambda} \leq L_F, L_g \leq L_F$ and $L_F \geq 1$.

*Proof.* For all $\mathbf{x}_1 = (\mathbf{w}_1^\top, \lambda_1)^\top, \mathbf{x}_2 = (\mathbf{w}_2^\top, \lambda_2)^\top \in \mathcal{X}$, and let $\mathbf{d}(\mathbf{x}) = (0, \cdots, 0, \log(g(\mathbf{x})) + \rho)^\top \in \mathbb{R}^{d+1}$, by expansion we have

$$\|\nabla F(\mathbf{x}_1) - \nabla F(\mathbf{x}_2)\|$$
$$= \|\nabla f_{\lambda_1}(g(\mathbf{x}_1))\nabla g(\mathbf{x}_1) + \mathbf{d}(\mathbf{x}_1) - \nabla f_{\lambda_2}(g(\mathbf{x}_2))\nabla g(\mathbf{x}_2) - \mathbf{d}(\mathbf{x}_2)\|$$
$$\leq \|\nabla f_{\lambda_1}(g(\mathbf{x}_1))\nabla g(\mathbf{x}_1) - \nabla f_{\lambda_2}(g(\mathbf{x}_2))\nabla g(\mathbf{x}_2)\| + |\log(g(\mathbf{x}_1)) - \log(g(\mathbf{x}_2))|$$
$$\leq \|\nabla f_{\lambda_1}(g(\mathbf{x}_1))\nabla g(\mathbf{x}_1) - \nabla f_{\lambda_1}(g(\mathbf{x}_2))\nabla g(\mathbf{x}_1)\| + \|\nabla f_{\lambda_1}(g(\mathbf{x}_2))\nabla g(\mathbf{x}_1) - \nabla f_{\lambda_2}(g(\mathbf{x}_2))\nabla g(\mathbf{x}_1)\|$$
$$+ \|\nabla f_{\lambda_2}(g(\mathbf{x}_2))\nabla g(\mathbf{x}_1) - \nabla f_{\lambda_2}(g(\mathbf{x}_2))\nabla g(\mathbf{x}_2)\| + |g(\mathbf{x}_1) - g(\mathbf{x}_2)|.$$

Noting the Lipschtiz continuous of $g(x)$ and $\nabla g(x)$, we obtain

$$\|\nabla F(\mathbf{x}_1) - \nabla F(\mathbf{x}_2)\|$$
$$\leq (L_{\nabla f_{\lambda_1}} L_g + 1)|g(\mathbf{x}_1) - g(\mathbf{x}_2)| + \frac{\|\nabla g(\mathbf{x}_1)\|}{g(\mathbf{x}_2)}\|\lambda_1 - \lambda_2\| + L_{f_{\lambda_2}}\|\nabla g(\mathbf{x}_1) - \nabla g(\mathbf{x}_2)\|$$
$$\overset{(a)}{\leq} (L_{\nabla f_{\lambda_1}} L_g^2 + L_g)\|\mathbf{x}_1 - \mathbf{x}_2\| + \|\nabla g(\mathbf{x}_1)\|\|\lambda_1 - \lambda_2\| + L_{f_{\lambda_2}} L_{\nabla_g}\|\mathbf{x}_1 - \mathbf{x}_2\|$$
$$\leq (L_{\nabla f_{\lambda_1}} L_g^2 + 2L_g + L_{f_{\lambda_2}} L_{\nabla_g})\|\mathbf{x}_1 - \mathbf{x}_2\|$$
$$\overset{(b)}{\leq} (\tilde{\lambda} L_g^2 + 2L_g + \tilde{\lambda} L_{\nabla_g} + 1 + \tilde{\lambda})\|\mathbf{x}_1 - \mathbf{x}_2\|.$$

where the inequality (a) is due to $g(\mathbf{x}_2) \geq 1$ and the inequality (b) is due to the upper bound of $\lambda$. Thus, $L_F = \tilde{\lambda} L_g^2 + 2L_g + \tilde{\lambda} L_{\nabla_g} + 1 + \tilde{\lambda}$. $\qquad\square$

## A.1   Proof of Lemma 1

*Proof.* Recall the primal problem:

$$p^* = \max_{\{\mathbf{p} \in \Delta_n, D(\mathbf{p}, 1/n) \leq \rho\}} \sum_{i=1}^n p_i \ell_i(\mathbf{w}) + \lambda_0 D(\mathbf{p}, 1/n).$$

Invoking dual variable $\bar{\lambda}$, we obtain the dual problem:

$$q^* = \min_{\bar{\lambda} \geq 0} \max_{\mathbf{p} \in \Delta_n} \sum_{i=1}^n p_i \ell_i(\mathbf{w}) - \bar{\lambda}(D(\mathbf{p}, 1/n) - \rho) - \lambda_0 D(\mathbf{p}, 1/n). \tag{9}$$

Set $\bar{\mathbf{p}} = (1/n, \ldots, 1/n)$, which is a Slater vector satisfying $D(\bar{\mathbf{p}}, 1/n) - \rho < 0$. Applying Lemma 3 in (Nedić & Ozdaglar, 2009), we have

$$|\bar{\lambda}^*| \leq \frac{1}{\rho}\left(q^* - \sum_{i=1}^n \bar{p}_i \ell_i(\mathbf{w}) - \lambda_0 D(\bar{\mathbf{p}}, 1/n)\right).$$

Since the primal problem is concave in term of $\mathbf{p}$ given $\mathbf{w}$, we have $p^* = q^*$. Therefore,

$$|\bar{\lambda}^*| \leq \frac{1}{\rho}\left(p^* - \sum_{i=1}^n \bar{p}_i \ell_i(\mathbf{w})\right)$$
$$= \frac{1}{\rho}\left(\sum_{i=1}^n \bar{p}_i^* \ell_i(\mathbf{w}) - \lambda_0 D(\mathbf{p}^*, 1/n) - \sum_{i=1}^n \bar{p}_i \ell_i(\mathbf{w})\right)$$
$$\leq \frac{C}{\rho}, \tag{10}$$

where the last inequality is because $|\ell_i(\mathbf{w})| \leq C$ for $\mathbf{w} \in \mathcal{W}$. Let $\lambda = \bar{\lambda} + \lambda_0$, we have

$$q^* = \min_{\lambda \geq \lambda_0} \max_{\mathbf{p} \in \Delta_n} \sum_{i=1}^n p_i \ell_i(\mathbf{w}) - \lambda(D(\mathbf{p}, 1/n) - \rho) - \lambda_0 \rho.$$

Section E will also show

$$q^* = \min_{\lambda \geq \lambda_0} \lambda \log\left(\frac{1}{n}\sum_{i=1}^n \exp\left(\frac{\ell_i(\mathbf{w})}{\lambda}\right)\right) + \lambda(\rho - \rho_0).$$

By Eq. (10), we have the optimal solution of above optimization problem $|\lambda^*| \leq |\bar{\lambda}^*| + \lambda_0 \leq \lambda_0 + \frac{C}{\rho}$, which complete the proof

$\square$

# B  Proofs in Section 4

## B.1  Technical Lemmas

**Lemma 9.** *Suppose Assumption 2 holds and $i \sim \mathcal{D}$ and $s$ are initialized with $s_1 = \exp(\frac{\ell_i(\mathbf{w}_1)}{\lambda_1})$. Then for every $t \in \{1, \cdots T\}$ we have*

$$\mathbb{E}[\|g(\mathbf{x}_{t+1}) - s_{t+1}\|^2] \leq \mathbb{E}\left[(1-\beta)\|g(\mathbf{x}_t) - s_t\|^2 + \frac{2L_g^2\|\mathbf{x}_{t+1} - \mathbf{x}_t\|^2}{\beta} + \beta^2\sigma^2\right].$$

*Taking summation of $\mathbb{E}[\|g(\mathbf{x}_{t+1}) - s_{t+1}\|^2]$ from 1 to T, we have*

$$\sum_{t=1}^{T}\mathbb{E}[\|g(\mathbf{x}_t) - s_t\|^2] \leq \mathbb{E}\left[\frac{\|g(\mathbf{x}_1) - s_1\|^2}{\beta} + \frac{2L_g^2}{\beta^2}\sum_{t=1}^{T}\|\mathbf{x}_{t+1} - \mathbf{x}_t\|^2 + \beta T\sigma^2\right]. \tag{11}$$

*Proof.* Note that $s_{t+1} = (1-\beta)s_t + \beta g_i(\mathbf{x}_{t+1})$ and $\mathbb{E}[g(\mathbf{x}_{t+1}) - g_i(\mathbf{x}_{t+1})] = 0$, then by simple expansion we have

$$\begin{aligned}
&\mathbb{E}[\|g(\mathbf{x}_{t+1}) - s_{t+1}\|^2]\\
&= \mathbb{E}[\|\beta(g(\mathbf{x}_{t+1}) - g_i(\mathbf{x}_{t+1})) + (1-\beta)(g(\mathbf{x}_{t+1}) - s_t)\|^2]\\
&= \mathbb{E}[\beta^2\|g(\mathbf{x}_{t+1}) - g_i(\mathbf{x}_{t+1})\|^2 + (1-\beta)^2\|g(\mathbf{x}_{t+1}) - s_t\|^2]\\
&\quad + 2\underbrace{\mathbb{E}[\langle g(\mathbf{x}_{t+1}) - g_i(\mathbf{x}_{t+1}), g(\mathbf{x}_{t+1}) - s_t\rangle]}_{0}\\
&= \mathbb{E}[\beta^2\|g(\mathbf{x}_{t+1}) - g_i(\mathbf{x}_{t+1})\|^2 + (1-\beta)^2\|g(\mathbf{x}_{t+1}) - g(\mathbf{x}_t) + g(\mathbf{x}_t) - s_t\|^2]. \tag{12}
\end{aligned}$$

Invkoing Lemma 7 to Eq. (12) and recalling Assumption 2 , we obtain

$$\begin{aligned}
&\mathbb{E}[\|g(\mathbf{x}_{t+1}) - s_{t+1}\|^2]\\
&\overset{(a)}{\leq} \mathbb{E}[\beta^2\|g(\mathbf{x}_{t+1}) - g_i(\mathbf{x}_{t+1})\|^2 + (1-\beta)^2(1+\beta)\|g(\mathbf{x}_t) - s_t\|^2\\
&\quad + (1+\frac{1}{\beta})(1-\beta)^2\|g(\mathbf{x}_{t+1}) - g(\mathbf{x}_t)\|^2\\
&\overset{(b)}{\leq} \mathbb{E}\left[\beta^2\|g(\mathbf{x}_{t+1}) - g_i(\mathbf{x}_{t+1})\|^2 + (1-\beta)\|g(\mathbf{x}_t) - s_t\|^2 + \frac{2L_g^2\|\mathbf{x}_{t+1} - \mathbf{x}_t\|^2}{\beta}\right]\\
&\overset{(c)}{\leq} \mathbb{E}\left[(1-\beta)\|g(\mathbf{x}_t) - s_t\|^2 + \frac{2L_g^2\|\mathbf{x}_{t+1} - \mathbf{x}_t\|^2}{\beta} + \beta^2\sigma^2\right].
\end{aligned}$$

where the inequality $(a)$ is due to $(a+b)^2 \leq (1+\beta)a^2 + (1+\frac{1}{\beta})b^2$, the inequality $(b)$ is because of $(1-\beta)^2 \leq 1$, $(1+\frac{1}{\beta}) \leq \frac{2}{\beta}$ and the Lemma 7 and the inequality $(c)$ is from Assumption 2. $\square$

**Lemma 10.** *Under Assumption 1, run Algorithm 1 with $\eta L_F \leq 1/4$, and then the output $\mathbf{x}_R$ of Algorithm 1 satisfies*

$$\mathbb{E}_R[\text{dist}(0, \hat{\partial}\bar{F}(\mathbf{x}_R))^2] \leq \frac{2 + 40L_F\eta}{T}\sum_{t=1}^{T}\|\mathbf{z}_t - \nabla F(\mathbf{x}_t)\|^2 + \frac{2\Delta}{\eta T} + \frac{40L_F\Delta}{T}. \tag{13}$$

*Proof.* The proof of this lemma follow the proof of Theorem 2 in (Xu et al., 2019).

Recall the update of $\mathbf{x}_{t+1}$ is

$$\begin{aligned}
\mathbf{x}_{t+1} &= \Pi_{\mathcal{X}}(\mathbf{x}_t - \eta\mathbf{z}_t)\\
&= \underset{\mathbf{x}\in\mathbb{R}^{d+1}}{\arg\min}\{\delta_{\mathcal{X}}(\mathbf{x}) + \langle\mathbf{z}_t, \mathbf{x} - \mathbf{x}_t\rangle + \frac{1}{2\eta}\|\mathbf{x} - \mathbf{x}_t\|^2\}.
\end{aligned}$$

then by Exercise 8.8 and Theorem 10.1 of (Rockafellar & Wets, 1998) we know

$$-\mathbf{z}_t - \frac{1}{\eta}(\mathbf{x}_{t+1} - \mathbf{x}_t) \in \hat{\partial}\delta_{\mathcal{X}}(\mathbf{x}_{t+1}),$$

which implies that

$$\nabla F(\mathbf{x}_{t+1}) - \mathbf{z}_t - \frac{1}{\eta}(\mathbf{x}_{t+1} - \mathbf{x}_t) \in \nabla F(\mathbf{x}_{t+1}) + \hat{\partial}\delta_{\mathcal{X}}(\mathbf{x}_{t+1}) = \hat{\partial}\bar{F}(\mathbf{x}_{t+1}). \tag{14}$$

By the update of $\mathbf{x}_{t+1}$, we also have,

$$\delta_{\mathcal{X}}(\mathbf{x}_{t+1}) + \langle \mathbf{z}_t, \mathbf{x}_{t+1} - \mathbf{x}_t \rangle + \frac{1}{2\eta}\|\mathbf{x}_{t+1} - \mathbf{x}_t\|^2 \le \delta_{\mathcal{X}}(\mathbf{x}_t).$$

Since $F(\mathbf{x})$ is smooth with parameter $L_F$, then

$$F(\mathbf{x}_{t+1}) \le F(\mathbf{x}_t) + \langle \nabla F(\mathbf{x}_t), \mathbf{x}_{t+1} - \mathbf{x}_t \rangle + \frac{L_F}{2}\|\mathbf{x}_{t+1} - \mathbf{x}_t\|^2.$$

Combing the above two inequalities, we get

$$\langle \mathbf{z}_t - \nabla F(\mathbf{x}_t), \mathbf{x}_{t+1} - \mathbf{x}_t \rangle + \frac{1}{2}(1/\eta - L)\|\mathbf{x}_{t+1} - \mathbf{x}_t\|^2 \le \bar{F}(\mathbf{x}_t) - \bar{F}(\mathbf{x}_{t+1}).$$

That is

$$\frac{1}{2}(1/\eta - L_F)\|\mathbf{x}_{t+1} - \mathbf{x}_t\|^2 \le \bar{F}(\mathbf{x}_t) - \bar{F}(\mathbf{x}_{t+1}) - \langle \mathbf{z}_t - \nabla F(\mathbf{x}_t), \mathbf{x}_{t+1} - \mathbf{x}_t \rangle$$

$$\le \bar{F}(\mathbf{x}_t) - \bar{F}(\mathbf{x}_{t+1}) + \eta\|\mathbf{z}_t - \nabla F(\mathbf{x}_t)\|^2 + \frac{1}{4\eta}\|\mathbf{x}_t - \mathbf{x}_{t+1}\|^2,$$

where the last inequality uses Young's inequality $\langle \mathbf{a}, \mathbf{b} \rangle \le \|\mathbf{a}\|^2 + \frac{\|\mathbf{b}\|^2}{4}$. Then by rearranging the above inequality and summing it across $t = 1, \cdots, T$, we have

$$\sum_{t=1}^{T} \frac{1 - 2\eta L_F}{4\eta}\|\mathbf{x}_{t+1} - \mathbf{x}_t\|^2 \le \bar{F}(\mathbf{x}_1) - \bar{F}(\mathbf{x}_{T+1}) + \sum_{t=1}^{T} \eta\|\mathbf{z}_t - \nabla F(\mathbf{x}_t)\|^2$$

$$\le \bar{F}(\mathbf{x}_1) - \inf_{\mathbf{x} \in \mathcal{X}} \bar{F}(\mathbf{x}) + \sum_{t=1}^{T} \eta\|\mathbf{z}_t - \nabla F(\mathbf{x}_t)\|^2$$

$$\le \Delta + \sum_{t=1}^{T} \eta\|\mathbf{z}_t - \nabla F(\mathbf{x}_t)\|^2. \tag{15}$$

By the same method used in the proof of Theorem 2 in Xu et al. (2019), we have the following inequality,

$$\sum_{t=1}^{T} \|\mathbf{z}_t - \nabla F(\mathbf{x}_{t+1}) + \frac{1}{\eta}(\mathbf{x}_{t+1} - \mathbf{x}_t)\|^2 \le 2\sum_{t=1}^{T} \|\mathbf{z}_t - \nabla F(\mathbf{x}_t)\|^2 + \frac{2\Delta}{\eta}$$

$$+ (2L_F^2 + \frac{3L_F}{\eta})\sum_{t=1}^{T} \|\mathbf{x}_{t+1} - \mathbf{x}_t\|^2. \tag{16}$$

Recalling $\eta L_F \le \frac{1}{4}$ and combining Eq. (15) and Eq. (16), we obtain

$$\sum_{t=1}^{T} \|\mathbf{z}_t - \nabla F(\mathbf{x}_{t+1}) + \frac{1}{\eta}(\mathbf{x}_{t+1} - \mathbf{x}_t)\|^2$$

$$\overset{(a)}{\le} 2\sum_{t=1}^{T} \|\mathbf{z}_t - \nabla F(\mathbf{x}_t)\|^2 + \frac{2\Delta}{\eta} + \frac{5L_F}{\eta}\left(\frac{1}{1/4 - \eta_1 L_F/2}\left(\eta_1\Delta + \eta_1\sum_{t=1}^{T} \eta_t\|\mathbf{z}_t - \nabla F(\mathbf{x}_t)\|^2\right)\right)$$

$$\overset{(b)}{\le} 2\sum_{t=1}^{T} \|\mathbf{z}_t - \nabla F(\mathbf{x}_t)\|^2 + \frac{2\Delta}{\eta} + 40L_F\Delta + 40\eta L_F\sum_{t=1}^{T} \|\mathbf{z}_t - \nabla F(\mathbf{x}_t)\|^2. \tag{17}$$

where inequality (a) is due to $(2L_F^2 + \frac{3L_F}{\eta}) \le \frac{5L_F}{\eta}$ and inequality (b) is due to $\frac{1}{1/4 - \eta L_F/2} \le 8$.

Recalling Eq. (14) and the output rule of Algorithm 1, we have

$$\mathbb{E}_R[\text{dist}(0, \hat{\partial}\bar{F}(\mathbf{x}_R))^2] \le \frac{1}{T}\sum_{t=1}^{T} \|\mathbf{z}_t - \nabla F(\mathbf{x}_{t+1}) + \frac{1}{\eta}(\mathbf{x}_{t+1} - \mathbf{x}_t)\|^2. \tag{18}$$

Then by combining Eqs. (17,18) together we have the Lemma. □

**Lemma 11.** *Under Assumption 1, 2, run Algorithm 1 with* $\eta \leq \frac{\beta}{4L_F\sqrt{4+20L_g^2}} \leq \frac{1}{4L_F}$*, and then we have*

$$\frac{1}{T}\sum_{t=1}^{T}\mathbb{E}[\|\mathbf{z}_t - \nabla F(\mathbf{x}_t)\|^2] \leq \frac{2\mathbb{E}[\|\mathbf{z}_1 - \nabla F(\mathbf{x}_1)\|^2]}{\beta T} + \frac{\Delta}{\eta T} + \frac{20L_F\mathbb{E}[\|g(\mathbf{x}_1) - s_1\|^2]}{\beta T} + 24\beta L_F^2\sigma^2.$$

*Proof.* To facilitate our proof statement, we define the following notations:

$$\nabla F(\mathbf{x}_t)^\top = (\nabla_\mathbf{w} F(\mathbf{x}_t)^\top, \nabla_\lambda F(\mathbf{x}_t)) = (\nabla f_{\lambda_t}(g(\mathbf{x}_t))\nabla_\mathbf{w} g(\mathbf{x}_t)^\top, \nabla f_{\lambda_t}(g(\mathbf{x}_t))\nabla_\lambda g(\mathbf{x}_t) + \log(g(\mathbf{x}_t)) + \rho)$$

$$\widetilde{\nabla} F(\mathbf{x}_t)^\top = (\nabla f_{\lambda_t}(g(\mathbf{x}_t))\nabla_\mathbf{w} g_i(\mathbf{x}_t)^\top, \nabla f_{\lambda_t}(g(\mathbf{x}_t))\nabla_\lambda g_i(\mathbf{x}_t) + \log(g(\mathbf{x}_t)) + \rho)$$

$$G(\mathbf{x}_t)^\top = (G_{\mathbf{w}_t}(\mathbf{x}_t)^\top, G_{\lambda_t}(\mathbf{x}_t)) = (\nabla f_{\lambda_t}(s_t)\nabla_\mathbf{w} g_i(\mathbf{x}_t)^\top, \nabla f_{\lambda_t}(s_t)\nabla_\lambda g_i(\mathbf{x}_t) + \log(s_t) + \rho).$$

It is worth to notice that $\mathbb{E}[\widetilde{\nabla} F(\mathbf{x}_t)] = \nabla F(\mathbf{x}_t)$.

For every iteration $t$, by simple expansion we have

$$I_t = \mathbb{E}[\|\nabla F(\mathbf{x}_t) - \mathbf{z}_t\|^2]$$
$$= \mathbb{E}[\|\nabla F(\mathbf{x}_t) - (1-\beta)\mathbf{z}_{t-1} - \beta G(\mathbf{x}_t)\|^2]$$
$$= \mathbb{E}[\|(1-\beta)(\nabla F(\mathbf{x}_t) - \nabla F(\mathbf{x}_{t-1})) + (1-\beta)\nabla F(\mathbf{x}_{t-1}) - (1-\beta)\mathbf{z}_{t-1} + \beta\nabla F(\mathbf{x}_t) - \beta G(\mathbf{x}_t)\|^2]$$
$$= \mathbb{E}[\|(1-\beta)\underbrace{(\nabla F(\mathbf{x}_t) - \nabla F(\mathbf{x}_{t-1}))}_{A} + (1-\beta)\underbrace{(\nabla F(\mathbf{x}_{t-1}) - \mathbf{z}_{t-1})}_{B}\|^2]$$
$$\quad + \mathbb{E}[\|\beta\underbrace{(\widetilde{\nabla} F(\mathbf{x}_t) - G(\mathbf{x}_t))}_{C} + \beta\underbrace{(\nabla F(\mathbf{x}_t) - \widetilde{\nabla} F(\mathbf{x}_t))}_{D}\|^2]$$
$$= \mathbb{E}[(1-\beta)^2\|A\|^2 + (1-\beta)^2\|B\|^2 + \beta^2\|C\|^2 + \beta^2\|D\|^2 + 2(1-\beta)(1-\beta)\langle A, B\rangle$$
$$\quad + 2\beta(1-\beta)\langle A, C\rangle + 2\beta(1-\beta)\langle A, D\rangle + 2(1-\beta)\beta\langle B, C\rangle + 2(1-\beta)\beta\langle B, D\rangle + 2\beta^2\langle C, D\rangle]$$
$$\overset{(a)}{=} \mathbb{E}[(1-\beta)^2\|A\|^2 + (1-\beta)^2\|B\|^2 + \beta^2\|C\|^2 + \beta^2\|D\|^2$$
$$\quad + 2(1-\beta)^2\langle A, B\rangle + 2(1-\beta)\beta\langle C, B\rangle + 2\beta(1-\beta)\langle A, C\rangle + 2\beta^2\langle C, D\rangle],$$

where the equality $(a)$ is due to $\mathbb{E}\langle\nabla F(\mathbf{x}_t) - \nabla F(\mathbf{x}_{t-1}), \nabla F(\mathbf{x}_t) - \widetilde{\nabla} F(\mathbf{x}_t)\rangle = 0$ and $\mathbb{E}\langle\mathbf{z}_{t-1} - \nabla F(\mathbf{x}_{t-1}), \nabla F(\mathbf{x}_t) - \widetilde{\nabla} F(\mathbf{x}_t)\rangle = 0$.

By Young's inequality, we have $(1-\beta)^2\langle A, B\rangle \leq (1-\beta)\langle A, B\rangle \leq \frac{2}{\beta}\|A\|^2 + \frac{(1-\beta)^2\beta}{8}\|B\|^2$, $2\beta(1-\beta)\langle C, B\rangle \leq \frac{(1-\beta)^2\beta}{2}\|B\|^2 + 2\beta\|C\|^2$, $2\beta(1-\beta)\langle A, C\rangle \leq (1-\beta)^2\|A\|^2 + \beta^2\|C\|^2$ and $2\beta^2\langle C, D\rangle \leq \beta^2\|C\|^2 + \beta^2\|D\|^2$. Therefore, noting $(1-\beta) < 1$ and $1/\beta > 1$, we can obtain

$$I_t \leq \mathbb{E}[(1-\beta)^2\|A\|^2 + (1-\beta)^2\|B\|^2 + \beta^2\|C\|^2 + \beta^2\|D\|^2$$
$$\quad + \frac{2}{\beta}\|A\|^2 + \frac{(1-\beta)^2\beta}{2}\|B\|^2 + 2\beta\|C\|^2 + \frac{(1-\beta)^2\beta}{2}\|B\|^2$$
$$\quad + (1-\beta)^2\|A\|^2 + \beta^2\|C\|^2 + \beta^2\|C\|^2 + \beta^2\|D\|^2]$$
$$\leq \mathbb{E}[(1-\beta)\|B\|^2 + \frac{4}{\beta}\|A\|^2 + 5\beta\|C\|^2 + 2\beta^2\|D\|^2]. \tag{19}$$

Thus recalling the defintion of $G(\mathbf{x}_t), \widetilde{\nabla} F(\mathbf{x}_t), \nabla F(\mathbf{x}_t)$ and applying the smoothness and Lipschitz continuity of $f_\lambda$ and $g$, we have

$$
\begin{aligned}
C &= \|\widetilde{\nabla} F(\mathbf{x}_t) - G(\mathbf{x}_t)\|^2 \\
&= \|\nabla f_{\lambda_t}(g(\mathbf{x}_t)) \nabla_\mathbf{w} g_i(\mathbf{x}_t) - \nabla f_{\lambda_t}(s_t) \nabla_{\mathbf{w}_t} g_i(\mathbf{x}_t)\|^2 \\
&\quad + \|\nabla f_{\lambda_t}(g(\mathbf{x}_t)) \nabla_\lambda g_i(\mathbf{x}_t) + \log(g(\mathbf{x}_t)) - \nabla f_{\lambda_t}(s_t) \nabla_\lambda g_i(\mathbf{x}_t) - \log(s_t)\|^2 \\
&\leq \|\nabla f_{\lambda_t}(g(\mathbf{x}_t)) \nabla_\mathbf{w} g_i(\mathbf{x}_t) - \nabla f_{\lambda_t}(s_t) \nabla_{\mathbf{w}_t} g_i(\mathbf{x}_t)\|^2 + 2\|\nabla f_{\lambda_t}(g(\mathbf{x}_t)) \nabla_\lambda g_i(\mathbf{x}_t) - \nabla f_{\lambda_t}(s_t) \nabla_\lambda g_i(\mathbf{x}_t)\|^2 \\
&\quad + 2\|\log(g(\mathbf{x}_t)) - \log(s_t)\|^2 \\
&\overset{(a)}{\leq} 2 L_g^2 L_{\nabla f_{\lambda_t}}^2 \|s_t - g(\mathbf{x}_t)\|^2 + 2\|s_t - g(\mathbf{x}_t)\|^2 \\
&\overset{(b)}{\leq} 2 L_F^2 \|s_t - g(\mathbf{x}_t)\|^2,
\end{aligned}
\tag{20}
$$

where the inequality $(a)$ is due to $|\log(g(\mathbf{x}_t)) - \log(s_t)| \leq |s_t - g(\mathbf{x}_t)|$ since $g(\mathbf{x}_t) \geq 1, s_t \geq 1$ for all $t = \{1, \cdots, T\}$ by the definition and initialzation of $g_i(\mathbf{x}_t), s_t$, and the inequality $(b)$ is due to $L_g^2 L_{\nabla f_{\lambda_t}}^2 + 1 \leq L_F^2$.

And by the similar method, we also have

$$
\begin{aligned}
D &= \|\nabla F(\mathbf{x}_t) - \widetilde{\nabla} F(\mathbf{x}_t)\|^2 \\
&= \|\nabla f_{\lambda_t}(g(\mathbf{x}_t)) \nabla_\mathbf{w} g(\mathbf{x}_t) - \nabla f_{\lambda_t}(g(\mathbf{x}_t)) \nabla_\mathbf{w} g_i(\mathbf{x}_t)\|^2 \\
&\quad + \|\nabla f_{\lambda_t}(g(\mathbf{x}_t)) \nabla_\lambda g(\mathbf{x}_t) + \log(g(\mathbf{x}_t)) + \rho - \nabla f_{\lambda_t}(g(\mathbf{x}_t)) \nabla_\lambda g_i(\mathbf{x}_t) - \log(g(\mathbf{x}_t)) - \rho\|^2 \\
&= \|\nabla f_{\lambda_t}(g(\mathbf{x}_t)) \nabla_\mathbf{w} g(\mathbf{x}_t) - \nabla f_{\lambda_t}(g(\mathbf{x}_t)) \nabla_\mathbf{w} g_i(\mathbf{x}_t)\|^2 \\
&\quad + \|\nabla f_{\lambda_t}(g(\mathbf{x}_t)) \nabla_\lambda g(\mathbf{x}_t) - \nabla f_{\lambda_t}(g(\mathbf{x}_t)) \nabla_\lambda g_i(\mathbf{x}_t)\|^2 \\
&\leq L_{f_{\lambda_t}}^2 \|\nabla g(\mathbf{x}_t) - \nabla g_i(\mathbf{x}_t)\|^2 \leq L_F^2 \|\nabla g(\mathbf{x}_t) - \nabla g_i(\mathbf{x}_t)\|^2.
\end{aligned}
\tag{21}
$$

Thus combining the Eqs. (19, 20, 21) and applying Assumption 2, we can obtain

$$
\begin{aligned}
&\mathbb{E}[\|\mathbf{z}_t - \nabla F(\mathbf{x}_t)\|^2] \\
&= \mathbb{E}\Big[(1-\beta)\|\mathbf{z}_{t-1} - \nabla F(\mathbf{x}_{t-1})\|^2 + \frac{4}{\beta}\|\nabla F(\mathbf{x}_t) - \nabla F(\mathbf{x}_{t-1})\|^2 \\
&\quad + 5\beta\|\widetilde{\nabla} F(\mathbf{x}_t) - G(\mathbf{x}_t)\|^2 + 2\beta^2\|\nabla F(\mathbf{x}_t) - \widetilde{\nabla} F(\mathbf{x}_t)\|^2\Big] \\
&\leq \mathbb{E}\Big[(1-\beta)\|\mathbf{z}_{t-1} - \nabla F(\mathbf{x}_{t-1})\|^2 + \frac{4}{\beta} L_F^2 \|\mathbf{x}_t - \mathbf{x}_{t-1}\|^2 + 10 L_F^2 \beta \|g(\mathbf{x}_t) - s_t\|^2\Big] + 2\beta^2 L_F^2 \sigma^2.
\end{aligned}
$$

Taking summation of $\mathbb{E}[\|\mathbf{z}_{t+1} - \nabla F(\mathbf{x}_{t+1})\|^2]$ from 1 to $T$ and invoking Lemma 9, we have

$$
\begin{aligned}
&\sum_{t=1}^T \mathbb{E}[\|\mathbf{z}_t - \nabla F(\mathbf{x}_t)\|^2] \\
&\leq \frac{\mathbb{E}[\|\nabla F(\mathbf{x}_1) - \mathbf{z}_1\|^2]}{\beta} + \frac{4 L_F^2}{\beta^2} \sum_{t=1}^T \mathbb{E}[\|\mathbf{x}_{t+1} - \mathbf{x}_t\|^2] + 10 L_F^2 \beta \sum_{t=1}^T \mathbb{E}[\|g(\mathbf{x}_t) - s_t\|^2] + 2\beta^2 L_F \sigma^2 \\
&\leq \frac{\mathbb{E}[\|\nabla F(\mathbf{x}_1) - \mathbf{z}_1\|^2]}{\beta} + \frac{4 L_F^2}{\beta^2} \sum_{t=1}^T \mathbb{E}[\|\mathbf{x}_{t+1} - \mathbf{x}_t\|^2] \\
&\quad + 10 L_F^2 \left(\mathbb{E}\left[\frac{\|g(\mathbf{x}_1) - s_1\|^2}{\beta} + \frac{2 L_g^2}{\beta^2} \sum_{t=1}^T \|\mathbf{x}_{t+1} - \mathbf{x}_t\|^2\right] + \beta T \sigma^2\right) + 2\beta L_F^2 T \sigma^2.
\end{aligned}
$$

Taking Eq. (15) into the above inequality, we have

$$\sum_{t=1}^{T} \mathbb{E}[\|\mathbf{z}_t - \nabla F(\mathbf{x}_t)\|^2]$$

$$\leq \frac{\mathbb{E}[\|\nabla F(\mathbf{x}_1) - \mathbf{z}_1\|^2]}{\beta} + (\frac{4L_F^2}{\beta^2} + \frac{20L_F^2 L_g^2}{\beta^2})\left(\frac{\eta}{1/4 - \eta L_F/2}\left(\Delta + \eta \sum_{t=1}^{T} \mathbb{E}[\|\mathbf{z}_t - \nabla F(\mathbf{x}_t)\|^2]\right)\right)$$

$$+ 10L_F^2\left(\frac{\mathbb{E}[\|g(\mathbf{x}_1) - s_1\|^2]}{\beta} + \beta T\sigma^2\right) + 2\beta L_F^2 T\sigma^2$$

$$\overset{(a)}{\leq} \frac{\mathbb{E}[\|\nabla F(\mathbf{x}_1) - \mathbf{z}_1\|^2]}{\beta} + (\frac{4L_F^2}{\beta^2} + \frac{20L_F^2 L_g^2}{\beta^2})\left(8\eta\left(\Delta + \eta \sum_{t=1}^{T} \mathbb{E}[\|\mathbf{z}_t - \nabla F(\mathbf{x}_t)\|^2]\right)\right)$$

$$+ 10L_F^2\left(\frac{\mathbb{E}[\|g(\mathbf{x}_1) - s_1\|^2]}{\beta} + \beta T\sigma^2\right) + 2\beta L_F^2 T\sigma^2$$

$$\overset{(b)}{\leq} \frac{\mathbb{E}[\|\mathbf{z}_1 - \nabla F(\mathbf{x}_1)\|^2]}{\beta} + \frac{\Delta}{2\eta} + \frac{1}{2}\sum_{t=1}^{T} \mathbb{E}[\|\mathbf{z}_t - \nabla F(\mathbf{x}_t)\|^2]$$

$$+ 10L_F^2\left(\frac{\mathbb{E}[\|g(\mathbf{x}_1) - s_1\|^2]}{\beta} + \beta T\sigma^2\right) + 2\beta L_F^2 T\sigma^2, \tag{22}$$

where the inequality (a) is due to $\eta L_F \leq 1/4$ and the inequality (b) is due to $8(4L_F^2 + 20L_F^2 L_g^2)\eta^2 \leq \frac{\beta^2}{2}$.

Rearranging terms and dividing $T$ on both sides of Eq. (22), we compelte the proof. $\square$

## B.2 Proof of Theorem 1

*Proof.* Since $\eta = \frac{\beta}{20L_F^2}$, $L_F \geq 1$ and $L_F \leq L_g$, it holds that $\eta \leq \frac{\beta}{4L_F\sqrt{4+20L_g^2}} \leq \frac{1}{4L_F}$ which satisfy the assumptions of $\eta$ in Lemma 10 and Lemma 11. Therefore, combining Lemma 10 and Lemma 11, we have

$$\mathbb{E}[\text{dist}(0, \hat{\partial}\bar{F}(\mathbf{x}_R))^2]$$

$$\leq \frac{2 + 40L_F\eta}{T}\sum_{t=1}^{T} \mathbb{E}[\|\mathbf{z}_t - \nabla F(\mathbf{x}_t)\|^2] + \frac{2\Delta}{\eta T} + \frac{40L_F\Delta}{T}$$

$$\leq \frac{12}{T}\sum_{t=1}^{T} \mathbb{E}[\|\mathbf{z}_t - \nabla F(\mathbf{x}_t)\|^2] + \frac{2\Delta}{\eta T} + \frac{20L_F\Delta}{T}$$

$$\leq \frac{24\mathbb{E}[\|\mathbf{z}_1 - \nabla F(\mathbf{x}_1)\|^2]}{\beta T} + \frac{12\Delta}{\eta T} + \frac{240L_F^2\mathbb{E}[\|g(\mathbf{x}_1) - s_1\|^2]}{\beta T} + 288L_F^2\beta\sigma^2 + \frac{2\Delta}{\eta T} + \frac{20L_F^2\Delta}{T}. \tag{23}$$

By the definition of $s_1$ and Assumption 2, it holds that

$$\mathbb{E}[\|s_1 - g(\mathbf{x}_1)\|^2] \leq \mathbb{E}[\|g_i(\mathbf{x}_1) - g(\mathbf{x}_1)\|^2] \leq \sigma^2. \tag{24}$$

Since $L_g^2 L_{\nabla f_{\lambda_1}}^2 \leq L_F^2$ and $2L_{f_{\lambda_1}}^2 \leq L_F^2$, we have

$$\mathbb{E}[\|\mathbf{z}_1 - \nabla F(\mathbf{x}_1)\|^2]$$

$$= \|\nabla f_{\lambda_1}(g_i(\mathbf{x}_1))\nabla g_i(\mathbf{x}_1) - \nabla f_{\lambda_1}(g(\mathbf{x}_1))\nabla g(\mathbf{x}_1)\|^2$$

$$= \|\nabla f_{\lambda_1}(g_i(\mathbf{x}_1)\nabla g_i(\mathbf{x}_1)) - \nabla f_{\lambda_1}(g(\mathbf{x}_1))\nabla g_i(\mathbf{x}_1) + \nabla f_{\lambda_1}(g(\mathbf{x}_1))\nabla g_i(\mathbf{x}_1) - \nabla f_{\lambda_1}(g(\mathbf{x}_1)\nabla g(\mathbf{x}_1))\|^2$$

$$\overset{(a)}{\leq} 2\|\nabla f_{\lambda_1}(g_i(\mathbf{x}_1)) - \nabla f_{\lambda_1}(g(\mathbf{x}_1))\|^2\|\nabla g_i(\mathbf{x}_1)\|^2 + 2\|\nabla f_{\lambda_1}(g_i(\mathbf{x}_1))\|^2\|\nabla g_i(\mathbf{x}_1) - \nabla g(\mathbf{x}_1)\|^2$$

$$\leq (2L_g^2 L_{\nabla f_{\lambda_1}}^2 + 2L_{f_{\lambda_1}}^2)\sigma^2 \leq 4L_F^2\sigma^2, \tag{25}$$

where the inequality $(a)$ is due to $\|\mathbf{a} + \mathbf{b}\|^2 \leq 2\|\mathbf{a}\|^2 + 2\|\mathbf{b}\|^2$.

Combining Eqs. (23,24,25), we obtain

$$\mathbb{E}[\text{dist}(0, \hat{\partial}\bar{F}(\mathbf{x}_R))^2]$$

$$\leq \frac{24\mathbb{E}[\|\mathbf{z}_1 - \nabla F(\mathbf{x}_1)\|^2]}{\beta T} + \frac{12\Delta}{\eta T} + \frac{240 L_F^2 \mathbb{E}[\|g(\mathbf{x}_1) - s_1\|^2]}{\beta T} + 288 L_F^2 \beta \sigma^2 + \frac{2\Delta}{\eta T} + \frac{20 L_F^2 \Delta}{T}$$

$$\leq \frac{96 L_F^2 \sigma^2}{\beta T} + \frac{12\Delta}{\eta T} + \frac{240 L_F^2 \sigma^2}{\beta T} + 288 L_F^2 \beta \sigma^2 + \frac{2\Delta}{\eta T} + \frac{20 L_F^2 \Delta}{T}$$

$$\leq \frac{96 L_F^2 \sigma^2}{\sqrt{T}} + \frac{240\Delta L_F^2}{\sqrt{T}} + \frac{528 L_F^2 \sigma^2}{\sqrt{T}} + \frac{40\Delta L_F^2}{\sqrt{T}} + \frac{20 L_F^2 \Delta}{T}$$

$$\leq (624\sigma^2 + 280\Delta)\frac{L_F^2}{\sqrt{T}} + \frac{20 L_F^2 \Delta}{T}.$$

This complete the proof. $\qquad \square$

# C   Proofs in Section 4.2

## C.1   Technical Lemmas

**Lemma 12.** *Let $\mathbf{z}_t = \nabla f_{\lambda_t}(s_t)\mathbf{q}_t + \mathbf{q}_{\lambda_t}$, where $\mathbf{q}_t = (\mathbf{v}_t^\top, u_t)^\top$, $\mathbf{q}_{\lambda_t} = (\mathbf{0}^\top, \log(s_t) + \rho)^\top$ and $\mathbf{0} \in \mathbb{R}^d$. Let $\|\varkappa_t\|^2 = \|s_t - g(\mathbf{x}_t)\|^2 + \|\mathbf{v}_t - \nabla_\mathbf{w} g(\mathbf{x}_t)\|^2 + |u_t - \nabla_\lambda g(\mathbf{x}_t)|^2$. Under Assumption 1, run Algorithm 2, and then for every $t \in \{1, \cdots T\}$ we have*

$$\|\mathbf{z}_t - \nabla F(\mathbf{x}_t)\|^2 \leq 4 L_F^2 \|\varkappa_t\|^2.$$

*Proof.* By simple expansion, it holds that

$$\|\mathbf{z}_t - \nabla F(\mathbf{x}_t)\|^2$$

$$= \|\nabla f_{\lambda_t}(g(\mathbf{x}_t))\nabla_\mathbf{w} g(\mathbf{x}_t) - \nabla f_{\lambda_t}(s_t)\mathbf{v}_t\|^2$$

$$\quad + \|\nabla f_{\lambda_t}(g(\mathbf{x}_t))\nabla_\lambda g(\mathbf{x}_t) - \nabla f_{\lambda_t}(s_t)\mathbf{v}_t + \log(g(\mathbf{x}_t)) - \log(s_t)\|^2$$

$$\overset{(a)}{\leq} 2\|\nabla f_{\lambda_t}(g(\mathbf{x}_t))\nabla_\mathbf{w} g(\mathbf{x}_t) - \nabla f_{\lambda_t}(s_t)\mathbf{v}_t\|^2 + 2\|\nabla f_{\lambda_t}(g(\mathbf{x}_t))\nabla_\lambda g(\mathbf{x}_t) - \nabla f_{\lambda_t}(s_t)u_t\|^2$$

$$\quad + 2\|g(\mathbf{x}_t) - s_t\|^2$$

$$= 2\|\nabla f_{\lambda_t}(g(\mathbf{x}_t))\nabla g(\mathbf{x}_t) - \nabla f_{\lambda_t}(s_t)\mathbf{q}_t\|^2 + 2\|g(\mathbf{x}_t) - s_t\|^2, \qquad (26)$$

where the inequality $(a)$ is because $\|\mathbf{a} + \mathbf{b}\|^2 \leq 2\|\mathbf{a}\|^2 + 2\|\mathbf{b}\|^2$, and $|\log(x) - \log(y)| \leq |x - y|$ for all $x, y \geq 1$.

Applying the smoothness and Lipschitz continuity of $f_\lambda$ and $g$, we obtain

$$\|\nabla f_{\lambda_t}(g(\mathbf{x}_t))\nabla g(\mathbf{x}_t) - \nabla f_{\lambda_t}(s_t)\mathbf{q}_t\|^2$$

$$= \|\nabla f_{\lambda_t}(g(\mathbf{x}_t))\nabla g(\mathbf{x}_t) - \nabla f_{\lambda_t}(s_t)\nabla g(\mathbf{x}_t) + \nabla f_{\lambda_t}(s_t)\nabla g(\mathbf{x}_t) - \nabla f_{\lambda_t}(s_t)\mathbf{q}_t\|^2$$

$$\leq 2\|\nabla f_{\lambda_t}(g(\mathbf{x}_t))\nabla g(\mathbf{x}_t) - \nabla f_{\lambda_t}(s_t)\nabla g(\mathbf{x}_t)\|^2 + 2\|\nabla f_{\lambda_t}(s_t)\nabla g(\mathbf{x}_t) - \nabla f_{\lambda_t}(s_t)\mathbf{q}_t\|^2$$

$$\leq 2 L_g^2 L_{\nabla f_{\lambda_t}}^2 \|s_t - g(\mathbf{x}_t)\|^2 + 2 L_{f_{\lambda_t}}^2 \|\mathbf{q}_t - \nabla g(\mathbf{x}_t)\|^2 + 2\|g(\mathbf{x}_t) - s_t\|^2. \qquad (27)$$

Noting $\|\mathbf{q}_t - \nabla g(\mathbf{x}_t)\|^2 = \|\mathbf{v}_t - \nabla_\mathbf{w} g(\mathbf{x}_t)\|^2 + |u_t - \nabla_\lambda g(\mathbf{x}_t)|^2$ and combining Eqs. (26, 27), we have

$$\|\mathbf{z}_t - \nabla F(\mathbf{x}_t)\|^2$$

$$\leq (4 L_g^2 L_{\nabla f_{\lambda_t}}^2 + 2)\|s_t - g(\mathbf{x}_t)\|^2 + 4 L_{f_{\lambda_t}}^2 \|\mathbf{q}_t - \nabla g(\mathbf{x}_t)\|^2$$

$$\leq 4 L_F^2 \|s_t - g(\mathbf{x}_t)\|^2 + 4 L_F^2 \|\mathbf{q}_t - \nabla g(\mathbf{x}_t)\|^2$$

$$= 4 L_F^2 (\|s_t - g(\mathbf{x}_t)\|^2 + \|\mathbf{v}_t - \nabla_\mathbf{w} g(\mathbf{x}_t)\|^2 + |u_t - \nabla_\lambda g(\mathbf{x}_t)|^2).$$

This complete the proof. $\qquad \square$

**Lemma 13.** *Under Assumption 1, 2, run Algorithm 2, and then for every $t \in \{1, \cdots T\}$ we have*

$$\mathbb{E}[\|\varkappa_{t+1}\|^2] \leq (1 - \beta_t)^2 \mathbb{E}[\|\varkappa_t\|^2] + 8(1 - \beta_t)^2 L_F^2 \mathbb{E}[\|\mathbf{x}_{t+1} - \mathbf{x}_t\|^2] + 6\beta_t^2 \sigma^2.$$

*Proof.* Since $s_{t+1} = (g_i(\mathbf{x}_{t+1}) + (1 - \beta)(s_t - g_i(\mathbf{x}_t))$, it holds that

$$
\begin{aligned}
&\mathbb{E}[\|s_{t+1} - g(\mathbf{x}_{t+1})\|^2] \\
&= \mathbb{E}[\|g_i(\mathbf{x}_{t+1}) + (1 - \beta_t)(s_t - g_i(\mathbf{x}_t)) - g(\mathbf{x}_{t+1})\|^2] \\
&\leq \mathbb{E}[\|(1 - \beta_t)(s_t - g(\mathbf{x}_t)) + \beta_t(g_i(\mathbf{x}_{t+1}) - g(\mathbf{x}_{t+1})) \\
&\quad + (1 - \beta_t)(g_i(\mathbf{x}_{t+1}) - g_i(\mathbf{x}_t) - (g(\mathbf{x}_{t+1}) - g(\mathbf{x}_t)))\|^2] \\
&= \mathbb{E}[(1 - \beta_t)^2 \|s_t - g(\mathbf{x}_t)\|^2] + \mathbb{E}[\|\beta_t(g_i(\mathbf{x}_{t+1}) - g(\mathbf{x}_{t+1})) \\
&\quad + (1 - \beta_t)(g_i(\mathbf{x}_{t+1}) - g_i(\mathbf{x}_t) - (g(\mathbf{x}_{t+1}) - g(\mathbf{x}_t)))\|^2],
\end{aligned}
\tag{28}
$$

where the last inequality is due to $\mathbb{E}[g_i(\mathbf{x}_{t+1}) - g(\mathbf{x}_{t+1})] = 0$.

Noting $\mathbb{E}[\langle g_i(\mathbf{x}_{t+1}) - g_i(\mathbf{x}_{t+1}), g(\mathbf{x}_{t+1}) - g(\mathbf{x}_t)\rangle] = \mathbb{E}[\|(g(\mathbf{x}_{t+1}) - g(\mathbf{x}_t))\|^2]$ and applying the Lipschitz continuty of $g_i(\mathbf{x})$, we have

$$
\begin{aligned}
&\mathbb{E}[\|g_i(\mathbf{x}_{t+1}) - g_i(\mathbf{x}_{t+1}) - (g(\mathbf{x}_{t+1}) - g(\mathbf{x}_t))\|^2] \\
&= \mathbb{E}[\|(g_i(\mathbf{x}_{t+1}) - g_i(\mathbf{x}_{t+1})\|^2 + \|(g(\mathbf{x}_{t+1}) - g(\mathbf{x}_t))\|^2 - 2\langle g_i(\mathbf{x}_{t+1}) - g_i(\mathbf{x}_{t+1}), g(\mathbf{x}_{t+1}) - g(\mathbf{x}_t)\rangle] \\
&= \mathbb{E}[\|(g_i(\mathbf{x}_{t+1}) - g_i(\mathbf{x}_{t+1})\|^2 - \|(g(\mathbf{x}_{t+1}) - g(\mathbf{x}_t))\|^2] \\
&\leq \mathbb{E}[\|(g_i(\mathbf{x}_{t+1}) - g_i(\mathbf{x}_{t+1})\|^2] \\
&\leq L_g^2 \mathbb{E}[\|\mathbf{x}_{t+1} - \mathbf{x}_t\|^2].
\end{aligned}
\tag{29}
$$

Combining Eqs. (28, 29) and invoking the Lipschitz continuty of $g_i(\mathbf{x})$, under Assumption 2, we have

$$
\begin{aligned}
&\mathbb{E}[\|s_{t+1} - g(\mathbf{x}_{t+1})\|^2] \\
&\leq (1 - \beta_t)^2 \mathbb{E}[\|s_t - g(\mathbf{x}_t)\|^2] \\
&\quad + 2\beta_t^2 \mathbb{E}[\|g_i(\mathbf{x}_{t+1}) - g(\mathbf{x}_t)\|^2] + 2(1 - \beta_t)^2 \mathbb{E}[\|g_i(\mathbf{x}_{t+1}) - g_i(\mathbf{x}_{t+1}) - (g(\mathbf{x}_{t+1}) - g(\mathbf{x}_t))\|^2] \\
&\leq (1 - \beta_t)^2 \mathbb{E}[\|s_t - g(\mathbf{x}_t)\|^2] + 2\beta_t^2 \sigma^2 + 2(1 - \beta_t)^2 L_g^2 \mathbb{E}[\|\mathbf{x}_{t+1} - \mathbf{x}_t\|^2].
\end{aligned}
\tag{30}
$$

In the same way, we also have

$$
\mathbb{E}[\|\mathbf{v}_{t+1} - \nabla_{\mathbf{w}} g(\mathbf{x}_{t+1})\|^2] \leq (1 - \beta_t)^2 \mathbb{E}[\|\mathbf{v}_t - \nabla_{\mathbf{w}} g(\mathbf{x}_t)\|^2] + 2\beta_t^2 \sigma^2 + 2(1 - \beta_t)^2 L_{\nabla g}^2 \mathbb{E}[\|\mathbf{x}_{t+1} - \mathbf{x}_t\|^2],
\tag{31}
$$

$$
\mathbb{E}[|u_{t+1} - \nabla_\lambda g(\mathbf{x}_{t+1})|^2] \leq (1 - \beta_t)^2 \mathbb{E}[|u_t - \nabla_\lambda g(\mathbf{x}_t)|^2] + 2\beta_t^2 \sigma^2 + 2(1 - \beta_t)^2 L_{\nabla g}^2 \mathbb{E}[\|\mathbf{x}_{t+1} - \mathbf{x}_t\|^2].
\tag{32}
$$

Therefore, combining Eqs. (30, 32, 31), we obtain

$$
\begin{aligned}
\mathbb{E}[\|\varkappa_{t+1}\|^2] &\leq (1 - \beta_t)^2 \mathbb{E}[\|\varkappa_t\|^2] + 6\beta_t^2 \sigma^2 + 4(1 - \beta_t)^2 (L_{\nabla g}^2 + L_g^2)\|\mathbf{x}_{t+1} - \mathbf{x}_t\|^2) \\
&\leq (1 - \beta_t)^2 \mathbb{E}[\|\varkappa_t\|^2] + 8(1 - \beta_t)^2 L_F^2 \mathbb{E}[\|\mathbf{x}_{t+1} - \mathbf{x}_t\|^2] + 6\beta_t^2 \sigma^2,
\end{aligned}
$$

where the last inequality applies $(L_{\nabla g}^2 + L_g^2) \leq 2L_F^2$. This complete the proof. $\square$

**Lemma 14.** *Under Assumption 1 and 2, for any $\alpha > 1$, let $k = \frac{\alpha\sigma^{2/3}}{L_F}$, $w = \max(2\sigma^2, (16L_F^2 k)^3)$ and $c = \frac{\sigma^2}{14 L_F k^3} + 130 L_F^4$. Then with $\eta_t = \frac{k}{(w+t\sigma^2)^{1/3}}$, $\beta_t = c\eta_t^2$ and after running $T$ iterations, Algrithm 2 satisfies*

$$
4L_F^4 \sum_{t=1}^T \eta_t \mathbb{E}[\|\varkappa_t\|^2] \leq \frac{\mathbb{E}[\|\varkappa_1\|^2]}{\eta_0} - \frac{\mathbb{E}[\|\varkappa_{T+1}\|^2]}{\eta_T} + \sum_{t=1}^T 6c^2 \eta_t^3 \sigma^2 + 64 L_F^2 \Delta.
$$

*Proof.* Since $w \geq (16L_F^2 k)^3$, it is easy to note that

$$
\eta_t \leq \eta_0 \leq \frac{1}{16 L_F^2} \leq \frac{1}{4 L_F}.
$$

In addition,

$$
\begin{aligned}
\beta_t = c\eta_t^2 &\leq c\eta_0^2 \leq (\frac{\sigma^2}{14 L_F k^3} + 130 L_F^4)\frac{1}{256 L_F^4} \\
&= \frac{\sigma^2 L_F^3}{14 L_F \alpha^3 \sigma^2} \frac{1}{256 L_F^4} + \frac{65}{128} = \frac{1}{14\alpha^3} \frac{1}{2556 L_F^2} + \frac{65}{128} \leq 1.
\end{aligned}
$$

With $\eta_t = \frac{k}{(w+t\sigma^2)^{1/3}}$, we obtain

$$\frac{1}{\eta_t} - \frac{1}{\eta_{t-1}} = \frac{(w+t\sigma^2)^{1/3} - (w+(t-1)\sigma^2)^{1/3}}{k} \overset{(a)}{\leq} \frac{\sigma^2}{3k(w+(t-1)\sigma^2)^{2/3}}$$

$$\overset{(b)}{\leq} \frac{\sigma^2}{3k(w/2+t\sigma^2)^{2/3}} \leq \frac{\sigma^2}{3k(w/2+t\sigma^2/2)^{2/3}} = \frac{2^{2/3}\sigma^2}{3k(w+t\sigma^2)^{2/3}}$$

$$= \frac{2^{2/3}\sigma^2}{3k^3}\eta_t^2 \overset{(c)}{\leq} \frac{2^{2/3}}{12L_F k^3}\eta_t \leq \frac{\sigma^2}{7Lk^3}\eta_t,$$

where the inequality (a) uses the inequality $(x+y)^{1/3} - x^{1/3} \leq \frac{yx^{-2/3}}{3}$, the inequality (b) is due to $w \geq 2\sigma^2$, and the inequality (c) is due to $\eta_t \leq \frac{1}{4L_F}$.

Noting $\beta_t = c\eta_t^2$ and $0 \leq (1-\beta_t) \leq 1$, by Lemma 13 we have

$$\frac{\mathbb{E}[\|\varkappa_{t+1}\|^2]}{\eta_t} - \frac{\mathbb{E}[\|\varkappa_t\|^2]}{\eta_{t-1}}$$

$$\leq (\frac{(1-\beta_t)^2}{\eta_t} - \frac{1}{\eta_{t-1}})\mathbb{E}[\|\varkappa_t\|^2] + 6c^2\eta_t^3\sigma^2 + \frac{8(1-\beta_t)^2 L_F^2}{\eta_t}\mathbb{E}[\|\mathbf{x}_{t+1} - \mathbf{x}_t\|^2]$$

$$\leq (\eta_t^{-1} - \eta_{t-1}^{-1} - 2c\eta_t)\mathbb{E}[\|\varkappa_t\|^2] + 6c^2\eta_t^3\sigma^2 + \frac{8(1-\beta_t)^2 L_F^2}{\eta_t}\mathbb{E}[\|\mathbf{x}_{t+1} - \mathbf{x}_t\|^2]$$

$$\leq -260L_F^4\eta_t\mathbb{E}[\|\varkappa_t\|^2] + 6c^2\eta_t^3\sigma^2 + \frac{8(1-\beta_t)^2 L_F^2}{\eta_t}\mathbb{E}[\|\mathbf{x}_{t+1} - \mathbf{x}_t\|^2], \tag{33}$$

where the last inequality is due to $\eta_t^{-1} - \eta_{t-1}^{-1} - 2c\eta_t \leq \frac{\sigma^2}{7L_F k^3}\eta_t - 2(\frac{\sigma^2}{14L_F k^3} + 130L_F^4)\eta_t \leq -260L_F^4\eta_t$.

Taking summation of Eq. (33) from 1 to $T$, we have

$$260L_F^4 \sum_{t=1}^T \eta_t\mathbb{E}[\|\varkappa_t\|^2] \leq \frac{\mathbb{E}[\|\varkappa_1\|^2]}{\eta_0} - \frac{\mathbb{E}[\|\varkappa_{T+1}\|^2]}{\eta_T} + \sum_{t=1}^T 6c^2\eta_t^3\sigma^2 + 8L_F^2 \sum_{t=1}^T \frac{1}{\eta_t}\mathbb{E}[\|\mathbf{x}_{t+1} - \mathbf{x}_t\|^2]. \tag{34}$$

In the same way with Eq. (15) and $\eta_t \leq \eta_1, \forall t \geq 1$, we could also have

$$\frac{1-2\eta_1 L_F}{4} \sum_{t=1}^T \frac{1}{\eta_t}\|\mathbf{x}_{t+1} - \mathbf{x}_t\|^2 \leq \sum_{t=1}^T \frac{1-2\eta_t L_F}{4\eta_t}\|\mathbf{x}_{t+1} - \mathbf{x}_t\|^2 \leq \Delta + \sum_{t=1}^T \eta_t\|\mathbf{z}_t - \nabla F(\mathbf{x}_t)\|^2. \tag{35}$$

Noting $\eta_1 L_F \leq \frac{1}{4}$ and invoking Lemma 12, we obtain

$$\sum_{t=1}^T \frac{1}{\eta_t}\mathbb{E}[\|\mathbf{x}_{t+1} - \mathbf{x}_t\|^2] \leq \frac{4}{1-2\eta_1 L_F}(\Delta + \sum_{t=1}^T \eta_t\mathbb{E}[\|\mathbf{z}_t - \nabla F(\mathbf{x}_t)\|^2])$$

$$\leq 8\Delta + 8\sum_{t=1}^T \eta_t\mathbb{E}[\|\mathbf{z}_t - \nabla F(\mathbf{x}_t)\|^2]$$

$$\leq 8\Delta + 32L_F^2 \sum_{t=1}^T \eta_t\mathbb{E}[\|\varkappa_t\|^2]. \tag{36}$$

Combining Eqs. (34, 36), we have

$$4L_F^4 \sum_{t=1}^T \eta_t\mathbb{E}[\|\varkappa_t\|^2] \leq \frac{\mathbb{E}[\|\varkappa_1\|^2]}{\eta_0} - \frac{\mathbb{E}[\|\varkappa_{T+1}\|^2]}{\eta_T} + \sum_{t=1}^T 6c^2\eta_t^3\sigma^2 + 64L_F^2\Delta. \tag{37}$$

This complete the proof. $\qquad\square$

## C.2   Proof of Theorem 2

*Proof.* Noting the monotonity of $\eta_t$ and dividing $\frac{\eta_1}{1/4-\eta_1 L_F/2}$ on both sides of Eq. (35), we have

$$\sum_{t=1}^T \|\mathbf{x}_{t+1} - \mathbf{x}_t\|^2 \leq \frac{1}{1/4-\eta_1 L_F/2}\left(\eta_1\Delta + \eta_1\sum_{t=1}^T \eta_t\|\mathbf{z}_t - \nabla F(\mathbf{x}_t)\|^2\right). \tag{38}$$

By the same method used in the proof of Theorem 2 in Xu et al. (2019), we have the following inequality,

$$\|\mathbf{z}_t - \nabla F(\mathbf{x}_{t+1}) + \frac{1}{\eta_t}(\mathbf{x}_t - \mathbf{x}_{t+1})\|^2 \leq 2\|\mathbf{z}_t - \nabla F(\mathbf{x}_t)\|^2 + \frac{2\left(\bar{F}(\mathbf{x}_{t+1}) - \bar{F}(\mathbf{x}_t)\right)}{\eta_t}$$
$$+ (2L_F^2 + \frac{3L_F}{\eta_t})\|\mathbf{x}_{t+1} - \mathbf{x}_t\|^2.$$

Multiplying $\eta_t$ on both sides of the above inequality and taking summation from 1 to $T$, we have

$$\sum_{t=1}^{T} \eta_t \|\mathbf{z}_t - \nabla F(\mathbf{x}_{t+1}) + \frac{1}{\eta_t}(\mathbf{x}_{t+1} - \mathbf{x}_t)\|^2$$

$$\overset{(a)}{\leq} 2\sum_{t=1}^{T} \eta_t \|\mathbf{z}_t - \nabla F(\mathbf{x}_t)\|^2 + 2\Delta + 5L_F \left( \frac{1}{1/4 - \eta_1 L_F/2} \left( \eta_1 \Delta + \eta_1 \sum_{t=1}^{T} \eta_t \|\mathbf{z}_t - \nabla F(\mathbf{x}_t)\|^2 \right) \right)$$

$$\overset{(b)}{\leq} 12\sum_{t=1}^{T} \eta_t \|\mathbf{z}_t - \nabla F(\mathbf{x}_t)\|^2 + 12\Delta, \tag{39}$$

where inequality (a) is due to $(2L_F^2 + \frac{3L_F}{\eta_t}) \leq \frac{5L_F}{\eta_t}$, inequality (b) is due to $\eta_1 L_F \leq \frac{1}{4}$ and $\frac{1}{1/4-\eta_1 L_F/2} \leq 8$.

Combining Eqs. (37, 39) and invoking Lemma 12 we have

$$\sum_{t=1}^{T} \eta_t \|\mathbf{z}_t - \nabla F(\mathbf{x}_{t+1}) + \frac{1}{\eta_t}(\mathbf{x}_{t+1} - \mathbf{x}_t)\|^2$$

$$\leq 48L_F^2 \sum_{t=1}^{T} \eta_t \mathbb{E}[\|\varkappa_t\|^2] + 12\Delta$$

$$\leq 12 \left( \frac{\mathbb{E}[\|\varkappa_1\|^2]}{\eta_0} - \frac{\mathbb{E}[\|\varkappa_{T+1}\|^2]}{\eta_T} + \sum_{t=1}^{T} 6c^2 \eta_t^3 \sigma^2 + 64L_F^2 \Delta \right) + 12\Delta. \tag{40}$$

Noting the monotonity of $\eta_t$ and dividing $T\eta_T$ on both sides of Eq. (40), we obtain

$$\frac{1}{T} \sum_{t=1}^{T} \|\mathbf{z}_t - \nabla F(\mathbf{x}_{t+1}) + \frac{1}{\eta_t}(\mathbf{x}_{t+1} - \mathbf{x}_t)\|^2$$

$$\leq 12 \left( \frac{\mathbb{E}[\|\varkappa_1\|^2]}{T\eta_T \eta_0} - \frac{\mathbb{E}[\|\varkappa_{T+1}\|^2]}{T\eta_T^2} + \frac{1}{T\eta_T} \sum_{t=1}^{T} 6c^2 \eta_t^3 \sigma^2 + \frac{64L_F^2 \Delta}{T\eta_T} \right) + \frac{12\Delta}{T\eta_T}. \tag{41}$$

Combining Eqs. (18, 41) and noting $\sum_{t=1}^{T} \eta_t^3 \leq \mathcal{O}(\log T)$, we get the conclusion that

$$\mathbb{E}[\text{dist}(0, \hat{\partial}\bar{F}(\mathbf{x}_R))^2] \leq \frac{1}{T} \sum_{t=1}^{T} \mathbb{E}[\|\mathbf{z}_t - \nabla F(\mathbf{x}_{t+1}) + \frac{1}{\eta_t}(\mathbf{x}_{t+1} - \mathbf{x}_t)\|^2]$$

$$\leq 12 \left( \frac{\mathbb{E}[\|\varkappa_1\|^2]}{T\eta_T \eta_0} + \frac{1}{T\eta_T} \sum_{t=1}^{T} 6c^2 \eta_t^3 \sigma^2 + \frac{64L_F^2 \Delta}{T\eta_T} \right) + \frac{12\Delta}{T\eta_T}$$

$$\leq \mathcal{O} \left( \frac{\log T}{T^{2/3}} \right).$$

This complete the proof. □

# D    Proofs in Section 5

## D.1    Technical Lemmas

**Lemma 15.** *If $\ell_i(\mathbf{w})$ is convex for all $i$, we can show that $F(\mathbf{w}, \lambda)$ is jointly convex in terms of $(\mathbf{w}, \lambda)$.*

*Proof.* We have

$$F(\mathbf{w}, \lambda) = \max_{\mathbf{p} \in \Delta_n} \underbrace{\sum_{i=1}^n p_i \ell_i(\mathbf{w}) - \lambda(\sum_{i=1}^n p_i \log(np_i) - \rho) - \lambda_0 \rho}_{G(\mathbf{w}, \lambda, \mathbf{p})}.$$

Since $G(\mathbf{w}, \lambda, \mathbf{p})$ is jointly convex in terms of $(\mathbf{w}, \lambda)$ for every fixed $\mathbf{p}$, $F(\mathbf{w}, \lambda)$ is jointly convex in terms of $(\mathbf{w}, \lambda)$. $\qquad\square$

**Lemma 16.** *Under Assumption 1, 2, run Algorithm 1 with $\eta \leq \frac{\beta}{4L_F\sqrt{9+20L_g^2}} \leq \frac{1}{6L_F}$ and apply SCDRO to the new objective $\bar{F}_\mu(\mathbf{x})$ by adding $\mu\mathbf{x}_t$ to $(\nabla f_{\lambda_t}(s_t)\nabla_\mathbf{w} g_i(\mathbf{x}_t)^\top, \nabla f_{\lambda_t}(s_t)\nabla_\lambda g_i(\mathbf{x}_t) + \log(s_t) + \rho)^\top$ in Eq. (1) of Algorithm 1, where $\mu$ is a small constant to be determined later. Without loss of the generality, we assume $0 < \mu \leq \frac{1}{2}$ and then we have*

$$\frac{1}{T}\sum_{t=1}^T \mathbb{E}[\|\mathbf{z}_t - \nabla F_\mu(\mathbf{x}_t)\|^2] \leq \frac{2\mathbb{E}[\|\mathbf{z}_1 - \nabla F_\mu(\mathbf{x}_1)\|^2]}{\beta T} + \frac{\Delta_\mu}{\eta T} + \frac{20L_F\mathbb{E}[\|g(\mathbf{x}_1) - s_1\|^2]}{\beta T} + 24\beta L_F^2\sigma^2.$$

*Proof.* To facilitate our proof statement, we define the following notations:

$$\nabla F_\mu(\mathbf{x}_t)^\top = (\nabla_\mathbf{w} F_\mu(\mathbf{x}_t)^\top, \nabla_\lambda F_\mu(\mathbf{x}_t))$$
$$= (\nabla f_{\lambda_t}(g(\mathbf{x}_t))\nabla_\mathbf{w} g(\mathbf{x}_t)^\top + \mu\mathbf{w}_t^\top, \nabla f_{\lambda_t}(g(\mathbf{x}_t))\nabla_\lambda g(\mathbf{x}_t) + \log(g(\mathbf{x}_t)) + \rho + \mu\lambda_t)$$
$$\widetilde{\nabla} F_\mu(\mathbf{x}_t)^\top$$
$$= (\nabla f_{\lambda_t}(g(\mathbf{x}_t))\nabla_\mathbf{w} g_i(\mathbf{x}_t)^\top + \mu\mathbf{w}_t^\top, \nabla f_{\lambda_t}(g(\mathbf{x}_t))\nabla_\lambda g_i(\mathbf{x}_t) + \log(g(\mathbf{x}_t)) + \rho + \mu\lambda_t)$$
$$G_\mu(\mathbf{x}_t)^\top$$
$$= (G_{\mathbf{w}_t}(\mathbf{x}_t)^\top, G_{\lambda_t}(\mathbf{x}_t)) = (\nabla f_{\lambda_t}(s_t)\nabla_\mathbf{w} g_i(\mathbf{x}_t)^\top + \mu\mathbf{w}_t^\top, \nabla f_{\lambda_t}(s_t)\nabla_\lambda g_i(\mathbf{x}_t) + \log(s_t) + \rho + \mu\lambda_t).$$

It is worth to notice that $\mathbb{E}[\widetilde{\nabla} F_\mu(\mathbf{x}_t)] = \nabla F_\mu(\mathbf{x}_t)$.

Since $F(\mathbf{x})$ is $L_F$-smooth, then we have $F_\mu(\mathbf{x})$ is $L_{F_\mu}$-smooth, where $L_{F_\mu} = (L_F + \mu)$. Noting $L_F > 1$ and $\mu \leq \frac{1}{2}$, we obtain $L_F + \mu \leq \frac{3}{2}L_F$. For every iteration $t$, by simple expansion we have

$$I_t = \mathbb{E}[\|\nabla F_\mu(\mathbf{x}_t) - \mathbf{z}_t\|^2]$$
$$= \mathbb{E}[\|\nabla F_\mu(\mathbf{x}_t) - (1-\beta)\mathbf{z}_{t-1} - \beta G_\mu(\mathbf{x}_t)\|^2]$$
$$= \mathbb{E}[\|(1-\beta)(\nabla F_\mu(\mathbf{x}_t) - \nabla F_\mu(\mathbf{x}_{t-1})) + (1-\beta)\nabla F_\mu(\mathbf{x}_{t-1}) - (1-\beta)\mathbf{z}_{t-1} + \beta\nabla F_\mu(\mathbf{x}_t) - \beta G_\mu(\mathbf{x}_t)\|^2]$$
$$= \mathbb{E}[\|(1-\beta)(\nabla F_\mu(\mathbf{x}_t) - \nabla F_\mu(\mathbf{x}_{t-1})) + (1-\beta)(\nabla F_\mu(\mathbf{x}_{t-1}) - \mathbf{z}_{t-1})\|^2]$$
$$\quad + \mathbb{E}[\|\beta(\widetilde{\nabla} F_\mu(\mathbf{x}_t) - G_\mu(\mathbf{x}_t)) + \beta(\nabla F_\mu(\mathbf{x}_t) - \widetilde{\nabla} F_\mu(\mathbf{x}_t))\|^2]$$
$$= \mathbb{E}[\|(1-\beta)\underbrace{(\nabla F_\mu(\mathbf{x}_t) - \nabla F_\mu(\mathbf{x}_{t-1}))}_{A} + (1-\beta)\underbrace{(\nabla F(\mathbf{x}_{t-1}) - \mathbf{z}_{t-1})}_{B}\|^2]$$
$$\quad + \mathbb{E}[\|\beta\underbrace{(\widetilde{\nabla} F(\mathbf{x}_t) - G(\mathbf{x}_t))}_{C} + \beta\underbrace{(\nabla F(\mathbf{x}_t) - \widetilde{\nabla} F(\mathbf{x}_t))}_{D}\|^2].$$

The above inequality shows that the only difference between $I_t$ in the proof of Lemma 11 and $I_t$ in the proof of Lemma 16 is term $A$.

Therefore, by the same method used in the proof of Lemma 11, we have

$$\sum_{t=1}^T \mathbb{E}[\|\mathbf{z}_t - \nabla F_\mu(\mathbf{x}_t)\|^2]$$
$$\leq \frac{\mathbb{E}[\|\nabla F_\mu(\mathbf{x}_1) - \mathbf{z}_1\|^2]}{\beta} + (\frac{4L_{F_\mu}^2}{\beta^2} + \frac{20L_F^2 L_g^2}{\beta^2})\left(\frac{\eta}{1/4 - \eta L_{F_\mu}/2}(\Delta_\mu + \eta\sum_{t=1}^T \mathbb{E}[\|\mathbf{z}_t - \nabla F_\mu(\mathbf{x}_t)\|^2])\right)$$
$$+ 10L_F^2\left(\frac{\mathbb{E}[\|g(\mathbf{x}_1) - s_1\|^2]}{\beta} + \beta T\sigma^2\right) + 2\beta L_F^2 T\sigma^2.$$

By $L_{F_\mu} \leq \frac{3}{2}L_F$ and $\eta L_{F_\mu} \leq \frac{3}{2}\eta L_F \leq 1/4$, it holds that

$$\sum_{t=1}^{T} \mathbb{E}[\|\mathbf{z}_t - \nabla F_\mu(\mathbf{x}_t)\|^2]$$

$$\leq \frac{\mathbb{E}[\|\nabla F_\mu(\mathbf{x}_1) - \mathbf{z}_1\|^2]}{\beta} + (\frac{9L_F^2}{\beta^2} + \frac{20L_F^2 L_g^2}{\beta^2}) \left(8\eta \left(\Delta_\mu + \eta \sum_{t=1}^{T} \mathbb{E}[\|\mathbf{z}_t - \nabla F_\mu(\mathbf{x}_t)\|^2]\right)\right)$$

$$+ 10L_F^2 \left(\frac{\mathbb{E}[\|g(\mathbf{x}_1) - s_1\|^2]}{\beta} + \beta T \sigma^2\right) + 2\beta L_F^2 T \sigma^2$$

$$\leq \frac{\mathbb{E}[\|\mathbf{z}_1 - \nabla F_\mu(\mathbf{x}_1)\|^2]}{\beta} + \frac{\Delta_\mu}{2\eta} + \frac{1}{2}\sum_{t=1}^{T} \mathbb{E}[\|\mathbf{z}_t - \nabla F_\mu(\mathbf{x}_t)\|^2]$$

$$+ 10L_F^2 \left(\frac{\mathbb{E}[\|g(\mathbf{x}_1) - s_1\|^2]}{\beta} + \beta T \sigma^2\right) + 2\beta L_F^2 T \sigma^2, \tag{42}$$

where the last inequality is due to $8(9L_F^2 + 20L_F^2 L_g^2)\eta^2 \leq \frac{\beta^2}{2}$.

Rearranging terms and dividing $T$ on both sides of Eq. (42), we complete the proof of this Lemma. $\qquad\square$

**Lemma 17.** *At the $k$-th stage of RASCDRO, let $\beta_k = c\eta_k^2$ and $c = 512L_F^4$ we have*

$$\frac{1}{8L_F^2 T_k} \sum_{t=1}^{T_k} \mathbb{E}[\|\mathbf{z}_t - \nabla F_\mu(\mathbf{x}_t)\|^2] \leq \frac{\mathbb{E}[\|\varkappa_k\|^2]}{\beta_k T_k} + 6\beta_k \sigma^2 + \frac{64L_F^2 \mathbb{E}[\Delta_k^\mu]\eta_k}{\beta_k T_k}, \tag{43}$$

*where $\Delta_k^\mu = F_\mu(\mathbf{x}_k) - \inf_{\mathbf{x}\in\mathcal{X}} F_\mu(\mathbf{x})$.*

*Proof.* Recall the definition of $\|\varkappa_t\|^2$ and by the same proof of Lemma 12 we have

$$\|\mathbf{z}_t - \nabla F_\mu(\mathbf{x}_t)\|^2 \leq 4L_F^2 \|\varkappa_t\|^2. \tag{44}$$

Denote $\varkappa_t$ at $k$th-stage as $\varkappa_k^t$, and by Lemma 13, at the $k$th-stage in RASCDRO we have

$$\mathbb{E}[\|\varkappa_k^{t+1}\|^2] \leq (1 - \beta_k)^2 \|\varkappa_k^t\|^2 + 6\beta_k^2 \sigma^2 + 8L_F^2(1 - \beta_k)^2 \|\mathbf{x}_{t+1} - \mathbf{x}_t\|^2$$

$$\leq (1 - \beta_k)^{2t} \|\varkappa_k\|^2 + 6\beta_k^2 \sigma^2 \sum_{i=1}^{t}(1 - \beta_k)^{2(t-i)}$$

$$+ 8L_F^2(1 - \beta_k)^2 \sum_{i=1}^{t}(1 - \beta_k)^{2(t-i)} \|\mathbf{x}_{i+1} - \mathbf{x}_i\|^2$$

$$\leq (1 - \beta_k)^{2t} \|\varkappa_k\|^2 + 6\beta_k \sigma^2 \tag{45}$$

$$+ 8L_F^2(1 - \beta_k)^2 \sum_{i=1}^{t}(1 - \beta_k)^{2(t-i)} \|\mathbf{x}_{i+1} - \mathbf{x}_i\|^2.$$

Combining Eqs. (44,45), we obtain

$$\frac{1}{4L_F^2 T_k} \sum_{t=1}^{T_k} \mathbb{E}[\|\mathbf{z}_t - \nabla F_\mu(\mathbf{x}_t)\|^2]$$

$$\leq \frac{1}{T_k} \sum_{t=1}^{T_k} \mathbb{E}[(1 - \beta_k)^2 \|\varkappa_k^t\|^2 + 6\beta_k^2 \sigma^2 + 8L_F^2(1 - \beta_k)^2 \|\mathbf{x}_{t+1} - \mathbf{x}_t\|^2]$$

$$\leq \frac{1}{T_k} \sum_{t=1}^{T_k}(1 - \beta_k)^{2t-2} \mathbb{E}[\|\varkappa_k\|^2] + 6\beta_k \sigma^2 + \frac{8L_F^2(1 - \beta_k)^2}{T_k} \sum_{t=1}^{T_k} \sum_{i=1}^{t-1}(1 - \beta_k)^{2(t-i)} \mathbb{E}[\|\mathbf{x}_{i+1} - \mathbf{x}_i\|^2].$$

Noting $\sum_{t=1}^{T_k}(1-\beta_k)^{2t-2} \le 1/\beta_k$ and invoking Eq. (38), we have

$$\frac{1}{4L_F^2 T_k}\sum_{t=1}^{T_k}\mathbb{E}[\|\mathbf{z}_t - \nabla F_\mu(\mathbf{x}_t)\|^2]$$

$$\le \frac{\mathbb{E}[\|\varkappa_k\|^2]}{\beta_k T_k} + 6\beta_k\sigma^2 + \frac{8L_F^2(1-\beta_k)^2}{\beta_k T_k}\sum_{t=1}^{T_k}\mathbb{E}[\|\mathbf{x}_{t+1} - \mathbf{x}_t\|^2]$$

$$\le \frac{\mathbb{E}[\|\varkappa_k\|^2]}{\beta_k T_k} + 6\beta_k\sigma^2 + \frac{8L_F^2(1-\beta_k)^2}{\beta_k T_k}\left(\frac{\eta_k}{1/4 - \eta_k L_{F_\mu}/2}\left(\mathbb{E}[\Delta_k^\mu] + \eta_k\sum_{t=1}^{T_k}\mathbb{E}[\|\mathbf{z}_t - \nabla F_\mu(\mathbf{x}_t)\|^2]\right)\right)$$

$$\le \frac{\mathbb{E}[\|\varkappa_k\|^2]}{\beta_k T_k} + 6\beta_k\sigma^2 + \frac{64L_F^2\mathbb{E}[\Delta_k^\mu]\eta_k}{\beta_k T_k} + \frac{64L_F^2\eta_k^2}{\beta_k T_k}\sum_{t=1}^{T_k}\|\mathbf{z}_t - \nabla F_\mu(\mathbf{x}_t)\|^2,$$

where the last inequality is due to $1/(1/4 - \eta_k L_{F_\mu}/2) \le 8, (1-\beta_k)^2 \le 1, L_{\nabla g}^2 + L_g^2 \le 2L_F^2$.

Invoking $\beta_k = c\eta_k^2$ and $c = 576L_F^4$ to above inequality, we get the conclusion that

$$\frac{1}{8L_F^2 T_k}\sum_{t=1}^{T_k}\mathbb{E}[\|\mathbf{z}_t - \nabla F_\mu(\mathbf{x}_t)\|^2] \le \frac{\mathbb{E}[\|\varkappa_k\|^2]}{\beta_k T_k} + 6\beta_k\sigma^2 + \frac{64L_F^2\mathbb{E}[\Delta_k^\mu]\eta_k}{\beta_k T_k}.$$

$\square$

## D.2 Proof of Lemma 3

*Proof.* Since $\ell_i(\mathbf{w})$ is convex for all $i$, by Lemma 15 we know $F(\mathbf{x})$ is convex. And thus by the definition of $\bar{F}_\mu(\mathbf{x})$ we have $\bar{F}_\mu(\mathbf{x})$ is a strongly convex function. Then by strong convexity, we have

$$\bar{F}_\mu(\mathbf{y}) \ge \bar{F}_\mu(\mathbf{x}) + \mathbf{v}^\top(\mathbf{y}-\mathbf{x}) + \frac{\mu}{2}\|\mathbf{y}-\mathbf{x}\|^2, \forall \mathbf{x},\mathbf{y}\in\mathcal{X}, \mathbf{v}\in\partial\bar{F}_\mu(\mathbf{x}).$$

Then

$$\inf_{\mathbf{x}\in\mathcal{X}}\bar{F}_\mu(\mathbf{x}) \ge \min_{\mathbf{y}\in\mathcal{X}}\bar{F}_\mu(\mathbf{x}) + \mathbf{v}^\top(\mathbf{y}-\mathbf{x}) + \frac{\mu}{2}\|\mathbf{y}-\mathbf{x}\|^2$$

$$\ge \min_{\mathbf{y}}\bar{F}_\mu(\mathbf{x}) + \mathbf{v}^\top(\mathbf{y}-\mathbf{x}) + \frac{\mu}{2}\|\mathbf{y}-\mathbf{x}\|^2$$

$$= \bar{F}_\mu(\mathbf{x}) - \frac{\|\mathbf{v}\|^2}{2\mu}, \quad \forall \mathbf{v}\in\partial\bar{F}_\mu(\mathbf{x}).$$

Hence, $\frac{\|\mathbf{v}\|^2}{2\mu} \ge \bar{F}_\mu(\mathbf{x}) - \inf_{\mathbf{x}\in\mathcal{X}}\bar{F}_\mu(\mathbf{x}), \forall \mathbf{v}\in\partial\bar{F}_\mu(\mathbf{x})$, which implies

$$\text{dist}(0, \partial\bar{F}_\mu(\mathbf{x}))^2 \ge 2\mu\left(\bar{F}_\mu(\mathbf{x}) - \bar{F}_\mu(\mathbf{x}_*)\right).$$

$\square$

## D.3 Proof of Lemma 4

*Proof.* We use inductions to prove $\mathbb{E}[\|\mathbf{z}_k - \nabla F_\mu(\mathbf{x}_k)\|^2] \le \mu\epsilon_k/4$, $\mathbb{E}[\|g(\mathbf{x}_k) - s_k\|^2] \le \mu\epsilon_k/4$ and $\mathbb{E}[F_\mu(\mathbf{x}_k) - \inf_{\mathbf{x}\in\mathcal{X}}F_\mu(\mathbf{x})] \le \epsilon_k$. Let's consider the first stage in the beginning.

Let $\epsilon_1 = \Delta_\mu$, thus $\mathbb{E}[F_\mu(\mathbf{x}_1) - \inf_{\mathbf{x}\in\mathcal{X}}F_\mu(\mathbf{x})] \le \epsilon_1$. And we can use a batch size of $4/\mu\epsilon_1$ for initialization.to make sure $\mathbb{E}[\|\nabla F_\mu(\mathbf{x}_1) - \mathbf{z}_1\|^2] \le \mu\epsilon_1/4, \mathbb{E}[\|s_1 - g(\mathbf{x}_1)\|^2] \le \mu\epsilon_1/4$.

Suppose that $\mathbb{E}[\|g(\mathbf{x}_{k-1}) - s_{k-1}\|^2] \le \mu\epsilon_{k-1}/4$, $\mathbb{E}[\|\mathbf{z}_{k-1} - \nabla F_\mu(\mathbf{x}_{k-1})\|^2] \le \mu\epsilon_{k-1}/4$ and $\mathbb{E}[F_\mu(\mathbf{x}_{k-1}) - \inf_{\mathbf{x}\in\mathcal{X}}F_\mu(\mathbf{x})] \le \epsilon_{k-1}$. By setting $\beta_{k-1} = \min\{\frac{\mu\epsilon_{k-1}}{384L_F^2\sigma^2}, \frac{1}{384L_F^2}\}, \eta_{k-1} = \min\{\frac{\mu\epsilon_{k-1}}{4608L_F^4\sigma^2}, \frac{1}{4608L_F^4}\}$ and $T_{k-1} = \max\{\frac{147456L_F^4\sigma^2}{\mu^2\epsilon_{k-1}}, \frac{147456L_F^4}{\mu}\}$, it is easy to obtain that $\eta_{k-1} \le \frac{\beta_{k-1}}{4L_F\sqrt{9+20L_g^2}}$. Therefore, invoking Lemma 16 we

have

$$\mathbb{E}[\|\mathbf{z}_k - \nabla F_\mu(\mathbf{x}_k)\|^2]$$

$$\leq \frac{1}{T_{k-1}} \sum_{t=1}^{T_{k-1}} \mathbb{E}[\|\mathbf{z}_t - \nabla F_\mu(\mathbf{x}_t)\|^2]$$

$$\leq \frac{\mathbb{E}[2\|\mathbf{z}_{k-1} - \nabla F_\mu(\mathbf{x}_{k-1})\|^2]}{\beta_{k-1} T_{k-1}} + \frac{\mathbb{E}[\Delta_{k-1}^\mu]}{\eta_{k-1} T_{k-1}} + \frac{20 L_F \mathbb{E}[\|g(\mathbf{x}_{k-1}) - s_{k-1}\|^2]}{\beta_{k-1} T_{k-1}} + 24\beta_k L_F^2 \sigma^2$$

$$\leq \frac{\mu\epsilon_{k-1}}{2\beta_{k-1} T_{k-1}} + \frac{\epsilon_{k-1}}{\eta_{k-1} T_{k-1}} + \frac{5 L_F \mu\epsilon_{k-1}}{\beta_{k-1} T_{k-1}} + 24\beta_{k-1} L_F^2 \sigma^2.$$

Without loss of the generality, we consider the case $\mu\epsilon_{k-1}/\sigma^2 \leq 1$. By definition have $\beta_{k-1} = \mu\epsilon_{k-1}/(384 L_F^2 \sigma^2)$, $\eta_{k-1} = \mu\epsilon_{k-1}/(4608 L_F^4 \sigma^2)$ and $T_{k-1} = 147456 L_F^4 \sigma^2/(\mu^2\epsilon_{k-1})$, which imply

$$\frac{1}{\beta_{k-1} T_{k-1}} \leq \frac{\mu}{384 L_F^2}, \quad \frac{1}{\eta_{k-1} T_{k-1}} \leq \frac{\mu}{32} \text{ and } 24\beta_{k-1} L_F^2 \sigma^2 \leq \frac{\mu\epsilon_{k-1}}{16}.$$

Then, note $L_F \geq 1$, $\mu < 1$ and $\epsilon_k = \epsilon_{k-1}/2$ we have

$$\mathbb{E}[\|\mathbf{z}_k - \nabla F_\mu(\mathbf{x}_k)\|^2] \leq \frac{\mu^2\epsilon_{k-1}}{768 L_F^2} + \frac{\mu\epsilon_{k-1}}{16} + \frac{5\mu^2\epsilon_{k-1}}{384 L_F} + \frac{\mu\epsilon_{k-1}}{8}$$

$$\leq \frac{\mu\epsilon_{k-1}}{768} + \frac{\mu\epsilon_{k-1}}{32} + \frac{5\mu\epsilon_k}{192} + \frac{\mu\epsilon_{k-1}}{16}$$

$$= \frac{\mu\epsilon_k}{192} + \frac{\mu\epsilon_k}{16} + \frac{\mu\epsilon_k}{40} + \frac{\mu\epsilon_k}{8}$$

$$\leq \frac{\mu\epsilon_k}{4}.$$

Next we need to show $\mathbb{E}[\|g(\mathbf{x}_k) - s_k\|^2] \leq \mu\epsilon_k/4$ under the assumption that $\mathbb{E}[\|g(\mathbf{x}_{k-1}) - s_{k-1}\|^2] \leq \mu\epsilon_{k-1}/4$.

By Lemma 9, we have

$$\mathbb{E}[\|g(\mathbf{x}_k) - s_k\|^2]$$

$$= \frac{1}{T_{k-1}} \sum_{t=1}^{T_{k-1}} \mathbb{E}[\|g(\mathbf{x}_t) - s_t\|^2]$$

$$\leq \frac{\mathbb{E}[\|g(\mathbf{x}_{k-1}) - s_{k-1}\|^2]}{\beta_{k-1} T_{k-1}} + \frac{2 L_g^2}{\beta_{k-1}^2 T_{k-1}} \sum_{t=1}^{T_{k-1}} \mathbb{E}[\|\mathbf{x}_{t+1} - \mathbf{x}_t\|^2] + \beta_{k-1}\sigma^2$$

$$\leq \frac{\mu\epsilon_{k-1}}{4\beta_{k-1} T_{k-1}} + \frac{2 L_g^2}{\beta_{k-1}^2 T_{k-1}} \left( \frac{\eta_{k-1}}{1/4 - \eta_{k-1} L_{F_\mu}/2} \left( \mathbb{E}[\Delta_{k-1}^\mu] + \eta_{k-1} \sum_{t=1}^{T_{k-1}} \mathbb{E}[\|\mathbf{z}_t - \nabla F_\mu(\mathbf{x}_t)\|^2] \right) \right)$$

$$+ \beta_{k-1}\sigma^2,$$

where $\Delta_{k-1}^\mu = F_\mu(\mathbf{x}_{k-1}) - \inf_{\mathbf{x} \in \mathcal{X}} F_\mu(\mathbf{x})$. With $1/(1/4 - \eta_{k-1} L_{F_\mu}/2) \leq 8$, $\mathbb{E}[\|g(\mathbf{x}_{k-1}) - s_{k-1}\|^2] \leq \mu\epsilon_{k-1}/4$ and $\mathbb{E}[F_\mu(\mathbf{x}_{k-1}) - \inf_{\mathbf{x} \in \mathcal{X}} F_\mu(\mathbf{x})] \leq \epsilon_{k-1}$, it holds that

$$\mathbb{E}[\|g(\mathbf{x}_k) - s_k\|^2] \leq \frac{\mu\epsilon_{k-1}}{2\beta_{k-1} T_{k-1}} + \frac{16 L_g^2 \eta_{k-1}\epsilon_{k-1}}{\beta_{k-1}^2 T_{k-1}} + \frac{4 L_g^2 \eta_{k-1}^2 \mu\epsilon_{k-1}}{\beta_{k-1}^2} + \beta_{k-1}\sigma^2$$

$$\leq \frac{\mu\epsilon_k}{384 L_F^2} + \frac{L_g^2 \mu\epsilon_{k-1}}{288 L_F^4} + \frac{L_g^2 \mu\epsilon_{k-1}}{36 L_F^4} + \frac{\mu\epsilon_{k-1}}{192 L_F^2}$$

$$\leq \frac{\mu\epsilon_k}{2}.$$

Invoking Lemma 10, at $(k-1)$-th stage $(k > 1)$ we have

$$\mathbb{E}[\text{dist}(0, \hat{\partial}\bar{F}_\mu(\mathbf{x}_k))^2]$$

$$\leq \frac{2 + 40 L_{F_\mu} \eta_{k-1}}{T_{k-1}} \sum_{t=1}^{T_{k-1}} \mathbb{E}[\|\mathbf{z}_t - \nabla F_\mu(\mathbf{x}_t)\|^2] + \frac{2\mathbb{E}[\Delta_{k-1}^\mu]}{\eta_{k-1}T_{k-1}} + \frac{40 L_{F_\mu} \mathbb{E}[\Delta_{k-1}^\mu]}{T_{k-1}}$$

$$\leq \frac{(2 + 40 L_{F_\mu}\eta_{k-1})\mu\epsilon_{k-1}}{4} + \frac{2\epsilon_{k-1}}{\eta_{k-1}T_{k-1}} + \frac{40 L_{F_\mu}\epsilon_{k-1}}{T_{k-1}}$$

$$\leq \frac{197\mu\epsilon_k}{192} + \frac{\mu\epsilon_k}{8} + \frac{40 L_{F_\mu}\mu\epsilon_{k-1}}{147456 L_F^4}$$

$$\leq 2\mu\epsilon_k,$$

where the second inequality is due to $L_{F_\mu}\eta_{k-1} \leq (3/2)L_F\eta_{k-1} \leq 1/1536$.

Since $F_\mu(\mathbf{x}_k) \leq \bar{F}_\mu(\mathbf{x}_k)$ and $\inf_{\mathbf{x}\in\mathcal{X}} F_\mu(\mathbf{x}) = \inf_{\mathbf{x}\in\mathcal{X}} \bar{F}_\mu(\mathbf{x})$, applying Lemma 3 we have

$$\mathbb{E}[F_\mu(\mathbf{x}_k) - \inf_{\mathbf{x}\in\mathcal{X}} F_\mu(\mathbf{x})] \leq \mathbb{E}[\bar{F}_\mu(\mathbf{x}_k) - \inf_{\mathbf{x}\in\mathcal{X}} \bar{F}_\mu(\mathbf{x})] \leq \frac{1}{2\mu}\mathbb{E}[\text{dist}(0, \hat{\partial}\bar{F}_\mu(\mathbf{x}_k))^2] \leq \frac{2\mu\epsilon_k}{2\mu} = \epsilon_k.$$

This complete the proof of this Lemma. $\qquad\square$

## D.4 Proof of Theorem 3

*Proof.* Invoking Lemma 4, then after $K = \mathcal{O}(\log_2(\epsilon_1/\epsilon))$ stages, we have

$$\mathbb{E}[F_\mu(\mathbf{x}_K) - \inf_{\mathbf{x}\in\mathcal{X}} F_\mu(\mathbf{x})] \leq \epsilon_K = \frac{\epsilon_1}{2^{K-1}} = \epsilon.$$

Since $\sum_{k=1}^K 2^k = \mathcal{O}(1/\epsilon)$, the overall oracle complexity is

$$\sum_{k=1}^K T_k + \frac{4}{\mu\epsilon_1} \leq 36864\sigma^2 L_F^4 \sum_{k=2}^K \frac{1}{\mu^2\epsilon_k} + \frac{4}{\mu\epsilon_1}$$

$$\leq \frac{36864\sigma^2 L_F^4}{\mu^2\epsilon} \sum_{k=1}^K \frac{1}{2^k} + \frac{4}{\mu\epsilon_1}$$

$$\leq \mathcal{O}(\frac{1}{\mu^2\epsilon}).$$

$\qquad\square$

## D.5 Proof of Corollary 1

It is easy to note that $F_\mu(\mathbf{x}_K) - F_\mu(\mathbf{x}_*) \leq F_\mu(\mathbf{x}_K) - \inf_{\mathbf{x}\in\mathcal{X}} F_\mu(\mathbf{x})$, where $\mathbf{x}_* = \arg\min_{\mathbf{x}\in\mathcal{X}} F(\mathbf{x})$. Therefore, if after $K$ stages it holds that $\mathbb{E}[F_\mu(\mathbf{x}_K) - \inf_{\mathbf{x}\in\mathcal{X}} F_\mu(\mathbf{x})] \leq \epsilon/2$ with an oracle complexity of $\mathcal{O}(1/\mu^2\epsilon)$, we have $\mathbb{E}[F_\mu(\mathbf{x}_K) - F_\mu(\mathbf{x}_*)] \leq \epsilon/2$ , i.e., $\mathbb{E}[F(\mathbf{x}_K) + \mu\|\mathbf{x}_K\|^2/2 - F(\mathbf{x}_*) - \mu\|\mathbf{x}_*\|^2/2] \leq \epsilon/2$. By Assumption 1(a) $\mathcal{W}$ is bounded by $R$, and then by setting $\mu = \epsilon/(2(R^2 + \tilde{\lambda}^2))$, with $\|\mathbf{x}\|^2 \leq (R^2 + \tilde{\lambda}^2)$ we have

$$\mathbb{E}[F(\mathbf{x}_K) - F(\mathbf{x}_*)] \leq \frac{\epsilon}{2} + (2(R^2 + \tilde{\lambda}^2))\frac{\mu}{2} \leq \frac{\epsilon}{2} + \frac{\epsilon}{2} \leq \epsilon$$

with an oracle complexity of $\mathcal{O}(1/\epsilon^3)$.

## D.6 Proof of Lemma 5

*Proof.* We use inductions to prove $\mathbb{E}[\|\varkappa_k\|^2] \leq \mu\epsilon_k/16L_F^2$ and $\mathbb{E}[F_\mu(\mathbf{x}_k) - \inf_{\mathbf{x}\in\mathcal{X}} F_\mu(\mathbf{x})] \leq \epsilon_k$. Let's consider the first stage in the beginning.

Let $\epsilon_1 = \Delta_\mu$, thus $\mathbb{E}[F_\mu(\mathbf{x}_1) - \inf_{\mathbf{x}\in\mathcal{X}} F_\mu(\mathbf{x})] \leq \epsilon_1$. And we can use a batch size of $48 L_F^2/\mu\epsilon_1$ for initialization to make sure $\mathbb{E}[\|\varkappa_1\|^2] = \mathbb{E}[\|s_1 - g(\mathbf{x}_1)\|^2 + \|\mathbf{v}_1 - \nabla_{\mathbf{w}}g(\mathbf{x}_1)\|^2 + |u_1 - \nabla_\lambda g(\mathbf{x}_1)|^2] \leq \mu\epsilon_1/16L_F^2$.

Suppose that $\mathbb{E}[\|\varkappa_{k-1}\|^2] \leq \mu\epsilon_{k-1}/16L_F^2$ and $\mathbb{E}[F_\mu(\mathbf{x}_{k-1}) - \inf_{\mathbf{x}\in\mathcal{X}} F_\mu(\mathbf{x})] \leq \epsilon_{k-1}$. By setting $\beta_{k-1} = \min\{\frac{\mu\epsilon_{k-1}}{768 L_F^2\sigma^2}, \frac{1}{768 L_F^2}\}, \eta_{k-1} = \min\{\frac{\sqrt{\mu\epsilon_{k-1}}}{18432 L_F^3\sigma^2}, \frac{1}{18432 L_F^4}\}$ and $T_{k-1} = \max\{\frac{147456 L_F^3\sigma}{\mu^{3/2}\sqrt{\epsilon_{k-1}}}, \frac{147456 L_F^4\sigma^2}{\mu\epsilon_{k-1}}, \frac{147456 L_F^4}{\mu}\}$.

Then following the above Lemma 17, for $k \geq 1$,

$$\mathbb{E}[\|\varkappa_k\|^2] \leq \frac{1}{4L_F^2 T_{k-1}} \sum_{t=1}^{T_{k-1}} \mathbb{E}[\|\mathbf{z}_t - \nabla F_\mu(\mathbf{x}_t)\|^2]$$

$$\leq \frac{2\mathbb{E}[\|\varkappa_{k-1}\|^2]}{\beta_{k-1} T_{k-1}} + 12\beta_{k-1}\sigma^2 + \frac{128 L_F^2 \mathbb{E}[\Delta_{k-1}^\mu]\eta_{k-1}}{\beta_{k-1} T_{k-1}}$$

$$\leq \frac{\mu\epsilon_{k-1}}{4L_F^2 \beta_{k-1} T_{k-1}} + 12\beta_{k-1}\sigma^2 + \frac{128 L_F^2 \epsilon_{k-1}\eta_{k-1}}{\beta_{k-1} T_{k-1}}.$$

Without loss of the generality, we consider the case $\mu\epsilon_{k-1}/\sigma^2 \leq 1$. By definition we have $\beta_{k-1} = \mu\epsilon_{k-1}/(768 L_F^2 \sigma^2)$, $\eta_{k-1} = \sqrt{\mu\epsilon_{k-1}}/(9216 L_F^3 \sigma)$, which imply

$$\frac{1}{\beta_{k-1} T_{k-1}} \leq \frac{1}{96 L_F^2}, \quad \frac{1}{\eta_{k-1} T_{k-1}} \leq \frac{\mu}{8} \text{ and } 12\beta_k \sigma^2 \leq \frac{\mu\epsilon_{k-1}}{64 L_F^2}.$$

Then, noting $L_F \geq 1$, $\mu < 1$ and $\epsilon_k = \epsilon_{k-1}/2$ we have

$$\mathbb{E}[\|\varkappa_k\|^2] \leq \frac{\mu\epsilon_{k-1}}{384 L_F^4} + \frac{\mu\epsilon_{k-1}}{64 L_F^2} + \frac{\mu\epsilon_{k-1}}{6912 L_F^4}$$

$$\leq \frac{\mu\epsilon_k}{192 L_F^2} + \frac{\mu\epsilon_k}{32 L_F^2} + \frac{\mu\epsilon_k}{3456 L_F^2}$$

$$\leq \frac{\mu\epsilon_k}{16 L_F^2}.$$

Then by Eq. (44), we have $\|\mathbf{z}_k - \nabla F_\mu(\mathbf{x}_k)\|^2 \leq 4L_F^2 \|\varkappa_k\|^2 \leq \mu\epsilon_k/4$. Invoking Lemma 10, at $(k-1)$-th stage $(k > 1)$ we have

$$\mathbb{E}[\text{dist}(0, \hat{\partial}\bar{F}_\mu(\mathbf{x}_k))^2]$$

$$\leq \frac{2 + 40 L_{F_\mu}\eta_{k-1}}{T_{k-1}} \sum_{t=1}^{T_{k-1}} \mathbb{E}[\|\mathbf{z}_t - \nabla F_\mu(\mathbf{x}_t)\|^2] + \frac{2\mathbb{E}[\Delta_{k-1}^\mu]}{\eta_{k-1} T_{k-1}} + \frac{40 L_{F_\mu}\mathbb{E}[\Delta_{k-1}^\mu]}{T_{k-1}}$$

$$\leq \frac{(2 + 40 L_{F_\mu}\eta_{k-1})\mu\epsilon_{k-1}}{2} + \frac{2\epsilon_{k-1}}{\eta_{k-1} T_{k-1}} + \frac{40 L_{F_\mu}\epsilon_{k-1}}{T_{k-1}}$$

$$\leq \frac{773\mu\epsilon_k}{768} + \frac{\mu\epsilon_k}{2} + \frac{40 L_{F_\mu}\mu\epsilon_{k-1}}{73728 L_F^4}$$

$$\leq 2\mu\epsilon_k,$$

where the second inequality is due to $L_{F_\mu}\eta_{k-1} \leq (3/2)L_F\eta_{k-1} \leq 1/3072$.

Since $F_\mu(\mathbf{x}_k) \leq \bar{F}_\mu(\mathbf{x}_k)$ and $\inf_{\mathbf{x}\in\mathcal{X}} F_\mu(\mathbf{x}) = \inf_{\mathbf{x}\in\mathcal{X}} \bar{F}\mu(\mathbf{x})$, applying Lemma 3 we have

$$\mathbb{E}[F_\mu(\mathbf{x}_k) - \inf_{\mathbf{x}\in\mathcal{X}} F_\mu(\mathbf{x})] \leq \mathbb{E}[\bar{F}_\mu(\mathbf{x}_k) - \inf_{\mathbf{x}\in\mathcal{X}} \bar{F}_\mu(\mathbf{x})] \leq \frac{1}{2\mu}\mathbb{E}[\text{dist}(0, \hat{\partial}\bar{F}_\mu(\mathbf{x}_k))^2] \leq \frac{2\mu\epsilon_k}{2\mu} = \epsilon_k.$$

This complete the proof of this Lemma. □

### D.7 Proof of Theorem 4

*Proof.* Invoking Lemma 5, then after $K = \mathcal{O}(\log_2(\epsilon_1/\epsilon))$ stages, we have

$$\mathbb{E}[F_\mu(\mathbf{x}_K) - \inf_{\mathbf{x}\in\mathcal{X}} F_\mu(\mathbf{x})] \leq \epsilon_K = \frac{\epsilon_1}{2^{K-1}} = \epsilon.$$

Since $\sum_{k=1}^K 2^k = \mathcal{O}(1/\epsilon)$, the overall oracle complexity is

$$\sum_{k=1}^K T_k + \frac{48 L_F^2}{\mu\epsilon_1} \leq \mathcal{O}\left(\sum_{k=2}^K \max\left(\frac{1}{\mu\epsilon_k}, \frac{1}{\mu^{3/2}\sqrt{\epsilon_k}}\right)\right) + \frac{48 L_F^2}{\mu\epsilon_1}$$

$$\leq \mathcal{O}\left(\sum_{k=2}^K \max\left(\frac{2^k}{\mu}, \frac{\sqrt{2}^k}{\mu^{3/2}\sqrt{\epsilon_k}}\right)\right) + \frac{48 L_F^2}{\mu\epsilon_1}$$

$$\leq \mathcal{O}\left(\max\left(\frac{1}{\mu\epsilon}, \frac{1}{\mu^{3/2}\sqrt{\epsilon}}\right)\right).$$

This complete the proof. □

## E   Derivation of the Compositional Formulation

Recall the original KL-constrained DRO problem:

$$\min_{\mathbf{w}\in\mathcal{W}} \max_{\{\mathbf{p}\in\Delta_n : D(\mathbf{p},\mathbf{1}/n)\leq\rho\}} \sum_{i=1}^{n} p_i\ell_i(\mathbf{w}) - \lambda_0 D(\mathbf{p},1/n),$$

where $\Delta_n = \{\mathbf{p}\in\mathbb{R}^n : \sum_{i=1}^n p_i = 1, 0 \leq p_i \leq 1\}$, $D(\mathbf{p},1/n)$ is the KL divergence and $\lambda_0$ is a small positive constant.

In order to tackle this problem, let us first consider the robust loss

$$\max_{\{\mathbf{p}\in\Delta_n : D(\mathbf{p},1/n)\leq\rho\}} \sum_{i=1}^{n} p_i\ell_i(\mathbf{w}) - \lambda_0 D(\mathbf{p},1/n).$$

And then we invoke the dual variable $\lambda$ to transform this primal problem to the following form

$$\max_{\mathbf{p}\in\Delta_n} \min_{\bar{\lambda}\geq 0} \sum_{i=1}^{n} p_i\ell_i(\mathbf{w}) - \bar{\lambda}(D(\mathbf{p},1/n)-\rho) - \lambda_0 D(\mathbf{p},1/n).$$

Since this problem is concave in term of $\mathbf{p}$ given $\mathbf{w}$, by strong duality theorem, we have

$$\max_{\mathbf{p}\in\Delta_n} \min_{\bar{\lambda}\geq 0} \sum_{i=1}^{n} p_i\ell_i(\mathbf{w}) - \bar{\lambda}(D(\mathbf{p},1/n)-\rho) - \lambda_0 D(\mathbf{p},1/n)$$

$$= \min_{\bar{\lambda}\geq 0} \max_{\mathbf{p}\in\Delta_n} \sum_{i=1}^{n} p_i\ell_i(\mathbf{w}) - \bar{\lambda}(D(\mathbf{p},1/n)-\rho) - \lambda_0 D(\mathbf{p},1/n).$$

Let $\lambda = \bar{\lambda} + \lambda_0$, we have

$$\min_{\bar{\lambda}\geq 0} \max_{\mathbf{p}\in\Delta_n} \sum_{i=1}^{n} p_i\ell_i(\mathbf{w}) - \bar{\lambda}(D(\mathbf{p},1/n)-\rho) - \lambda_0 D(\mathbf{p},1/n)$$

$$= \min_{\lambda\geq\lambda_0} \max_{\mathbf{p}\in\Delta_n} \sum_{i=1}^{n} p_i\ell_i(\mathbf{w}) - \lambda(D(\mathbf{p},1/n)-\rho) - \lambda_0\rho.$$

Then the original problem is equivalent to the following problem

$$\min_{\mathbf{w}\in\mathcal{W}} \min_{\lambda\geq\lambda_0} \max_{\mathbf{p}\in\Delta_n} \sum_{i=1}^{n} p_i\ell_i(\mathbf{w}) - \lambda(D(\mathbf{p},1/n)-\rho) - \lambda_0\rho,$$

Next we fix $\mathbf{x} = (\mathbf{w}^\top, \lambda)^\top$ and derive an optimal solution $\mathbf{p}^*(\mathbf{x})$ which depends on $\mathbf{x}$ and solves the inner maximization problem. We consider the following problem

$$\min_{\mathbf{p}\in\Delta_n} - \sum_{i=1}^{n} p_i\ell_i(\mathbf{w}) + \lambda D(\mathbf{p},1/n).$$

which has the same optimal solution $\mathbf{p}^*(\mathbf{x})$ with our problem.

There are three constraints to handle, i.e., $p_i \geq 0, \forall i$ and $p_i \leq 1, \forall i$ and $\sum_{i=1}^n p_i = 1$. Note that the constraint $p_i \geq 0$ is enforced by the term $p_i\log(p_i)$, otherwise the above objective will become infinity. As a result, the constraint $p_i < 1$ is automatically satisfied due to $\sum_{i=1}^n p_i = 1$ and $p_i \geq 0$. Hence, we only need to explicitly tackle the constraint $\sum_{i=1}^n p_i = 1$. To this end, we define the following Lagrangian function

$$L_{\mathbf{x}}(\mathbf{p},\mu) = - \sum_{i=1}^{n} p_i\ell_i(\mathbf{w}) + \lambda\left(\log n + \sum_{i=1}^{n} p_i\log(p_i)\right) + \mu(\sum_{i=1}^{n} p_i - 1),$$

where $\mu$ is the Lagrangian multiplier for the constraint $\sum_{i=1}^n p_i = 1$. The optimal solutions satisfy the KKT conditions:

$$-\ell_i(\mathbf{w}) + \lambda\left(\log(p_i^*(\mathbf{x})) + 1\right) + \mu = 0 \text{ and } \sum_{i=1}^{n} p_i^*(\mathbf{x}) = 1.$$

From the first equation, we can derive $p_i^*(\mathbf{x}) \propto \exp(\ell_i(\mathbf{w})/\lambda)$. Due to the second equation, we can conclude that $p_i^*(\mathbf{x}) = \frac{\exp(\ell_i(\mathbf{w})/\lambda)}{\sum_{i=1}^n \exp(\ell_i(\mathbf{w})/\lambda)}$. Plugging this optimal $\mathbf{p}^*(\mathbf{w})$ into the inner maximization problem, we have

$$\sum_{i=1}^n p_i^*(\mathbf{x})\ell_i(\mathbf{w}) - \lambda \left( \log n + \sum_{i=1}^n p_i^*(\mathbf{w}) \log(p_i^*(\mathbf{w})) \right) = \lambda \log \left( \frac{1}{n} \sum_{i=1}^n \exp \left( \frac{\ell_i(\mathbf{w})}{\lambda} \right) \right),$$

Therefore, we get the following equivalent problem to the original problem

$$\min_{\mathbf{w} \in \mathcal{W}} \min_{\lambda \geq \lambda_0} \lambda \log \left( \frac{1}{n} \sum_{i=1}^n \exp \left( \frac{\ell_i(\mathbf{w})}{\lambda} \right) \right) + \lambda \rho.$$

which is Eq. (2) in the paper.

