# OpenReview forum: "Stochastic Constrained DRO with a Complexity Independent of Sample Size"
_TMLR — Accepted by TMLR_

### Review · Reviewer_YnCn · 2023-03-08

**Summary Of Contributions:**

Broadly, this paper is concerned with empirical risk minimization using non-traditional notions of "risk" such as distributionally robust optimization (DRO) risk. They consider a rather particular case of DRO-like risk, namely the "worst" re-weighted average loss, where the weights are constrained not to be too far from a uniform distribution.

My understanding of the main claims of the authors is as follows. The most orthodox approach to tackling objectives of the form given in their (1) is to use primal-dual optimizers "out of the box", and this works, but when the number of samples $n$ grows large, this becomes computationally prohibitive. A very similar problem has also been studied fairly recently by Qi et al. (NeurIPS 2021); this is essentially identical to an exponential smoothing or "tilting" approximation to the worst-case loss, e.g. as in Li et al. (ICLR 2021; "tilted ERM" paper). The key difference here is that here the authors consider a constraint on the $\\max$ over the weights inside their objective (1), which leads to a joint minimization in both the original parameters $\\mathbf{w}$ and the "tilting" parameter $\\lambda$; see their (2). Without additional assumptions, the joint problem need not be smooth (gradients can become arbitrarily sharp as $\\lambda$ gets too large). The most straightforward way of dealing with this is to just assume the losses are bounded - this is the approach taken by the authors. This lets the authors attack (2) as a constrained (but smooth) problem, to which (primal) stochastic gradient-based methods can be applied. The "inner" part of the compositional objective has a sum of $n$ terms, and so to avoid the $n$-dependent overhead at each iteration, they make use of a moving average estimator, as well-established but convenient technique.

**Audience:**

Yes

**Broader Impact Concerns:**

No concerns.

**Claims And Evidence:**

Yes

**Requested Changes:**

Please see my comments above.

**Strengths And Weaknesses:**

*Strengths:*

- If we start from The basic problem of interest and desire to have per-iteration overhead free of $n$ is stated clearly, and so are the techniques (and assumptions) employed by the authors to obtain algorithms with formal guarantees.
- The authors provide a concrete optimization procedure suited to their problem of interest that is straightforward to implement and enjoys competitive theoretical guarantees, up to the formal assumptions.

*Weaknesses:*

I found the motivations for their problem of interest really hard to parse. The "unconstrained" case (with $\\rho = \\infty$) is easy, just minimize the tilted loss in $\\mathbf{w}$. If we won't want the KL divergence to be too big, setting $\\lambda\_{0}$ accordingly large is thus the easy/implicit way of doing this. Adding the explicit condition in terms of $\\rho$ on top of this seems to add a lot of complexity with no real clear motivation. In paragraph 3 of the intro, the authors give three (!!) reasons for this, none of which are convincing or really have anything to do with the reason to use (1) instead of the tilted ERM objective (aka the objective of Qi et al. NeurIPS 2021).

Closely related to the preceding weakness, the writing is quite bloated. The "three reasons" just mentioned are not really relevant to the actual problem. The authors then give a central question (box at top of p.2), with once again "three" qualifications, that correspond roughly to the (i)~(iii) in the following paragraph. Furthermore, in the next paragraph, there are "three" challenges to (2). I get these challenges and the techniques to deal with them, but all these lists are conceptual overload. Moreover, it makes it hard to really see what is new and interesting here. Especially considering sections 4 and 5, the authors make comments about novelty in the context of stochastic learning algorithms for compositional functions - if there is such novelty here, why stick to the special case of DRO-like formulation? Again this clouds the overall story, making it hard to evaluate what is new/interesting.

I think the results here are strictly speaking novel and appear to be correct, with claims that are essentially sound (i.e., aligned with the results presented). On the other hand, the constrained version of the KL-regularized DRO objective here seems to come out of nowhere and be very poorly motivated, save for the fact that "there don't exist any results for this setting yet". Perhaps I have misunderstood something, but my feeling is that if the SO results for compositional functions stand on their own as contributions, there is no need to latch on to the rather unnatural constrained variant used here. Since the results appear solid, I am left on the borderline, but with some re-writing I think the paper might have an audience.

---

> ### Author Response · Authors · 2023-05-05
> **Thanks for the comments and suggestions!**
>
>
> > **Q1:** I found the motivations for their problem of interest really hard to parse. Adding the explicit condition in terms of
> $\rho$ on top of this seems to add a lot of complexity with no real clear motivation.
>
>
> **Answer:** Solving the constrained DRO with $\rho$ offers the capability of optimizing the temperature parameter $\lambda$ in Eq. (2).  $\lambda_0$ only gives a lower bound for the temperature parameter while $\rho$ determines the optimal temperature value. We have strengthened our motivation for the considered problem. First, the majority of learning theory for DRO is at the constrained DRO formulations instead of regularized DRO formulations, e.g., Duchi et al. 2023.   Second,  the log-sum-exponential form with a temperature parameter $\lambda$ is widely used in many ML/AI methods, e.g., contrastive self-supervised learning (Yuan et al. 2022). A recent paper Qiu et al. accepted to ICML 2023 has built on our technique for optimizing individualized temperature parameters in self-supervised learning and achieved better performance than existing methods, e.g., SimCLR, CLIP.  We have also added the above discussion in the revision to better motivate our problem (please refer to the place marked in red in the introduction section), and we hope this revision will be satisfactory to you.
>
> Duchi et al. (2023). Statistics of Robust Optimization: A Generalized Empirical Likelihood Approach.
>
> Yuan et al. (2022). Provable Stochastic Optimization for Global Contrastive Learning: Small Batch Does Not Harm Performance.
>
> Qiu et al. (2023)  Not All Semantics are Created Equal: Contrastive Self-supervised Learning with Automatic Temperature Individualization.
>
>
> > **Q2:** Especially considering sections 4 and 5, the authors make comments about novelty in the context of stochastic learning algorithms for compositional functions - if there is a such novelty here, why stick to the special case of DRO-like formulation? Again this clouds the overall story, making it hard to evaluate what is new/interesting.
>
> **Answer:** It is our belief that solving the constrained DRO by the proposed stochastic compositional optimization algorithms has practical value (cf response to Q1). The reviewer's comment "if there is a such novelty here, why stick to the special case of DRO-like formulation?" sounds like the reviewer does not agree the novelty here, which is a pity.  It is important to note that our algorithmic development and theoretical analysis in Sec. 4 and Sec. 5 are written for more abstract problems and broader audience. We are not hiding such generality.

---

### Review · Reviewer_6JEJ · 2023-04-01

**Summary Of Contributions:**

This paper studies a constrained DRO with a KL regularizer and constraint, where the main goal of the paper is to derive a solver with optimal complexity for both convex and nonconvex losses that is independent of the sample size. The main tool in this paper is to rewrite the dual form of the problem, which leads to a primal only compositional problem, and then two algorithms (one of which is the accelerated variant) are presented for solving the problem. The authors analyze the convergence rate and compare that against several existing solvers.

**Audience:**

Yes

**Claims And Evidence:**

Yes

**Requested Changes:**

Overall, this is a paper on an important research area with a rich set of results. However, there are several issues with framing and experimental/theoretical comparisons that have been pointed out in the weaknesses, which I would like to see addressed in a revision.

**Strengths And Weaknesses:**

Strengths

* The paper studies an important problem on distributionally robust optimization and provides principled methods to solve it.

* The paper provides convergence guarantees for the proposed solver, and also provides an accelerated version with optimal oracle complexity. The paper studies such guarantees both for convex and non-convex losses.

* The paper shows several (large-scale) experiments where they show their proposed algorithms cobverges faster and achieves a better solution compared to several baselines.

Weaknesses

* It is not clear what the objective in (1) offers beyond the unconstrained DRO problem. In particular, the impact of $\rho$ and $\lambda_0$ is similar practically, and I am not sure if $\rho$ has an operational meaning for practitioners that $\lambda_0$ cannot offer.

* The objective in (1) is also slightly different from those of the prior work in their Table 1 (Levy et al; Namkoong & Duchi et al); hence I am not sure if the comparison of the convergence rate is apples-to-apples.

* The duality between constrained DRO and the formulation in (2) that is at the heart of this paper is observed previously in several works and has been extensively studied, cf. (Föllmer & Knispel, 2011)(Shapiro et al), and has been recently of interest by others, e.g., (Qi et al., 2020) and (Li et al., 2021). This has not been sufficiently highlighted here as what is being offered on top of the rich literature.

Föllmer, Hans, and Thomas Knispel. "Entropic risk measures: Coherence vs. convexity, model ambiguity and robust large deviations." Stochastics and Dynamics 11, no. 02n03 (2011): 333-351.

Shapiro, Alexander, Darinka Dentcheva, and Andrzej Ruszczynski. Lectures on stochastic programming: modeling and theory. Society for Industrial and Applied Mathematics, 2021.

Qi, Qi, Yi Xu, Rong Jin, Wotao Yin, and Tianbao Yang. "Attentional biased stochastic gradient for imbalanced classification." arXiv preprint arXiv:2012.06951 (2020).

Li, Tian, Ahmad Beirami, Maziar Sanjabi, and Virginia Smith. "On tilted losses in machine learning: Theory and applications." arXiv preprint arXiv:2109.06141 (2021).

* A simple solver for (2) would be to use the solvers by (Qi et al, 2020, Li et al, 2021) and add an additional gradient step on $\lambda$ (perhaps with a momentum term). It is unclear what the compositional view offers on top of this simple solver theoretically and empirically. In fact, in Eq (6), the gradients to update $w$ is the same as that of (Qi et al, 2020, Li et al, 2021) if we ignore the moving average on $v$ because $\nabla f_{\lambda} (s)$ is just $\lambda * (1/s).$



* Given the rich experimental setup of (Li et al. 2021), it is not clear why that baseline is missing.

* It seems to me that for non-convex problem in (2) can be solved for all $\lambda_0$ and should not need $\lambda_0>0$. Is this constraint needed?

* The formula in (2) has a resemblance to Entropic Value at Risk (Ahmadi-Javid, 2021) and Tilted Value at Risk, see (Li et. al, 2021), as both of those works offer recipes to optimize $\lambda$ in addition to the tilted risk. That connection has not been explored.

Ahmadi-Javid, Amir. "Entropic value-at-risk: A new coherent risk measure." Journal of Optimization Theory and Applications 155 (2012): 1105-1123.

---

> ### Author Response · Authors · 2023-05-05
> **Thanks for the comments and suggestions (1)**
>
> > **Q1**: It is not clear what the objective in (1) offers beyond the unconstrained DRO problem. In particular, the impact of and is similar practically, and I am not sure if has an operational meaning for practitioners that cannot offer.
>
> **A** Solving the constrained DRO with $\rho$ offers the capability of optimizing the temperature parameter $\lambda$ in Eq. (2).  $\lambda_0$ only gives a lower bound for the temperature parameter while $\rho$ determines the optimal temperature value. We have strengthened our motivation for the considered problem. First, the majority of learning theory for DRO is at the constrained DRO formulations instead of regularized DRO formulations, e.g., Duchi et al. 2023.   Second,  the log-sum-exponential form with a temperature parameter $\lambda$ is widely used in many ML/AI methods, e.g., constrastive self-supervised learning (Yuan et al. 2022). A recent paper Qiu et al. accepted to ICML 2023 has built on our technique for optimizing individualized temperature parameter in self-supervised learning and achieved better performance than existing methods, e.g., SimCLR, CLIP.
>
> Duchi et al. (2023). Statistics of Robust Optimization: A Generalized Empirical Likelihood Approach.
>
> Yuan et al. (2022). Provable Stochastic Optimization for Global Contrastive Learning: Small Batch Does Not Harm Performance.
>
> Qiu et al. (2023)  Not All Semantics are Created Equal: Contrastive Self-supervised Learning with Automatic Temperature Individualization.
> >**Q2:** The objective in (1) is also slightly different from those of the prior work in their Table 1 (Levy et al; Namkoong \& Duchi et al); hence I am not sure if the comparison of the convergence rate is apples-to-apples.
>
> **A** : Levy et al. (2020) established a convergence rate of the general form of the DRO problem with both constraints and regularization, which cover the specific form of our paper. Besides, although the objective of Namkoong \& Duchi (2016) is slightly different from ours, their primal-dual method can be applied to our KL-constraint DRO problem. Therefore, the comparison is apples-to-apples.
> > **Q3:**. The duality between constrained DRO and the formulation in (2) that is at the heart of this paper has been observed previously in several works and has been extensively studied, cf. (Föllmer & Knispel, 2011)(Shapiro et al), and has been recent of interest by others, e.g., (Qi et al., 2020) and (Li et al., 2021). This has not been sufficiently highlighted here as what is being offered on top of the rich literature.
>
> **A**: Yes, the establishment of the duality between constrained DRO and the formulation in (2)  itself actually does not require much technical effort. However, with the compositional formulation in hand, there are several challenges to be addressed for achieving optimal complexities for both convex and non-convex loss functions. And the main contribution of our paper is to tackle these technical challenges and achieve optimal complexities in both convex and non-convex settings.
> The first difficulty is how to tackle the unbounded smoothness constant of the objective. To address this difficulty, we derive an upper bound of the Lagrangian multiplier $\lambda$ in Lemma 1, which allows us to establish the bounded smoothness constant of the objective in Lemma 1. The second difficulty comes from the introduced domain constraint on $\lambda$ for deriving the optimal complexities. In an existing analysis of compositional optimization algorithms, either (i) the problem is assumed to be unconstrained, e.g., Qi et al (2021), or (ii) the complexity is sub-optimal, e.g., Ghadimi et al. (2020), or (iii) the problem is restricted, e.g., the outer function $f$ is convex and non-decreasing as assumed in Zhang \& Lan (2020). To the best of our knowledge, this is {\bf the first result } for stochastic compositional optimization with a domain constraint that enjoys the optimal complexities for both convex and non-convex objectives.
> We add the above discussion in the revision,  please refer to the place marked in red on page 3.
> >**Q4:** A simple solver for (2) would be to use the solvers by (Qi et al, 2020, Li et al, 2021) and add an additional gradient step on $\lambda$ (perhaps with a momentum term). It is unclear what the compositional view offers on top of this simple solver theoretically and empirically. In fact, in Eq (6), the gradients to update $w$ is the same as that of (Qi et al, 2020, Li et al, 2021) if we ignore the moving average on $v$ because $\nabla f_{\lambda}(s)$ is just $\lambda * (1/s)$.
>
> **A:** It is true that the algorithmic update is similar to prior works. However, the key differences lie at the analysis of the algorithms for solving constrained compositional optimization problems and deriving the optimal complexities for both non-convex and convex functions. Please refer to discussions under Theorem 1, 2, Corollary 2 for in-depth discussion.

---

> > ### Comment · Reviewer_6JEJ · 2023-05-05
> > **Clarification on Q6**
> >
> > Thanks for your thorough responses, which I will go through as soon as possible. However, I thought I send you a clarification point on **Q6** as soon as possible.
> >
> > If the loss function, $\ell$ is smooth but non-convex, then (2) is smooth but non-convex for all $\lambda$, see (Li et al., 2021). I wonder if your analysis would cover the whole range of $\lambda$ and not just $\lambda > \lambda_0$ for non-convex functions. Of course, even if loss is convex, it may no longer be convex for $\lambda < \lambda_0$ so the analysis in the convex case does not necessarily extend here.

---

> > > ### Author Response · Authors · 2023-05-05
> > > **Thank you for clarification! Confusion about smoothness.**
> > >
> > > We politely disagree the comment "If the loss function,  $\ell$ is smooth but non-convex, then (2) is smooth but non-convex for all $\lambda$, see (Li et al., 2021)."    If $\lambda=0$, the objective in (2) becomes $\max_i \ell_i(\mathbf w)$, even if $\ell$ is smooth, the resulting objective is still non-smooth.  The reviewer refers us to Li et al. 2021. If the reviewer means the paper "On Tilted Losses in Machine Learning: Theory and Applications". Indeed, their results confirm (2) is non-smooth when $\lambda=0$. If you check Lemma 3 in their paper, they proved their objective given fixed $t=1/\lambda$ is smooth with a smoothness parameter $\beta(t)$. However, they did not prove that $\beta(t)$ is bounded when $t\rightarrow\infty$. Indeed, as they mentioned in the sentence below Lemma 3 "$\beta(t)$ is bounded for all negative $t$ and moderately positive $t$, where it scales linearly with $t$ when $t\rightarrow \infty$." Hence, when $t\rightarrow \infty$ (equivalently $\lambda\rightarrow 0$), the smoothness parameter $\beta(t)$ becomes infinity. This is not the typical smoothness assumption that is used for convergence analysis of non-convex functions, i.e.,  the smoothness parameter is a constant.
> > >
> > > Please let us know if you agree with us or point to us the related result in Li et al 2021 that proves the smoothness parameter of the objective is finite when $t\rightarrow \infty$.

---

> > > > ### Comment · Reviewer_6JEJ · 2023-05-05
> > > > **Agreed**
> > > >
> > > > You are right that $\lambda \to 0^+$ will make the overall objective non-smooth! I apologize for my confusion as $\lambda$ in (2) corresponds to $1/t$ in (Li et al. 2021).
> > > >
> > > > But I think you may still be able to explore $\lambda<0,$ which corresponds to $t <0$ in (Li et al., 2021), but I am not sure how much added value that may provide.
> > > >
> > > > Thanks,\
> > > > Reviewer 6JEJ

---

> > > > > ### Author Response · Authors · 2023-05-06
> > > > > **Extendable but with extra efforts**
> > > > >
> > > > > Optimizing for $\lambda<0$ can be handled but with some extra efforts, which is less challenging than the considered problem.
> > > > >
> > > > > First, it is not reasonable to simply solve eq (2) in the paper with $\lambda<0$ as it has no physical meaning.
> > > > >
> > > > > In order to derive a meaningful formulation with $\lambda<0$, we need to consider the min-min DRO formulation:
> > > > > $$\min_{\mathbf w}\min_{\mathbf p\in\Delta, D(\mathbf p, 1/n)\leq \rho} \sum_i p_i \ell_i $$
> > > > > This formulation is meaningful as $\rho=0$ recovers the empirical risk and $\rho>\log (n)$ recovers the min-loss. The min-loss is shown to be robust to the noisy labeled data (Majidi et al. 2021, Qi et al. 2020a).  According to Lagrangian dual theory to handle $\min_{\mathbf p\in\Delta, D(\mathbf p, 1/n)\leq \rho} \sum_i p_i \ell_i$, we can derive the following equivalent formulation:
> > > > > $$\min_{\mathbf w}\max_{\lambda<0} \lambda \log \frac{1}{n}\sum_{i=1}^n\exp(\frac{\ell_i}{\lambda}) + \lambda\rho.$$ Hence, the problem becomes non-convex concave optimization with a compositional objective. The objective is smooth with respect to both $\mathbf w$ and $\lambda$. The smooth non-convex concave optimization has been studied widely literature. But the difference here is that the objective is compositional.  A similar problem has been tackled in the literature (Yuan et al. 2022). We expect that one can combine the analysis for non-convex concave optimization (e.g., Rafique et al, Zhang et al.) and that for compositional optimization to derive an efficient algorithm and convergence guarantee. However, including such development in the present paper would be beyond our scope.
> > > > >
> > > > >
> > > > > Majidi et al. 2021. Exponentiated Gradient Reweighting for Robust Training Under Label Noise and Beyond
> > > > >
> > > > > Yuan et al. 2022 (ICLR). COMPOSITIONAL TRAINING FOR END-TO-END DEEP AUC MAXIMIZATION.
> > > > >
> > > > > Rafique et al. Weakly-Convex Concave Min-Max Optimization: Provable Algorithms and Applications in Machine Learning.
> > > > >
> > > > > Zhang et al. SAPD+: An Accelerated Stochastic Method for Nonconvex-Concave Minimax Problems.

---

> ### Author Response · Authors · 2023-05-05
> **Thanks for the comments and suggestions (2)**
>
> >**Q5:** Given the rich experimental setup of (Li et al. 2021), it is not clear why that baseline is missing.
>
> **A:**
> We have added the baseline of ABSGD ( Li et al. 2021, and Qi et al 2020)on ImageNet-LT and iNaturalist2018 and in Table 3 for the non-convex setting. The proposed algorithm achieves better results
>
> >**Q6:**  It seems to me that for non-convex problem in (2) can be solved for all $\lambda_0$ and should not need $\lambda_0 > 0$ Is this constraint needed?
>
> **A:** It is needed for convergence analysis of non-convex functions. Otherwise, it is non-smooth and non-convex problems and difficult to derive a non-asymptotic convergence rate as in the paper.
>
>
> >**Q7:** The formula in (2) has a resemblance to Entropic Value at Risk (Ahmadi-Javid, 2021) and Tilted Value at Risk, see (Li et. al, 2021), as both of those works offer recipes to optimize $\lambda$ in addition to the tilted risk.
>
> **A:** We have added a citation to Ahmadi-Javid 2012. Although both works have considered formulations for optimizing the temperature parameter $\lambda$, none of them have proposed any efficient algorithms for solving the resulting objective with non-asymptotic convergence guarantee.

---

### Review · Reviewer_J7uW · 2023-04-20

**Summary Of Contributions:**

Due to its promising performance on noisy labels, imbalanced data, and adversarial data, large-scale DRO optimization is gaining interest. Primal-dual algorithms solve DRO problems but incur overhead for managing a $n$-dimensional dual variable, where $n$ is the sample size. For large-scale deep learning, where $n$ could be millions or billions, this is undesirable. Thus, dual-free methods for DRO issues have become popular. This work develops  dual-free algorithms for solving the following constrained DRO problem,
$$
\min _{\mathbf{w} \in \mathcal{W}} \max _{\left}{\mathbf{p} \in \delta_n: D(\mathbf{p}, \mathbf{1} / n) \leq \rho\right\}} \sum_{i=1}^n p_i \ell_i(\mathbf{w})-\lambda_0 D(\mathbf{p}, \mathbf{1} / n),
$$
where $\mathbf{w}$ denotes the model parameter, $\mathcal{W}$ is a closed convex set, $\Delta_n=\left\{\mathbf{p} \in \mathbb{R}^n: \sum_{i=1}^n p_i=1, p_i \geq 0\right\}$ denotes an $n$-dimensional simplex, $\ell_i(\mathbf{w})$ denotes a loss function on the $i$-th data, $D(\mathbf{p}, \mathbf{1} / n)=\sum_{i=1}^n p_i \log \left(p_i n\right)$ is the Kullback-Leibler (KL)-divergence measure between $\mathbf{p}$ and uniform probabilities $\mathbf{1} / n \in \mathbb{R}^n$, and $\rho$ is the constraint parameter, and $\lambda_0 > 0$ is a small constant.  KL regularization ensures smoothness of the objective function.

The main contributions are
- derivation of  an equivalent primal-only formulation that is compositional;
- development of two algorithms for non-convex losses and extending them for convex losses;
- establishing an optimal complexity for both convex and non-convex losses. In particular, for a non-convex and smooth loss function $\ell_i(\mathbf{w})$, the algorithm achieves an oracle complexity of $\widetilde{\mathcal{O}}\left(1 / \epsilon^3\right)^1$ for finding a $epsilon$-stationary solution, and for a convex and smooth loss function, we achieve $\mathcal{O}\left(1 / \epsilon^2\right)$ for finding a $epsilon$-optimal solution. These results match primal-dual algorithms best complexities but have a per-iteration complexity of $\mathcal{O}(d)$, regardless of sample size $n$.


**Audience:**

Yes

**Claims And Evidence:**

Yes

**Requested Changes:**

- include a discussion of the equivalence (1)-(3) in the main text: this is really one of the essential points in your contribution, even if the derivation is quite elementary.
- The domain of model parameter $\mathcal{W}$ is bounded by $R$, i.e., for all $\mathbf{w} \in \mathcal{W}$, we have $\|\mathbf{w}\| \leq R$: Clumsy, to be reformulated
- I do not like $$i \sim \mathcal{D}$" in the assumption. Say for all $i \in \mathca{D}$ this is more correct [of course, I have understood what is meant]
- The descriptions of the algorithm should include all the updates.  We are half way there, we would prefer to have all the equations in front of us

**Strengths And Weaknesses:**

Strengths:
- The motivations, constructions and the proofs are very understandable, and the paper itself is written in an educational manner. I was able to navigate my way through the numerous steps of the constructions  without much problem [My only complaint is that the description of the equivalence between (1) and (2) is included in the notations paragraph, whereas it is one of the key here]
- The effectiveness of the proposed algorithm for solving non-convex and convex constrained DRO problems is clearly demonstrated by empirical work.
Weaknesses
- A relative weakness, but the subject has given rise to a great deal of work, the proposed solution is an improvement of the existing but does not "change" the game.

---

> ### Author Response · Authors · 2023-05-05
> **Thanks for the comments and suggestions**
>
> > A relative weakness, but the subject has given rise to a great deal of work, the proposed solution is an improvement of the existing but does not "change" the game.
>
> **Answer:** We agree with the reviewer this is not a game changer paper. However, we would like to point out this paper has made meaningful progress not only for theoretical analysis of compositional functions but also for solving practical problems.  It is notable that the log-sum-exponential form with a temperature parameter $\lambda$ is widely used in many ML/AI methods, e.g., constrastive self-supervised learning (Yuan et al. 2022). A recent paper Qiu et al. accepted to ICML 2023 has built on our technique for optimizing individualized temperature parameter in self-supervised learning and achieved better performance than existing methods, e.g., SimCLR, CLIP.
>
> >**Requested Changes:**
> 1. include a discussion of the equivalence (1)-(3) in the main text: this is really one of the essential points in your contribution, even if the derivation is quite elementary.
>
> **Answer**: we have modified in the paper
> 2. The domain of model parameter $\mathcal{W}$ is bounded by R, i.e., for all $\mathbf{w} \in \mathcal{W}$, we have $|\mathbf{w}|\leq R$: Clumsy, to be reformulated
>
>  **Answer** : We have changed this to ``The domain of model parameter $\mathcal{W}$ is bounded  such that there exists $R>0$ it holds $\|\mathbf w\|\leq R$  for any $\mathbf w\in\mathcal W$".
> 3. I do not like $i \sim \mathcal{D}$ in the assumption, say $\forall \in\mathcal{D}$ this is more correct [of course, I have understood what is meant.
>
>  **Answer** : We have changed it.
> 4.  The descriptions of the algorithm should include all the updates. We are halfway there, we would prefer to have all the equations in front of us.
>
> **Answer** : We have changed it.

---

### Review · Reviewer_SDL7 · 2023-04-23

**Summary Of Contributions:**

This paper investigates a class of constrained Distributionally Robust Optimization (DRO) problems with KL divergence and proposes a dual-free stochastic algorithm for both non-convex and convex losses. The paper achieves nearly optimal complexity for non-convex losses and optimal complexity for convex losses, independent of the sample size. Experiments are presented on solving imbalanced classification problems, demonstrating the performance of the proposed algorithms in terms of convergence speed and generalization ability.

**Audience:**

Yes

**Claims And Evidence:**

Yes

**Requested Changes:**

Goes back to my technical concern, can you further improve the complexity bound dependence on $\lambda_0$?

Stronger motivation to study the proposed problems should be provided.





**Strengths And Weaknesses:**

Strengths

Instead of directly solving the constrained DRO problem, the paper focuses on an equivalent formulation and solves a composite optimization problem. The paper derives an upper bound for lambda in eq(2) to address the challenge of nonsmoothness of the objective function. Stochastic algorithms are proposed for solving the composite optimization problem of nonconvex loss and are further extended to the convex setting by applying a multistage restart strategy. The paper uses the moving average technique to compute stochastic estimators for both gradients and function values of the inner function. The proposed algorithms achieve complexity results independent of sample size, which is also comparable to state-of-the-art algorithms for solving constrained DRO problems with KL divergence.

 Weaknesses

- This paper considers a particular formulation of the DRO problem eq(1), which has a KL divergence constraint as well as a KL regularization term in the objective function. This KL regularization term helps ensure the smoothness of the objective function and thus achieves a fast convergence rate. The paper argues that this new form is not studied in the literature. However, the intuition and benefits of doing this are still not very clear to me, as the KL regularization term basically has the same effect as the KL constraint. Why having both of them?
- It appears (from Lemma 7) that the condition number crucially depends $L_0$, which further **exponentially depends** on $1/\lambda_0$. Although I am not an expert in DRO, this looks surprising to me. For example, in the experiment you take $\lambda_0=1e-3$. In this case, shouldn't the condition number make the complexity bound trivial?
- The main techniques used in this paper include estimating inner function values and gradients by moving average and a restart strategy for solving problems with convex losses. The accelerated version of the algorithm is inspired by Qi et al. (2021), and other techniques are not highly innovative.

---

> ### Author Response · Authors · 2023-05-05
> **Thanks for the comments and suggestions (1)**
>
> >  **Q1:** This paper considers a particular formulation of the DRO problem eq(1), which has a KL divergence constraint as well as a KL regularization term in the objective function. This KL regularization term helps ensure the smoothness of the objective function and thus achieves a fast convergence rate. The paper argues that this new form is not studied in the literature. However, the intuition and benefits of doing this are still not very clear to me, as the KL regularization term basically has the same effect as the KL constraint. Why have both of them?
>
> **Answer:**  The KL constraint and KL regularization serves difference purposes in our paper.  The KL constraint enables the algorithm to learn the optimal temperature parameter $\lambda$ is a certain range. The KL regularization with a small regularization parameter $\lambda_0$  smoothes the objective, which also imposes a lower bound of  the temperature parameter $\lambda$. It follows the famous Nesterov's smoothing technique of non-smooth functions (Nesterov 2004). This technique is also commonly used for solving DRO problems with constraint. For example, Levy et al.(2020)  studied the KL-regularization CVaR-constraint DRO as a smoothed version of the CVaR-constraint DRO (for more detail of this technique, please refer to the literature of optimal transport distances, e.g., Cuturi 2013). It is notable that our technique could be useful for optimizing individualized temperature parameter in contrastive self-supervised learning (Yuan et al 2022). This has been proved to be useful in a recent paper Qiu et al. accepted to ICML 2023.
>
> Reference:  Nesterov (2004). Smooth minimization of non-smooth functions.
>
> Yuan et al. (2022). Provable Stochastic Optimization for Global Contrastive Learning: Small Batch Does Not Harm Performance.
>
> Qiu et al. (2023)  Not All Semantics are Created Equal: Contrastive Self-supervised Learning with Automatic Temperature Individualization.'
>
> >**Q2:**  It appears (from Lemma 7) that the condition number crucially depends on $L_0$, which further exponentially depends on $1/\lambda_0$, Although I am not an expert in DRO, this looks surprising.
>
> **Answer:**  1. Yes, as we pointed out in our paper (the footnote of page 5),  the smoothness constant exponentially depends on $1/\lambda_0$. \item  Such exponential dependence on problem parameters also occurred in some other stochastic methods solving constrained DRO, like Dual SGM in Levy et al. (2020). To address the KL-constrained DRO problem, Dual SGM has the same exponential smoothness constant dependent on the problem parameter, while FastDRO requires an even stronger smoothness assumption (Assumption A2 in Levy et al.(2020)) that inverse cumulative density function of the loss $\ell_i$ are Lipschitz continuous (as mentioned in Levy et al.(2020), as pointed out in their paper ``This is a rather strong assumption that fails whenever $S$ is discrete or $\ell(x; S)$ is distributed as two separate bulks."
>
>  2. We conjecture that the exponential dependence is due to the technical challenge of dealing with non-convex functions and it is an artifact of the deficiency of analysis for convex functions. For convex problems, we expect one could derive much better dependence on $\lambda_0$. In particular, Zhang \& Lan shows that the primal-only algorithm for a compositional function (e.g. Algorithm 1 in the paper) is equivalent to a primal-dual algorithm for solving a corresponding min-max problem. Their analysis yields a polynomial dependence on problem parameters. However, we cannot directly apply their theory to our problem as their theory requires the outer function to be convex, which is not the case in our problem $f(\cdot)=\lambda\log(\cdot)$. However, we believe it is possible to derive a better rate by decomposing the first term in our objective into three level nested function, $f_1\circ f_2\circ g$, where $f_1(\cdot)  =-\lambda\log(\cdot)$ and $f_2(\cdot)=1/\cdot$. In this way, both $f_1$ and $f_2$ are monotonically decreasing and convex, it will be possible to derive a fast rate with polynomial dependence on $\lambda_0$ following Zhang \& Lan's theory. However, we consider this to be non-trivial and a future work.
>
> Zhang \& Lan (2022). Optimal Algorithms for Convex Nested Stochastic Composite Optimization.

---

> ### Author Response · Authors · 2023-05-05
> **Thanks for the comments and suggestions (2)**
>
> > **Q3:** In the experiment, you take $\lambda_0 = 1e-3$. In this case, shouldn't the condition number make the complexity bound trivial?
>
> **Answer:** In the experiment, we did not see any problem of using such a small value of $\lambda_0$. There would be a gap between theory and practice.
>
>
> >**Q4**: The main techniques used in this paper include estimating inner function values and gradients by moving averages and a restart strategy for solving problems with convex losses. The accelerated version of the algorithm is inspired by Qi et al. (2021), and other techniques are not highly innovative.
>
> **Answer:** We agree that the algorithmic technique is similar to existing works, e.g., (Ghadimi et al. 2020). But our analysis is different due to the constraint in the objective $\lambda\in\Omega$ and this difference is the key to derive the optimal rate for a constrained convex compositional problem. In particular, we directly prove the convergence of the regular subgradient of the objective function.  For a detailed discussion on the technical novelty of our analysis,  please refer to the discussions under Theorem 1, Theorem 2, and Corollary 2 in the paper.
>
> > **Requested Changes:**  This goes back to my technical concern, can you further improve the complexity-bound dependence on $\lambda_0$?
> 	Stronger motivation to study the proposed problems should be provided.
>
> **Answer:** We have strengthened the motivation by pointing out the proposed technique can be used for optimizing the temperature parameter is a vast literature of ML using temperature scaled softmax loss, which is similar to our objective. In terms of technical concern, please refer to our response to Q2.

---

### Decision · Action_Editors · 2023-05-31

**Recommendation:** Accept with minor revision

**Comment:**

This paper studies a constrained DRO with a KL regularizer and constraint, where the main goal of the paper is to derive a solver with optimal complexity for both convex and nonconvex losses that is independent of the sample size. In general, the paper is well written, technically sounded, and experimental results are convincing.  It has nice theoretical findings that could be useful beyond the constrained DRO problem. During the discussion, the authors have successfully addressed some of reviewer concerns. As a result, three reviewers are positive on the paper.  There are still some minor comments which can be addressed by minor revision.
- While the current limited set of experimental results don't fully support the claim that the formulation could be used to tune
λ in tilted loss (2), I think it is okay if the authors add a disclaimer/limitation that this is a motivation for the work, and this paper takes a step towards that goal.
- There are still some issues with the density of the material, the authors have addressed some of these points in the revised version.

**Audience:**

This paper proposes an optimization algorithm with optimal complexity for both convex and nonconvex losses.

**Claims And Evidence:**

This paper investigates a class of constrained Distributionally Robust Optimization (DRO) problems with KL divergence and proposes a dual-free stochastic algorithm for both non-convex and convex losses. The algorithm is well grounded with convincing experiment validations.